# Understanding Benign Overfitting in Gradient-based Meta Learning

**Lisha Chen**
Rensselaer Polytechnic Institute
Troy, NY, USA
chenl21@rpi.edu

**Songtao Lu**
IBM Research
Yorktown Heights, NY, USA
songtao@ibm.com

**Tianyi Chen**
Rensselaer Polytechnic Institute
Troy, NY, USA
chentianyi19@gmail.com

## Abstract

Meta learning has demonstrated tremendous success in few-shot learning with limited supervised data. In those settings, the meta model is usually overparameterized. While the conventional statistical learning theory suggests that overparameterized models tend to overfit, empirical evidence reveals that overparameterized meta learning methods still work well – a phenomenon often called "benign overfitting." To understand this phenomenon, we focus on the meta learning settings with a challenging bilevel structure that we term the gradient-based meta learning, and analyze its generalization performance under an overparameterized meta linear regression model. While our analysis uses the relatively tractable linear models, our theory contributes to understanding the delicate interplay among data heterogeneity, model adaptation and benign overfitting in gradient-based meta learning tasks. We corroborate our theoretical claims through numerical simulations.

## 1  Introduction

Meta learning, also referred to as "learning to learn", usually learns a prior model from multiple tasks so that the learned model is able to quickly adapt to unseen tasks [43, 26]. Meta learning has been successfully applied to few-shot learning learning [2, 13], image recognition [52], federated learning [29], reinforcement learning [21] and communication systems [10]. While there are many exciting meta learning methods today, in this paper, we will study a representative meta learning setting where the goal is to learn a shared initial model that can quickly adapt to task-specific models. This adaptation may take an explicit form such as the output of one gradient descent step, which is referred to as the model agnostic meta learning (MAML) method [21]. Alternatively, the adaptation step may take an implicit form such as the solution of another optimization problem, which is referred to as the implicit MAML (iMAML) method [39]. Since both MAML and iMAML will solve a bilevel optimization problem, we term them the *gradient-based meta learning* thereafter. In many cases, overparameterized models are used as the initial models in meta learning for quick adaptation. For example, Resnet-based MAML models typically have around 6 million parameters, but are trained on 1-3 million meta-training data [12]. Training such initial models is often difficult in meta learning because the number of training data is much smaller than the dimension of the model parameter.

Previous works on meta learning mainly focus on addressing the optimization challenges or analyzing the generalization performance with sufficient data [18, 19, 14]. Different from these works, we are particularly interested in the generalization performance of the sought initial model in practical scenarios where the total number of data from all tasks is *smaller than* the dimension of the initial model, which we term *overparameterized meta learning*. In those overparameterized regimes, the generalization error of meta learning models is not fully understood. Motivated by this, we ask:

*If and when overparameterized meta learning models would lead to overfitting, provably?*

36th Conference on Neural Information Processing Systems (NeurIPS 2022).

Empirical studies have demonstrated that the MAML with overparameterized model generally performs better than MAML with underparameterized model [12] – a phenomenon often called "benign overfitting." To show this, we plot in Figure 1 the empirical results from Table A5 in [12]. Complementing this, we take an initial step by answering this theoretical question in the meta linear regression setting.

## 1.1 Prior art

We review prior art that we group in the following three categories.

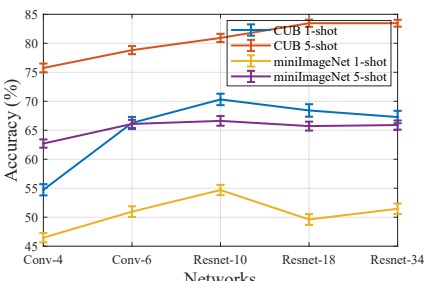

Figure 1: Accuracy vs networks with increasing dimensions for MAML on few-shot image classification with different datasets [12].

**Benign overfitting analysis.** The empirical success of overparameterized deep neural networks has inspired theoretical studies of overparameterized learning. The most closest line of work is *benign overfitting* in linear regression [5], which provides excess risk that measures the difference between expected population risk of the empirical solution and the optimal population risk. Analysis of overparameterized linear regression model with the minimum-norm solution. It concludes that certain data covariance matrices lead to benign overfitting, explaining why overparameterized models that perfectly fit the noisy training data can work well during testing. The analysis has been extended to ridge regression [46], multi-class classification [50], and adversarial learning with linear models [8]. While previous theoretical efforts on benign overfitting largely focused on linear models, most recently, the analysis of benign overfitting has been extended to two-layer neural networks [7, 33, 22]. However, existing works mainly study benign overfitting for empirical risk minimization problems, rather than bilevel problems such as gradient-based meta learning, which is the focus of this work.

**Meta learning.** Early works of meta learning build black-box recurrent models that can make predictions based on a few examples from new tasks [43, 26, 2, 13], or learn shared feature representation among multiple tasks [44, 48]. More recently, meta learning approaches aim to find the initialization of model parameters that can quickly adapt to new tasks with a few number of optimization steps such as MAML [21, 38, 41]. The empirical success of meta learning has also stimulated recent interests on building the theoretical foundation of meta learning methods.

**Generalization of meta learning.** The *excess risk*, as a metric of generalization ability of gradient-based meta learning has been analyzed recently [15, 3, 9, 49, 4, 19]. The generalization of meta learning has been studied in [32] in the context of mixed linear regression, where the focus is on investigating when abundant tasks with small data can compensate for lack of tasks with big data. Generalization performance has also been studied in a relevant but different setting - representation based meta learning [14, 17]. Information theoretic bounds have been proposed in [30, 11], which bound the generalization error in terms of mutual information between the input training data and the output of the meta-learning algorithms. The PAC-Bayes framework has been extended to meta learning to provide a PAC-Bayes meta-population risk bound [1, 40, 16, 20]. These works mostly focus on the case where the meta learning model is underparameterized; that is, the total number of meta training data from all tasks is larger than the dimension of the model parameter. Recently, *overparameterized* meta learning has attracted much attention. Bernacchia [6] suggests that in overparameterized MAML, negative learning rate in the inner loop is optimal during meta training for linear models with Gaussian data. Sun et al. [45] shows that the optimal representation in representation-based meta learning is overparameterized and provides sample complexity for the method of moment estimator. Besides our work, a concurrent work [27] also studies a common setting where the meta learning models incur overparameterization in the meta level, and we both cover the nested MAML method. However, the two studies differ in terms of how the empirical solution of the meta parameter is obtained. In our case, we consider the minimum $\ell$-2 norm solution, while [27] consider the solution trained with $T$-step stochastic gradient descent (SGD). Furthermore, our analysis covers both MAML and iMAML, while [27] only considers MAML.

Table 1: A comparison with closely related prior work on meta learning with linear models. "Reps." and "Gradient" refer to representation based methods and gradient-based methods; "Per-task" refers to the per-task level overparameterization and "Meta" refers to the meta level overparameterization.

| Prior work | Type of meta learning | | Overparameterization | | Methods | Focus of analysis |
|---|---|---|---|---|---|---|
| | Reps. | Gradient | Per-task | Meta | | |
| Bai et al. [3] | | ✓ | ✓ | | iMAML | Train-validation split |
| Bernacchia [6] | | ✓ | | ✓ | MAML | Optimal step size |
| Chen et al. [9] | - | ✓ | ✓ | | MAML, BMAML | Test risk comparison |
| Huang et al. [28] | | ✓ | | ✓ | MAML | SGD solution |
| Kong et al. [32] | - | - | ✓ | | - | Effect of small data tasks |
| Saunshi et al. [42] | ✓ | | ✓ | | - | Train-validation split |
| Sun et al. [45] | ✓ | | | ✓ | - | Optimal representation |
| Ours | | ✓ | | ✓ | MAML, iMAML | Benign overfitting |

Compared to the most relevant works, our work is different in the following aspects. Compared to the works that also analyze generalization error or sample complexity in linear meta learning models such as [15, 3, 9], we focus on the overparameterized case when the total number of training data is smaller than the dimension of the model parameter. Compared to the work that focus on *representation-based* meta learning with a bilinear structure [45], we consider initialization-based meta learning methods with a bilevel structure such as MAML and iMAML. Furthermore, we provide tight analysis of the excess risk with explicit consideration of the benign overfitting condition.

A summary of key differences compared to prior art is provided in Table 1. We distinguish two different overparameterization settings: i) the *per-task level overparameterization* where the dimension of model parameter is larger than the number of training data per task, but smaller than the total number of data across all tasks; and, ii) the *meta level overparameterization* where the dimension of model parameter is larger than the total number of training data from all tasks.

## 1.2 This work

This paper provides a unifying analysis of the generalization performance for meta learning problems with overparameterized meta linear models. To our best knowledge, this is the first work that provides the condition for benign overfitting in gradient-based meta learning including MAML and iMAML.

**Technical challenges.** Before we introduce the key result of our paper, we first highlight the challenges of analyzing the generalization of gradient-based meta learning and characterizing its benign overfitting condition, compared to the non-bilevel setting such as in [5, 46, 45].

**T1)** Due to the bilevel structure of gradient-based meta learning, the solution to the meta training objective involves high order terms of data covariance. As a result, the dominating term in the excess risk propagated from the label noise contains higher order terms, which is harder to quantify and can potentially lead to orders of magnitude higher excess risk than the linear regression case [5, 46, 45].

**T2)** The existing analysis of benign overfitting in single-level problems [5, 46] has a solution that is directly related to the data covariance matrix. However, due to the nested structure of gradient-based meta learning and thus the solution matrix, the solution matrix is a function of both the data covariance matrix and the hyperparameters such as the step size. Therefore, what kind of data matrices can satisfy the benign overfitting condition cannot be directly implied.

**T3)** Due to the multi-task learning nature of meta learning, the excess risk of MAML depends on the heterogeneity across different tasks in terms of both the task data covariance and the ground truth task parameter. As a result, the data covariance matrices from different tasks have different eigenvectors. This is in contrast to the linear regression case where all the data follow the same distribution.

**Contributions.** In view of challenges, our contributions can be summarized as follows.

**C1)** Focusing on the relatively tractable linear models, we derive the excess risk for the minimum-norm solution to overparameterized gradient-based meta learning including MAML and iMAML. Specifically, the excess risk upper bound adopts the following form

$$\text{Cross-task variance} + \text{Per-task variance} + \text{Bias}$$

where the *cross-task variance* quantifies the error caused by finite task number and the variation of the ground truth task specific parameter, which is a unique term compared to

single task learning. The *bias* quantifies the bias resulting from the minimum-norm solution. And the *per-task variance* quantifies the error caused by noise in the training data.

**C2)** We compare the benign overfitting condition for the overparameterized gradient-based meta learning models and that for the empirical risk minimization (ERM) which learns a single shared parameter for all tasks. We show that overfitting is more likely to happen in MAML and its variants such as implicit MAML than in ERM. In addition, larger data heterogeneity across tasks will make overfitting more likely to happen.

**C3)** We discuss the choice of hyperparameter, e.g., the step size in MAML and the weight of the regularizer in iMAML, such that if the data leads to benign overfitting in ERM, it also leads to benign overfitting in MAML and iMAML. We show that a negative step size can preserve benign overfitting in MAML. This is complementary to the recent discovery that the optimal step size of overparameterized MAML during training is negative [6].

## 2 Problem Formulation and Methods

In this section, we will introduce the problem setup and the considered meta learning methods.

**Problem setup.** In the meta-learning setting, assume task $m$ is drawn from a task distribution, i.e. $m \sim \mathcal{M}$. For each task $m$, we observe $N$ samples with input feature $x_m \in \mathcal{X}_m \subset \mathbb{R}^d$ and target label $y_m \in \mathcal{Y}_m \subset \mathbb{R}$ drawn i.i.d. from a task-specific data distribution $\mathcal{P}_m$. These samples are collected in the dataset $\mathcal{D}_m = \{(x_{m,n}, y_{m,n})\}_{n=1}^{N}$, which is divided into the train and validation datasets, denoted as $\mathcal{D}_m^{\mathrm{tr}}$ and $\mathcal{D}_m^{\mathrm{va}}$. And $|\mathcal{D}_m^{\mathrm{tr}}| = N_{\mathrm{tr}}$ and $|\mathcal{D}_m^{\mathrm{va}}| = N_{\mathrm{va}}$ with $N = N_{\mathrm{tr}} + N_{\mathrm{va}}$. We use the empirical loss $\ell_m(\theta_m, \mathcal{D}_m)$ of per-task parameter $\theta_m \in \Theta_m$ as a measure of the performance. In this paper, we consider regression problems, where $\ell_m$ is defined as the mean squared error.

The goal for gradient-based meta learning methods, such as MAML [21] and iMAML [39], is to learn an initial parameter $\theta_0 \in \Theta_0$, which, with an adaptation method $\mathcal{A} : \Theta_0 \times (\mathcal{X}_m \times \mathcal{Y}_m)^{N_{\mathrm{tr}}} \to \Theta_m$, can generate a per-task parameter $\theta_m$ that performs well on the validation data for task $m$. Given $M$ tasks, our meta-learning objective is computed as the average of the per-task objective, given by

$$\text{Meta training objective} \qquad \mathcal{L}^{\mathcal{A}}(\theta_0, \mathcal{D}) := \frac{1}{M} \sum_{m=1}^{M} \ell_m(\mathcal{A}(\theta_0, \mathcal{D}_m^{\mathrm{tr}}), \mathcal{D}_m^{\mathrm{va}}). \qquad (1)$$

Obtaining the empirical solution $\hat{\theta}_0^{\mathcal{A}}$ by minimizing (1) under a meta learning method $\mathcal{A}$, in the meta testing stage, we evaluate $\hat{\theta}_0^{\mathcal{A}}$ on the population risk, given by

$$\text{Meta testing objective} \qquad \mathcal{R}^{\mathcal{A}}(\hat{\theta}_0^{\mathcal{A}}) := \mathbb{E}_m \left[ \mathbb{E}_{\mathcal{D}_m} \left[ \ell_m(\mathcal{A}(\hat{\theta}_0^{\mathcal{A}}, \mathcal{D}_m^{\mathrm{tr}}), \mathcal{D}_m^{\mathrm{va}}) \right] \right]. \qquad (2)$$

**Methods.** We focus on understanding the generalization of two representative gradient-based meta learning methods MAML [21] and iMAML [39] in the overparameterized regime. MAML obtains the task-specific parameter $\hat{\theta}_m(\theta_0)$ by taking one step gradient descent with step size $\alpha$ of the per-task loss function $\ell_m$ from the initial parameter $\theta_0$, that is

$$\mathcal{A}(\theta_0, \mathcal{D}_m^{\mathrm{tr}}) = \theta_0 - \alpha \nabla_{\theta_0} \ell_m(\theta_0, \mathcal{D}_m^{\mathrm{tr}}). \qquad (3)$$

On the other hand, iMAML obtains the task-specific parameter $\hat{\theta}_m$ from the initial parameter $\theta_0$ by optimizing the task-specific loss regularized by the distance between $\hat{\theta}_m$ and $\theta_0$, that is

$$\mathcal{A}(\theta_0, \mathcal{D}_m^{\mathrm{tr}}) = \arg\min_{\theta} \ \ell_m(\theta, \mathcal{D}_m^{\mathrm{tr}}) + \frac{\gamma}{2} \|\theta - \theta_0\|^2 \qquad (4)$$

where $\gamma > 0$ is the weight of the regularizer. As summarized in Figure 2, MAML has smaller computation complexity than iMAML since iMAML requires solving an inner problem during adaptation, while iMAML may achieve smaller test error since it explicitly minimize the loss.

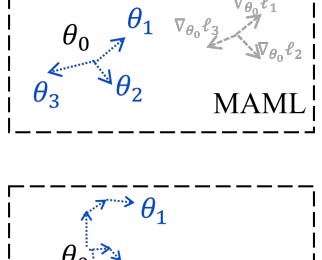

Figure 2: Two types of meta learning.

# 3 Main Results: Benign Overfitting for Gradient-based Meta Learning

In this section, we introduce the meta linear regression model and some necessary assumptions for the analysis. We present the main results, highlight the key steps of the proof and conduct simulations to verify our results. Due to space limitations, we will defer the proofs to the supplementary document.

## 3.1 Meta linear regression setting

To make a precise analysis, we will assume the following linear data model. Denoting the ground truth parameter on task $m$ as $\theta_m^\star \in \mathbb{R}^d$, and the noise as $\epsilon_m$, we assume the data model for task $m$ is

$$y_m = \theta_m^{\star\top} x_m + \epsilon_m. \tag{5}$$

Given the linear model (5), the meta training problem (1) with adaptation method (3) or (4) generally have unique solutions when $d \leq NM$. However, when the meta model $\theta_0$ and thus the per-task model $\theta_m$ are overparameterized, i.e. $d > NM$, the training problem (1) may have multiple solutions. In the subsequent analysis, we will analyze the performance of the *minimum norm* solution because recent advances in training overparameterized models reveal that gradient descent-based methods converge to the minimum norm solution [24, 35]. We provide a formal definition below.

**Definition 1** (Minimum norm solution). *Denote* $\mathbf{X}_m^{\mathrm{va}} := [x_{m,1}, \ldots, x_{m,N_{\mathrm{va}}}]^\top \in \mathbb{R}^{N_{\mathrm{va}} \times d}$, $\mathbf{y}_m^{\mathrm{va}} := [y_{m,1}, \ldots, y_{m,N_{\mathrm{va}}}]^\top \in \mathbb{R}^{N_{\mathrm{va}}}$. *With* $\mathcal{A}(\theta, \mathcal{D}_m^{\mathrm{tr}})$ *being either* (3) *or* (4)*, the minimum norm solution to the meta training problem* (1) *under the linear regression loss is expressed by*

$$\min_{\theta_0} \|\theta_0\|^2 \quad \text{s.t.} \quad \theta_0 \in \arg\min_\theta \sum_{m=1}^M \left\| \mathbf{X}_m^{\mathrm{va}} \mathcal{A}(\theta, \mathcal{D}_m^{\mathrm{tr}}) - \mathbf{y}_m^{\mathrm{va}} \right\|^2. \tag{6}$$

In our analysis, we make the following basic assumptions.

**Assumption 1** (Overparameterized model). *The total number of meta training data is smaller than the dimension of the model parameter; i.e.* $NM < d$.

**Assumption 2** (SubGaussian data). *The noise* $\epsilon_m$ *is subGaussian with* $\mathbb{E}[\epsilon_m] = 0$ *and* $\mathbb{E}[\epsilon_m^2] = \sigma^2$. *For the $m$-th task, data* $x_m = \mathbf{V}_m \mathbf{\Lambda}_m^{\frac{1}{2}} \mathbf{z}_m$*, where* $\mathbf{z}_m$ *has centered, independent, $\sigma_x$-subGaussian entries;* $\mathbb{E}[\mathbf{z}_m] = \mathbf{0}, \mathbb{E}[\mathbf{z}_m \mathbf{z}_m^\top] = \mathbf{I}_d$*, with* $\mathbf{I}_d$ *being a $d \times d$ identity matrix.*

**Assumption 3** (Data covariance matrix). *1) Assume for all* $m \in [M], i \in [d], \lambda_{m,i} > 0$, $\mathrm{Tr}(\mathbf{\Lambda}_m), \mathrm{Tr}(\mathbf{\Lambda})$ *are bounded, i.e. for all* $m \in [M]$, $\mathrm{Tr}(\mathbf{\Lambda}_m) \leq c_\lambda$. *2) Cross-task data heterogeneity* $\mathbb{V}(\{\mathbf{Q}_m\}_{m=1}^M) := \max_{i,m} |(\lambda_i - \lambda_{m,i})/\lambda_i|$ *is bounded above and below.*

**Assumption 4** (Task parameter). *The ground truth parameter* $\theta_m^\star$ *is independent of* $\mathbf{X}_m$ *and satisfies* $\mathrm{Cov}[\theta_m^\star] = (R^2/d)\mathbf{I}_d$*, where $R$ is a constant, and the entries of* $\theta_m^\star$ *are i.i.d.* $\mathcal{O}(R/\sqrt{d})$*-subGaussian.*

Assumption 1 defines the setting that the meta level is overparameterized, which has also been used in [45]. Note that Assumptions 2-4 are common in the analysis of meta learning in [15, 3, 9, 23].

With the linear data model (5), the (minimum norm) solutions to the meta training objective (1) and the meta testing objective (2) can be computed analytically which we will summarize next.

**Proposition 1. (Empirical and population level solutions)** *Under the meta linear regression model* (5)*, the meta testing objective of method* $\mathcal{A}$ *in* (2) *can be equivalently written as*

$$\mathcal{R}^{\mathcal{A}}(\theta_0) = \mathbb{E}_m \left[ \|\theta_0 - \theta_m^\star\|_{\mathbf{W}_m^{\mathcal{A}}}^2 \right] \tag{7}$$

*where the matrix* $\mathbf{W}_m^{\mathcal{A}}$ *and its empirical version* $\hat{\mathbf{W}}_m^{\mathcal{A}}$ *are given in Table 2 with* $\hat{\mathbf{Q}}_m^{\mathrm{al}} := \frac{1}{N} \mathbf{X}_m^{\mathrm{al}\top} \mathbf{X}_m^{\mathrm{al}}$. *The optimal solutions to the meta-test risk and the minimum-norm solutions to the empirical meta training loss are given below respectively*

$$\theta_0^{\mathcal{A}} := \arg\min_{\theta_0} \mathcal{R}^{\mathcal{A}}(\theta_0) = \mathbb{E}_m \left[ \mathbf{W}_m^{\mathcal{A}} \right]^{-1} \mathbb{E}_m \left[ \mathbf{W}_m^{\mathcal{A}} \theta_m^\star \right] \tag{8a}$$

$$\hat{\theta}_0^{\mathcal{A}} := \arg\min_{\theta_0} \mathcal{L}^{\mathcal{A}}(\theta_0, \mathcal{D}) = \left( \sum_{m=1}^M \hat{\mathbf{W}}_m^{\mathcal{A}} \right)^\dagger \left( \sum_{m=1}^M \hat{\mathbf{W}}_m^{\mathcal{A}} \theta_m^\star \right) + \Delta_M^{\mathcal{A}} \tag{8b}$$

*where* $^\dagger$ *denotes the Moore-Penrose pseudo inverse;* $\Delta_M^{\mathcal{A}}$ *is an error term that depends on* $\mathbf{X}_m, \epsilon_m$, *and specified in the supplementary document.*

To study overfitting in the meta learning model, we quantify its generalization ability via the widely used metric - *excess risk*. The excess risk of method $\mathcal{A}$ (which can be "ma" for MAML and "im" for iMAML), with an empirical solution $\hat{\theta}_0^{\mathcal{A}}$ and population solution $\theta_0^{\mathcal{A}}$, is defined as

$$\mathcal{E}^{\mathcal{A}}(\hat{\theta}_0^{\mathcal{A}}) := \mathcal{R}^{\mathcal{A}}(\hat{\theta}_0^{\mathcal{A}}) - \mathcal{R}^{\mathcal{A}}(\theta_0^{\mathcal{A}}). \tag{9}$$

Table 2: Weight matrices under different method $\mathcal{A}$.

| Method | Weight matrices |
|--------|-----------------|
| ERM | $\mathbf{W}_m^{\mathrm{er}} = \mathbf{Q}_m$ |
| | $\hat{\mathbf{W}}_m^{\mathrm{er}} = \hat{\mathbf{Q}}_m$ |
| MAML | $\mathbf{W}_m^{\mathrm{ma}} = (\mathbf{I} - \alpha\mathbf{Q}_m)\mathbf{Q}_m(\mathbf{I} - \alpha\mathbf{Q}_m)$ |
| | $\hat{\mathbf{W}}_m^{\mathrm{ma}} = (\mathbf{I} - \alpha\hat{\mathbf{Q}}_m^{\mathrm{tr}})\hat{\mathbf{Q}}_m^{\mathrm{va}}(\mathbf{I} - \alpha\hat{\mathbf{Q}}_m^{\mathrm{tr}})$ |
| iMAML | $\mathbf{W}_m^{\mathrm{im}} = (\gamma^{-1}\mathbf{Q}_m + \mathbf{I})^{-1}\mathbf{Q}_m(\gamma^{-1}\mathbf{Q}_m + \mathbf{I})^{-1}$ |
| | $\hat{\mathbf{W}}_m^{\mathrm{im}} = (\gamma^{-1}\hat{\mathbf{Q}}_m^{\mathrm{tr}} + \mathbf{I})^{-1}\hat{\mathbf{Q}}_m^{\mathrm{va}}(\gamma^{-1}\hat{\mathbf{Q}}_m^{\mathrm{tr}} + \mathbf{I})^{-1}$ |

In (9), the excess risk measures the difference between the population risk of the empirical solution, $\hat{\theta}_0$ and the optimal population risk. Given total number of training samples $MN$, if $d \to \infty$, the classic learning theory implies that the excess risk $\mathcal{E}^{\mathcal{A}}(\hat{\theta}_0^{\mathcal{A}})$ also grows, which leads to overfitting [25]. The larger the excess risk, the further the empirical solution $\hat{\theta}_0^{\mathcal{A}}$ is from the optimal population solution $\theta_0^{\mathcal{A}}$, indicating more severe *overfitting*.

## 3.2 Main results

With the closed-form solutions given in Proposition 1, we are ready to bound the excess risk of MAML and iMAML in the overparameterized linear regime. For notation brevity, we first introduce some universal constants such as $c_0, c_1, c_2, \ldots$, and only present the dominating terms in the subsequent results. The precise presentation of remaining terms are deferred to the supplementary document.

We first decompose the excess risk into three terms in Proposition 2.

**Proposition 2.** *Define* $\mathbf{W}^{\mathcal{A}} := \mathbb{E}_m[\mathbf{W}_m^{\mathcal{A}}]$. *The excess risk of a meta learning method* $\mathcal{A}$ *can be bounded by*

$$\mathcal{E}^{\mathcal{A}}(\hat{\theta}_0^{\mathcal{A}}) \lesssim \mathcal{E}_{\theta_m^\star} + \mathcal{E}_{\epsilon_m} + \mathcal{E}_b \tag{10}$$

*where the first term* $\mathcal{E}_{\theta_m^\star}$ *is a function of* $\theta_m^\star, \theta_0^{\mathcal{A}}, \mathbf{W}^{\mathcal{A}}, \hat{\mathbf{W}}_m^{\mathcal{A}}$, *which quantifies the weighted variance of the ground truth task specific parameters* $\theta_m^\star$; *the second term* $\mathcal{E}_{\epsilon_m}$, *as a function of* $\epsilon_m$, *is the weighted noise variance; and the third term* $\mathcal{E}_b$, *as a function of* $\theta_0^{\mathcal{A}}, \mathbf{W}^{\mathcal{A}}, \hat{\mathbf{W}}_m^{\mathcal{A}}$, *is the bias of the minimum-norm solution in overparameterized MAML or iMAML.*

Based on this decomposition, as we will show in Section 4, the bound of the excess risk can be derived from the bound of these three terms $\mathcal{E}_{\theta_m^\star}, \mathcal{E}_{\epsilon_m^\star}, \mathcal{E}_b$, respectively, which gives Theorem 1.

**Theorem 1** (Excess risk bound). *Suppose Assumptions 1-4 hold. Let* $\mu_1(\cdot) \geq \mu_2(\cdot) \ldots$ *denote the eigenvalues of a matrix in the descending order. For the meta linear regression problem with the minimum-norm solution (6), for* $0 \leq k \leq d$, *define the effective ranks as*

$$r_k\left(\mathbf{W}^{\mathcal{A}}\right) := \frac{\sum_{i>k} \mu_i\left(\mathbf{W}^{\mathcal{A}}\right)}{\mu_{k+1}\left(\mathbf{W}^{\mathcal{A}}\right)}; \qquad R_k\left(\mathbf{W}^{\mathcal{A}}\right) := \frac{\left(\sum_{i>k} \mu_i(\mathbf{W}^{\mathcal{A}})\right)^2}{\sum_{i>k} \mu_i^2\left(\mathbf{W}^{\mathcal{A}}\right)}. \tag{11}$$

*With the cross-task data heterogeneity* $\mathbb{V}$ *defined in Assumption 3, if there exist universal constants* $c_1, c_2, c_3 > 1$ *such that the effective dimension* $k^* = \min\{k \geq 0 : r_k(\mathbf{W}^{\mathcal{A}}) \geq c_1 NM\}$, $c_2 \log(1/\delta) < NM$ *and* $k^* < NM/c_3$, *then with probability at least* $1 - \delta$, *the excess risk satisfies*

$$\mathcal{E}^{\mathcal{A}}(\hat{\theta}_0^{\mathcal{A}}) \lesssim \|\mathbb{E}[\theta_m^\star]\|^2 \bar{\lambda} \sqrt{\frac{r_0(\mathbf{W}^{\mathcal{A}})}{MN}} + \sigma^2 \left(\frac{k^*}{MN} + \frac{MN}{R_{k^*}(\mathbf{W}^{\mathcal{A}})}\right) \left(1 + \mathbb{V}(\{\mathbf{W}_m^{\mathcal{A}}\}_{m=1}^M)\right). \tag{12}$$

Theorem 1 provides the excess risk bound via the effective ranks. In (11), the effective ranks $r_k$ and $R_k$ of a matrix capture the distribution of the eigenvalues of this matrix, and the effective dimension $k^*$ determines the above upper bound by considering the asymmetry of the eigenvalues of the solution matrix. The idea is to choose $k^*$ that makes $R_{k^*}$ large enough and keeps $k^*$ small enough compared to $MN$ so that the variance term of the excess risk is controlled. For example, $r_0$ is the trace normalized by the largest eigenvalue, which is bounded above by $R_0$. And both $r_0$ and $R_0$ are no larger than the rank of the matrix, and they are equal to the rank only when all non-zero eigenvalues are equal. If the eigenvalues distribute more uniformly, the effective rank will be larger, otherwise smaller.

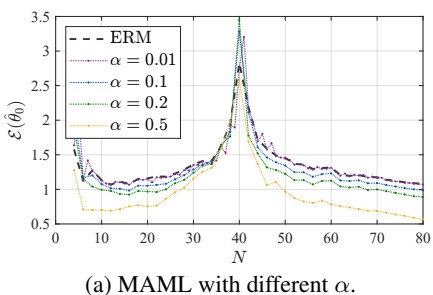
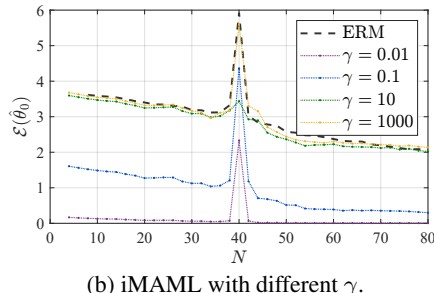

(a) MAML with different $\alpha$.      (b) iMAML with different $\gamma$.

Figure 3: Excess risk vs number of samples ($N$) with different hyperparameters ($M = 10, d = 200$).

**Remark 1.** 1) The definition of effective rank has been also given in [5] but only on the data matrix $\mathbf{Q}$. And our setting reduces to the single task ERM learning, or the linear regression case in [5], when $M = 1, \theta_m^\star = \theta_0, \mathbf{W}_m^{\mathcal{A}} = \mathbf{Q}$, which implies that the cross-task variance in (10) as well as the data heterogeneity $\mathbb{V}(\cdot)$ reduces to zero. Accordingly, Theorem 1 reduces to Theorem 4 in [5].

2) Given Theorem 1, in order to control the excess risk of solution $\hat{\theta}_0^{\mathcal{A}}$, we want $r_0(\mathbf{W}^{\mathcal{A}})$ to be small compared to the total number of training samples $MN$, but $r_{k^*}(\mathbf{W}^{\mathcal{A}})$ and $R_{k^*}(\mathbf{W}^{\mathcal{A}})$ to be large compared to $MN$. In addition, the cross-task heterogeneity $\mathbb{V}$ should be small. Since for a matrix $\mathbf{W}, r_k(\mathbf{W}) \leq R_k(\mathbf{W}) \leq d$, this suggests the model benefits from overparameterization.

Building upon Theorem 1, we now discuss the conditions for "benign overfitting", which refers to the situation that overparameterization does not "harm" the excess risk, or the excess risk still vanishes when $d > MN$ and $N, M, d$ increase.

> **Definition 2** (Condition for benign overfitting in meta learning). *The weight matrices $\mathbf{W}^{\mathcal{A}}$ for method $\mathcal{A}$ satisfy the* benign overfitting condition *in gradient-based meta learning, if and only if*
>
> $$\lim_{NM,d \to \infty} \frac{r_0(\mathbf{W}^{\mathcal{A}})}{NM} = \lim_{NM,d \to \infty} \frac{k^*}{NM} = \lim_{NM,d \to \infty} \frac{NM}{R_{k^*}(\mathbf{W}^{\mathcal{A}})} = 0. \quad (13)$$

This guarantees the excess risk (12) goes to zero in overparameterized meta learning models with sufficient training data from all tasks. To provide an intuitive explanation, Figure 3 plots the population risk versus the number of the training data, which demonstrates the "double descent" curve. Namely, as $N$ increases, $\mathcal{E}(\hat{\theta}_0)$ first decreases, then increases and then decreases again, as is discovered in overparameterized neural networks [36]. The trend in Figure 3 is similar to the trend observed in [37]. When $d/(NM) > 1$, the model is overparameterized, which can overfit the training data, leading to larger excess risk as $N$ decreases. However, Figure 3 shows the excess risk does not become too large as $N$ decreases, indicating that overfitting does not severely harm the population risk in this case.

### 3.3 Examples and discussion

In this section, we discuss how the benign overfitting condition (13) in gradient-based meta learning reduces to that in single task linear regression; e.g., in [5, 46]. We also provide examples to show

**Q1)** *how certain properties of meta training data affect the excess risk; and,*

**Q2)** *how to choose the hyperparameters that preserve benign overfitting.*

**Data covariance and cross-task heterogeneity.** Theorem 1 reveals that the excess risk depends on both the eigenvalues of the data covariance matrix $\mathbf{Q}_m$, and the cross-task data heterogeneity, measured by $\mathbb{V}(\{\mathbf{Q}_m\}_{m=1}^M)$. We give an example below to better demonstrate how these two properties of gradient-based meta training data affect the excess risk.

**Example 1** (Data covariance). *Suppose $\mathbf{Q}_m = \mathrm{diag}(\mathbf{I}_{d_1}, \beta\mathbf{I}_{d-d_1}), \forall m$. Set $M = 10, d = 200, d_1 = 20, \alpha = 0.1$ for MAML and $\gamma = 10^3$ for iMAML. Then the benign overfitting condition (13) is satisfied by MAML and iMAML. We plot the excess risk under different $\beta$ in Figure 4.*

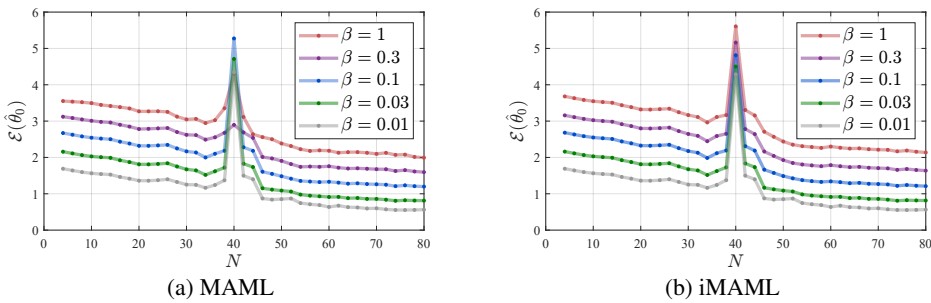

(a) MAML

(b) iMAML

Figure 4: Excess risks vs number of samples ($N$) for $\mathbf{Q}_m = \mathrm{diag}(\mathbf{I}_{d_1}, \beta\mathbf{I}_{d-d_1})$ with different $\beta$.

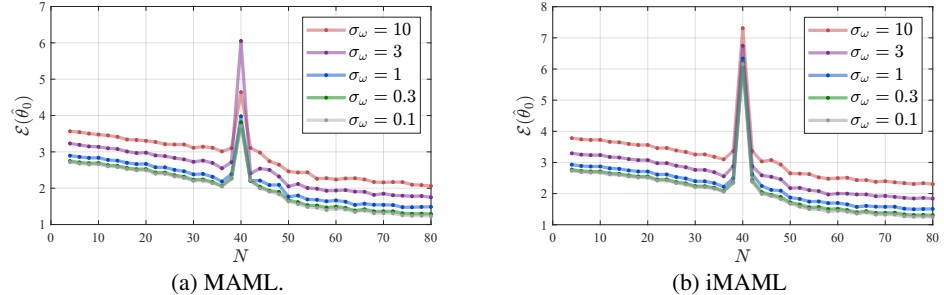

(a) MAML.

(b) iMAML

Figure 5: Excess risks of MAML and iMAML vs number of training samples ($N$) for $\mathbf{Q}_m = |1 + \omega_m| \mathrm{diag}(\mathbf{I}_{d_1}, \beta\mathbf{I}_{d-d_1}), \omega_m \sim \mathcal{N}(0, \sigma_\omega^2)$ with different $\sigma_\omega$.

From Figure 4 we can observe that given a fixed number of training data $N$, the population risk increases with $\beta$ for both MAML and iMAML. This observation verifies our theory since larger $\beta$ results in a smaller $R_k^{\mathcal{A}}(\mathbf{W}^{\mathcal{A}})$, leading to a larger upper bound on the variance term in (12).

Example 1 demonstrates how the per-task data matrix $\mathbf{Q}_m$ affects the excess risk. We consider another example that demonstrates how the data heterogeneity across tasks affects the excess risk.

**Example 2** (Data heterogeneity). *Suppose $\mathbf{Q}_m = |\omega_m + 1| \mathrm{diag}(\mathbf{I}_{d_1}, \beta\mathbf{I}_{d-d_1})$ with $\omega_m \sim \mathcal{N}(0, \sigma_\omega^2)$ for all $m$. Set $M = 10, d = 200, d_1 = 20, \beta = 0.3, \alpha = 0.1$ for MAML and $\gamma = 0.1$ for iMAML. Then it satisfies the benign overfitting condition* (13) *for MAML and iMAML. Figure 5 plots the excess risk with different choices of $\sigma_\omega$.*

Observing from Figure 5 that the larger $\sigma_\omega^2$, the higher the excess risk, and the more difficult for the benign overfitting condition to be satisfied for both MAML and iMAML. Therefore, compared to ERM with a single task, the benign overfitting condition for MAML is more restrictive as it imposes constraints for both the expected data covariance $\mathbf{Q}_m$, and the data heterogeneity $\mathbb{V}(\{\mathbf{W}_m^{\mathcal{A}}\}_{m=1}^M)$.

**Connection to multi-task ERM.** To compare benign overfitting in the gradient-based meta learning with that in the conventional ERM, where $\theta_m = \theta_0$, we can set the step size $\alpha = 0$ in MAML, or $\gamma \to \infty$ in iMAML, and $N_{\mathrm{va}} = N$, which reduces to conventional ERM without adaptation.

Compared to that of MAML and iMAML in (13), the benign overfitting condition is less restrictive for ERM since it does not impose constraints on $\alpha$ or $\gamma$. Intuitively, benign overfitting is more likely to happen in MAML or iMAML than in ERM. The hyperparameters $\alpha$ and $\gamma$ will affect the eigenvalues of $\mathbf{W}_m^{\mathrm{ma}}, \mathbf{W}_m^{\mathrm{im}}$, respectively, thus affecting their corresponding excess risk. Here we provide a sufficient condition where the benign overfitting condition in ERM is preserved in MAML or iMAML. We summarize the results in the corollary below.

**Corollary 1** (Hyperparameters that preserve benign overfitting). *Recall $\lambda_1$ is the largest eigenvalue of $\mathbf{Q}$. For MAML, when $0 < \alpha \leq \frac{1}{3\lambda_1}$, and for iMAML, when $\gamma \geq \lambda_1$, then the effective ranks of $\mathbf{W}^{\mathrm{ma}}$ and $\mathbf{W}^{\mathrm{im}}$ are bounded above and below by a positive constant times the effective rank of $\mathbf{Q}$, and therefore the benign overfitting condition holds for MAML and iMAML if it holds for ERM. To summarize, there are constants $c_1, c_2, c_3, c$ such that for $k^* = \min\{k \geq 0 : r_k(\mathbf{Q}) \geq c_1 NM\}$. For*

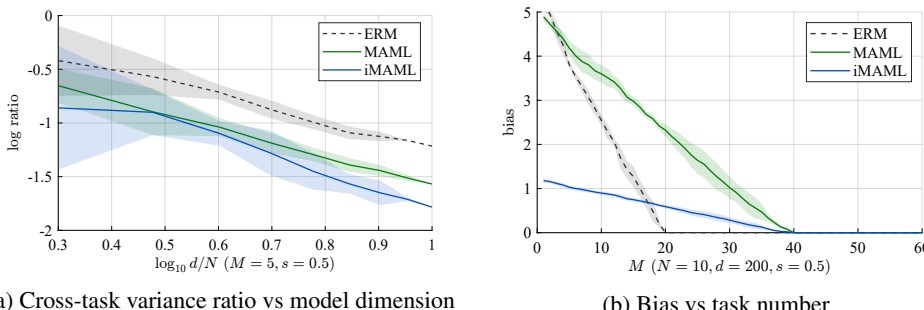

(a) Cross-task variance ratio vs model dimension  (b) Bias vs task number

Figure 6: Cross-task variance and bias versus task number to elaborate Lemma 1 and Lemma 2.

$\delta < 1$, $c_2 \log(1/\delta) < NM$ and $k^* < NM/c_3$, with probability at least $1 - 7e^{-2NM/c}$, it follows

$$\mathcal{E}^{\mathcal{A}}(\hat{\theta}_0^{\mathcal{A}}) \lesssim \|\mathbb{E}[\theta_m^\star]\|^2 \bar{\lambda} \sqrt{\frac{r_0(\mathbf{Q})}{MN}} + \sigma^2 \left( \frac{k^*}{MN} + \frac{MN}{R_{k^*}(\mathbf{Q})} \right) \left( 1 + \mathbb{V}(\{\mathbf{Q}_m\}_{m=1}^M) \right). \quad (14)$$

**Remark 2.** For MAML, let the unordered eigenvalues $\tilde{\mu}_i(\mathbf{W}^{\mathrm{ma}}) = \lambda_i(1 - \alpha\lambda_i)^2$. One challenge to control $\tilde{\mu}_i(\mathbf{W}^{\mathrm{ma}})$ is that $\tilde{\mu}_i(\mathbf{W}^{\mathrm{ma}})$ are not necessarily monotonic w.r.t. $\lambda_i$; that is, it does not necessarily hold that $\tilde{\mu}_1 \geq \tilde{\mu}_2 \geq \cdots \geq \tilde{\mu}_d$. For any $\lambda_i \geq \lambda_j$, if $\tilde{\mu}_i(\mathbf{W}^{\mathcal{A}}) \geq \tilde{\mu}_j(\mathbf{W}^{\mathcal{A}})$, then we say the order of the eigenvalues is preserved. For this to hold, it requires $\tilde{\mu}_i(\lambda_i)$ to be a monotonically non-decreasing function of $\lambda_i$, which yields $\alpha \leq \frac{1}{3\lambda_1}$. Similar results can be obtained for iMAML by controlling the value of $\gamma$. And the bound on $\alpha$ or $\gamma$ further ensures that $\tilde{\mu}_i(\mathbf{W}^{\mathcal{A}})$ is bounded above and below by a positive constant times the effective rank of $\mathbf{Q}$.

## 4 Proof Outline

In this section, we highlight the key steps of the proof for Theorem 1. We achieve so by analyzing the three terms in Proposition 2 respectively.

The first two terms in (10) can be bounded based on the concentration inequalities on subGaussian variables, given in Lemmas 1 and 2.

**Lemma 1** (Bound on cross-task variance). *With probability at least $1 - \delta$, it follows*

$$\mathcal{E}_{\theta_m^\star} = \left\| \left( \sum_{m=1}^M \hat{\mathbf{W}}_m^{\mathcal{A}} \right)^\dagger \left( \sum_{m=1}^M \hat{\mathbf{W}}_m^{\mathcal{A}}(\theta_m^\star - \theta_0^{\mathcal{A}}) \right) \right\|_{\mathbf{W}^{\mathcal{A}}}^2 \leq \tilde{\mathcal{O}}\left( \frac{N}{d} \right) \mathcal{E}_{\epsilon_m} \quad (15)$$

*where $\widetilde{O}(\cdot)$ hides the log polynomial dependence on $N, M, d$.*

The cross-task variance term analzyed in Lemma 1 is unique in meta learning, which captures the data heterogeneity across different tasks. To elaborate Lemma 1, we plot the cross-task variance versus the task number in Figure 6a with task number $M = 5$, training validation split parameter $s = N_{\mathrm{tr}}/N = 0.5$, per-task data number $N = 10$. This figure demonstrates that the ratio of cross-task variance and per-task variance decreases with $d/N$, which is consistent with Lemma 1.

**Lemma 2** (Bound on bias). *For any $1 < \log(1/\delta) < MN_{\mathrm{va}}$, with probability at least $1 - \delta$, we have*

$$\mathcal{E}_b \lesssim \|\theta_0^{\mathcal{A}}\|^2 \|\mathbf{W}^{\mathcal{A}}\| \max \left\{ \sqrt{\frac{r_0(\mathbf{W}^{\mathcal{A}})}{MN_{\mathrm{va}}}}, \frac{r_0(\mathbf{W}^{\mathcal{A}})}{MN_{\mathrm{va}}}, \sqrt{\frac{\log(1/\delta)}{MN_{\mathrm{va}}}} \right\}. \quad (16)$$

This term is similar to the bias term in the linear regression case, but directly depending on the solution matrix $\mathbf{W}$ instead of the data matrix $\mathbf{Q}$. To elaborate Lemma 2, Figure 6b demonstrates that the bias term decays with $M$ until it reaches zero when the model is underparameterized. These two terms in Lemma 1 and Lemma 2 do not go to infinity as $N, M, d$ increase.

Note that, the key step is the bound on $\mathcal{E}_{\epsilon_m}$, which is the dominating term in the decomposition of excess risk (10) in the overparameterized regime. We will bound it below.

**Lemma 3** (Bound on per-task variance). *There exist constants $c_1, c_2, c_3$ such that for $0 \leq k \leq 2NM/c_1$, $r_k(\mathbf{W}^{\mathcal{A}}) \geq c_2 NM$, and $k_0 \leq k$, with probability at least $1 - 7e^{-2NM/c_3}$, it follows*

$$\mathcal{E}_{\epsilon_m} \lesssim \left( \frac{k_0}{MN_{\text{va}}} + \frac{MN_{\text{va}}}{R_{k_0}(\mathbf{W}^{\mathcal{A}})} \right) \left( 1 + \mathbb{V}(\{\mathbf{W}_m^{\mathcal{A}}\}_{m=1}^M) \right). \tag{17}$$

Note that, in the single task linear regression case, the there is no cross-task data heterogeneity, i.e., $\mathbb{V} = 0$. This term is unique in the meta learning setting with multiple tasks. Plugging the results of Lemmas 1, 2 and 3 into (10), we will reach Theorem 1.

## 5 Conclusions and Limitations

This paper studies the generalization performance of the gradient-based meta learning with an overparameterized model. For a precise analysis, we focus on linear models where the total number of data from all tasks is smaller than the dimension of the model parameter. We show that when the data heterogeneity across tasks is relatively small, the per-task data covariance matrices with certain properties lead to benign overfitting for gradient-based meta learning with the minimum-norm solution. This explains why overparameterized meta learning models can generalize well in new data and new tasks. Furthermore, our theory shows that overfitting is more likely to happen in meta learning than in ERM, especially when the data heterogeneity across tasks is relatively high.

One limitation of this work is that the analysis focuses on the meta linear regression case. While this analysis can capture practical cases where we reuse the feature extractor from pre-trained models and only meta-train the parameters in the last linear layer, it is also promising to extend our analysis to nonlinear cases via means of random features and neural tangent kernels in the future work.

## Acknowledgments

This work was partially supported by National Science Foundation MoDL-SCALE Grant 2134168 and the Rensselaer-IBM AI Research Collaboration (`http://airc.rpi.edu`), part of the IBM AI Horizons Network (`http://ibm.biz/AIHorizons`).

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
