# Supplementary Material

In this supplementary document, we present the missing derivations of some claims, as well as the proofs of all the lemmas and theorems in the paper.

# Table of Contents

# A    Notations

We use $[\mathbf{X}_m]$ to represent row stack of matrices $\mathbf{X}_m$ with indices $m$, i.e.

$$[\mathbf{X}_m] = \left[\mathbf{X}_1^\top, \mathbf{X}_2^\top, \ldots, \mathbf{X}_M^\top\right]^\top.$$

For a given square matrix $\mathbf{D}_m$, define

$$\mathrm{diag}[\mathbf{D}_m] = \begin{bmatrix} \mathbf{D}_1 & \mathbf{0} & \ldots & \mathbf{0} \\ \mathbf{0} & \mathbf{D}_2 & & \vdots \\ \vdots & & \ddots & \mathbf{0} \\ \mathbf{0} & \ldots & \mathbf{0} & \mathbf{D}_M \end{bmatrix}.$$

We use $\mu_i(\cdot)$ to denote the $i$-th eigenvalue of a matrix with descending order, $\|\cdot\|$ to denote the operator norm, and $\|\cdot\|_\mathrm{F}$ to denote the Frobenious norm.

For any matrix $\mathbf{M} \in \mathbb{R}^{n\times d}$, denote $\mathbf{M}_{0:k}$ to be the matrix which is comprised of the first $k$ columns of $\mathbf{M}$, and $\mathbf{M}_{k:d}$ to be the matrix comprised of the rest of the columns of $\mathbf{M}$. For any vector $\eta \in \mathbb{R}^d$ denote $\eta_{0:k}$ to be the vector comprised of the first $k$ components of $\eta$, and $\eta_{k:\infty}$ to be the vector comprised of the rest of the coordinates of $\eta$. Denote $\mathbf{\Lambda}_{0:k} = \mathrm{diag}(\lambda_1, \ldots, \lambda_k)$, and $\mathbf{\Lambda}_{k:\infty} = \mathrm{diag}(\lambda_{k+1}, \lambda_{k+2}, \ldots)$, $\mathbf{\Lambda}_{k:d} = \mathrm{diag}(\lambda_{k+1}, \lambda_{k+2}, \ldots \lambda_d)$.

For $t \geq 0$, $N \in \mathbb{Z}^+$, define $c_{r_0}(r_0(\mathbf{\Lambda}), N, t) := \max\left\{\sqrt{\frac{r_0(\mathbf{\Lambda})}{N}}, \frac{r_0(\mathbf{\Lambda})}{N}, \sqrt{\frac{t}{N}}, \frac{t}{N}\right\}$.

We use $\mathbb{E}[\cdot]$ to denote expectation and $\mathrm{Cov}[\cdot]$ to denote covariance.

We use superscript "ma" and "im" to represent quantities related to the MAML and iMAML algorithms, respectively. For notation simplicity, we omit the superscript $\mathcal{A}$ when the arguments hold for both MAML and iMAML.

# B  Proof of Proposition 1

**Proposition 3** (Empirical and population level solutions). *Under the data model* (5), *the meta-test risk of method $\mathcal{A}$ defined in* (2) *can be computed by*

$$\mathcal{R}^{\mathcal{A}}(\theta_0) = \mathbb{E}_m\big[\|\theta_0 - \theta_m^\star\|_{\mathbf{W}_m^{\mathcal{A}}}^2\big] + c.$$

*The optimal solutions to the meta-test risk and the minimum-norm solution are given below respectively*

$$\theta_0^{\mathcal{A}} := \arg\min_{\theta_0} \mathcal{R}^{\mathcal{A}}(\theta_0) = \mathbb{E}_m\big[\mathbf{W}_m^{\mathcal{A}}\big]^{-1}\mathbb{E}_m\big[\mathbf{W}_m^{\mathcal{A}}\theta_m^\star\big] \tag{18a}$$

$$\hat{\theta}_0^{\mathcal{A}} := \arg\min_{\theta_0} \mathcal{L}^{\mathcal{A}}(\theta_0, \mathcal{D}) = \Big(\sum_{m=1}^M \hat{\mathbf{W}}_m^{\mathcal{A}}\Big)^\dagger \Big(\sum_{m=1}^M \hat{\mathbf{W}}_m^{\mathcal{A}}\theta_m^\star\Big) + \Delta_M^{\mathcal{A}} \tag{18b}$$

*where $^\dagger$ denotes the Moore–Penrose pseudo inverse, the error term $\Delta_M^{\mathcal{A}}$ is a polynomial function of $M, N, d$, which will be specified in the following sections for MAML and iMAML. And $\hat{\mathbf{Q}}_m^{\mathrm{al}} := \frac{1}{N}\mathbf{X}_m^{\mathrm{al}\top}\mathbf{X}_m^{\mathrm{al}}$. The weight matrices of different methods, $\mathbf{W}_m^{\mathcal{A}}$ and $\hat{\mathbf{W}}_m^{\mathcal{A}}$, are given in Table 2.*

## B.1  Model agnostic meta learning method

Without loss of generality, assume $\sigma = 1$ to simplify notation. We use meta-test risk $\mathcal{R}_N^{\mathcal{A}}$ to represent expected test risk with finite number of adaptation data $N$ during testing, which is slightly different compared to population risk $\mathcal{R}^{\mathcal{A}} = \lim_{N\to\infty} \mathcal{R}_N^{\mathcal{A}}$. The MAML meta-test risk is defined as [23]

$$\mathcal{R}_N^{\mathrm{ma}}(\theta_0) := \mathbb{E}\left[\Big(y_m - \hat{\theta}_m^{\mathrm{ma}}(\theta_0, \mathcal{D}_{m,N})^\top x_m\Big)^2\right]$$

$$= \mathbb{E}_m\left[\|\theta_0 - \theta_m^\star\|_{\mathbf{W}_{m,N}^{\mathrm{ma}}}^2\right] + 1 + \frac{\alpha^2}{N}\mathbb{E}_m[\mathrm{Tr}(\mathbf{Q}_m^2)] \tag{19}$$

where the matrix is defined as

$$\mathbf{W}_{m,N}^{\mathrm{ma}} = \mathbb{E}_{\hat{\mathbf{Q}}_m}\left[(\mathbf{I} - \alpha\hat{\mathbf{Q}}_m)\mathbf{Q}_m(\mathbf{I} - \alpha\hat{\mathbf{Q}}_m)\right]$$

$$= (\mathbf{I} - \alpha\mathbf{Q}_m)\mathbf{Q}_m(\mathbf{I} - \alpha\mathbf{Q}_m) + \frac{\alpha^2}{N}\left(\mathbb{E}_{x_{m,i}}\big[x_{m,i}x_{m,i}^\top\mathbf{Q}_m x_{m,i}x_{m,i}^\top\big] - \mathbf{Q}_m^3\right). \tag{20}$$

Assume during meta testing, we have infinite adaptation data, i.e., $N \to \infty$, then the optimal population risk of MAML is

$$\mathcal{R}^{\mathrm{ma}}(\theta_0) = \lim_{N\to\infty} \mathcal{R}_N^{\mathrm{ma}}(\theta_0) = \mathbb{E}_m\left[\|\theta_0 - \theta_m^\star\|_{\mathbf{W}_m^{\mathrm{ma}}}^2\right] + 1. \tag{21}$$

In MAML, define $\theta_0^{\mathrm{ma}}$ as the minimizer of the optimal population risk of MAML, given by

$$\theta_0^{\mathrm{ma}} = \arg\min_{\theta_0} \mathcal{R}^{\mathrm{ma}}(\theta_0) = \arg\min_{\theta_0} \mathbb{E}_m\big[\|\theta_0 - \theta_m^\star\|_{\mathbf{W}_m^{\mathrm{ma}}}^2\big] = \mathbb{E}_m\big[\mathbf{W}_m^{\mathrm{ma}}\big]^{-1}\mathbb{E}_m\big[\mathbf{W}_m^{\mathrm{ma}}\theta_m^\star\big]. \tag{22}$$

Using the optimality condition of $\mathcal{L}^{\mathrm{ma}}(\theta_0, \mathcal{D})$ given in (1) , we have

$$\hat{\theta}_0^{\mathrm{ma}} = \Big(\sum_{m=1}^M \hat{\mathbf{W}}_m^{\mathrm{ma}}\Big)^\dagger \Big(\sum_{m=1}^M \hat{\mathbf{W}}_m^{\mathrm{ma}}\theta_m^\star + \big(\mathbf{I} - \alpha\hat{\mathbf{Q}}_m^{\mathrm{tr}}\big)\big(\frac{1}{N_{\mathrm{va}}}\mathbf{X}_m^{\mathrm{va}\top}\mathbf{e}_m^{\mathrm{va}} - \frac{\alpha}{N_{\mathrm{tr}}}\hat{\mathbf{Q}}_m^{\mathrm{va}}\mathbf{X}_m^{\mathrm{tr}\top}\mathbf{e}_m^{\mathrm{tr}}\big)\Big) \tag{23a}$$

$$\hat{\mathbf{W}}_m^{\mathrm{ma}} = (\mathbf{I} - \alpha\hat{\mathbf{Q}}_m^{\mathrm{tr}})\hat{\mathbf{Q}}_m^{\mathrm{va}}(\mathbf{I} - \alpha\hat{\mathbf{Q}}_m^{\mathrm{tr}}). \tag{23b}$$

Therefore, we can arrive at (8b) by defining

$$\Delta_M^{\mathrm{ma}} := \Big(\sum_{m=1}^M \hat{\mathbf{W}}_m^{\mathrm{ma}}\Big)^\dagger \Big(\sum_{m=1}^M (\mathbf{I} - \alpha\hat{\mathbf{Q}}_m^{\mathrm{tr}})\frac{1}{N_2}\mathbf{X}_m^{\mathrm{va}\top}\mathbf{e}_m^{\mathrm{va}} - (\mathbf{I} - \alpha\hat{\mathbf{Q}}_m^{\mathrm{tr}})\hat{\mathbf{Q}}_m^{\mathrm{va}}\frac{\alpha}{N_{\mathrm{tr}}}\mathbf{X}_m^{\mathrm{tr}\top}\mathbf{e}_m^{\mathrm{tr}}\Big). \tag{24}$$

## B.2 Implicit model agnostic meta learning method

For the iMAML method, the task-specific parameter $\hat{\theta}_m^{\text{im}}$ is computed from the initial parameter $\theta_0$ by optimizing the regularized task-specific empirical loss, given by

$$\hat{\theta}_m^{\text{im}}(\theta_0, \mathcal{D}_m) = \arg\min_{\theta_m} \frac{1}{N} \|\mathbf{y}_m - \mathbf{X}_m \theta_m\|^2 + \gamma \|\theta_m - \theta_0\|^2 \tag{25}$$

where $\gamma$ is the weight of the regularizer, and $\mathcal{D}_m$ is the adaptation data during meta-testing or training data during meta-training.

The estimated task-specific parameter can be computed by

$$\hat{\theta}_m^{\text{im}}(\theta_0, \mathcal{D}_m) = (\hat{\mathbf{Q}}_m^{\text{al}} + \gamma \mathbf{I})^{-1} \Big( \frac{1}{N} \mathbf{X}_m^\top \mathbf{y}_m + \gamma \theta_0 \Big). \tag{26}$$

The empirical loss of iMAML is defined as the average per-task loss, given by

$$\mathcal{L}_{M,N}^{\text{im}}(\theta_0, \mathcal{D}) = \frac{1}{M N_{\text{va}}} \sum_{m=1}^{M} \left\| \mathbf{y}_m^{\text{va}} - \mathbf{X}_m^{\text{va}} \hat{\theta}_m^{\text{im}}(\theta_0, \mathcal{D}_m^{\text{tr}}) \right\|^2 \tag{27}$$

whose minimizer is

$$\hat{\theta}_0^{\text{im}} = \arg\min_{\theta_0} \frac{1}{M N_{\text{va}}} \sum_{m=1}^{M} \left\| \mathbf{X}_{m,N}^{\text{va}} \theta_m^\star + \mathbf{e}_{m,N_{\text{va}}}^{\text{val}} - \mathbf{X}_m^{\text{va}} \hat{\theta}_m^{\text{im}}(\theta_0, \mathcal{D}_m^{\text{tr}}) \right\|^2. \tag{28}$$

Using the optimality condition of the above problem, we obtain

$$\hat{\theta}_0^{\text{im}} = \Big( \sum_{m=1}^{M} \hat{\mathbf{W}}_m^{\text{im}} \Big)^\dagger \Big( \sum_{m=1}^{M} \hat{\mathbf{W}}_m^{\text{im}} \theta_m^\star \Big) + \Delta_M^{\text{im}} \tag{29a}$$

$$\text{with} \quad \Delta_M^{\text{im}} = \Big( \sum_{m=1}^{M} \hat{\mathbf{W}}_m^{\text{im}} \Big)^\dagger \Big( \sum_{m=1}^{M} \gamma \Sigma_{\theta_m} \frac{1}{N_{\text{va}}} \mathbf{X}_m^{\text{va}\top} \mathbf{e}_{m,N}^{\text{va}} - \gamma^{-1} \hat{\mathbf{W}}_m^{\text{im}} \frac{1}{N_{\text{tr}}} \mathbf{X}_m^{\text{tr}\top} \mathbf{e}_m^{\text{tr}} \Big) \tag{29b}$$

where we define

$$\Sigma_{\theta_m} := \Big( \frac{1}{N_{\text{tr}}} \mathbf{X}_m^{\text{Tr}\top} \mathbf{X}_m^{\text{Tr}} + \gamma \mathbf{I} \Big)^{-1} = (\hat{\mathbf{Q}}_m^{\text{tr}} + \gamma \mathbf{I})^{-1} \tag{29c}$$

$$\hat{\mathbf{W}}_m^{\text{im}} := \gamma^2 \Sigma_{\theta_m} \frac{1}{N_{\text{va}}} \mathbf{X}_m^{\text{va}\top} \mathbf{X}_m^{\text{va}} \Sigma_{\theta_m} = \gamma^2 \Sigma_{\theta_m} \hat{\mathbf{Q}}_m^{\text{va}} \Sigma_{\theta_m}. \tag{29d}$$

The meta-test risk of iMAML is defined as

$$\mathcal{R}_{N_a}^{\text{im}}(\theta_0) = \mathbb{E}\big[ \big( y_m - \hat{\theta}_m^{\text{im}}(\theta_0, \mathcal{D}_{m,N_a})^\top x_m \big)^2 \big]$$

$$= \mathbb{E}_m \big[ \|\theta_0 - \theta_m^\star\|_{\mathbf{W}_{m,N_a}^{\text{im}}}^2 \big] + 1 + \frac{1}{N_a} \mathbb{E}[\gamma^{-2} \text{Tr}(\mathbf{W}_{m,N_a}^{\text{im}} \hat{\mathbf{Q}}_{m,N_a})] \tag{30a}$$

where the weight matrix is defined as

$$\mathbf{W}_{m,N_a}^{\text{im}} = \mathbb{E}_{x_m} \big[ (\hat{\mathbf{Q}}_{m,N_a} + \gamma \mathbf{I})^{-1} \mathbf{Q}_m (\hat{\mathbf{Q}}_{m,N_a} + \gamma \mathbf{I})^{-1} \big]$$

$$= \mathbf{W}_m^{\text{im}} \mathbb{E}_{x_m} \big[ \Sigma_{\theta_m} (\mathbf{Q}_m - \hat{\mathbf{Q}}_{m,N_a}) \mathbf{W}_m^{\text{im}} (\mathbf{Q}_m - \hat{\mathbf{Q}}_{m,N_a}) \Sigma_{\theta_m} + \Sigma_{\theta_m} (\mathbf{Q}_m - \hat{\mathbf{Q}}_{m,N_a}) \mathbf{W}_m^{\text{im}}$$

$$+ \mathbf{W}_m^{\text{im}} (\mathbf{Q}_m - \hat{\mathbf{Q}}_{m,N_a}) \Sigma_{\theta_m} \big] \tag{30b}$$

where $\mathbf{W}_m^{\text{im}} = (\gamma^{-1} \mathbf{Q}_m + \mathbf{I})^{-1} \mathbf{Q}_m (\gamma^{-1} \mathbf{Q}_m + \mathbf{I})^{-1}$.

Simplify the notation of $\mathbf{X}_{m,N_a}, \mathbf{y}_{m,N_a}, \hat{\mathbf{Q}}_{m,N_a}$ as $\mathbf{X}_m, \mathbf{y}_m, \hat{\mathbf{Q}}_m$. The derivation of (30) is given by

$$\mathcal{R}_{N_a}^{\text{im}}(\theta_0) = \mathbb{E}\big[ \|\hat{\theta}_m^{\text{im}}(\theta_0, \mathcal{D}_{m,N_a}) - \theta_m^\star\|_{\mathbf{Q}_m}^2 \big] + 1 \tag{31}$$

$$= \mathbb{E}\big[ \|(\hat{\mathbf{Q}}_m + \gamma \mathbf{I})^{-1} (\frac{1}{N_a} \mathbf{X}_m^\top \mathbf{y}_m + \gamma \theta_0) - \theta_m^\star\|_{\mathbf{Q}_m}^2 \big] + 1$$

$$\overset{(a)}{=} \mathbb{E}\Big[ \theta_0^\top \mathbf{W}_{m,N_a}^{\text{im}} \theta_0 + 2\gamma(\frac{1}{N_a} \mathbf{y}_m^\top \mathbf{X}_m \Sigma_{\theta_m} - \theta_m^{\star\top}) \mathbf{Q}_m \Sigma_{\theta_m} \theta_0 +$$

$$\frac{1}{N_a} \mathbf{y}_m^\top \mathbf{X}_m \Sigma_{\theta_m} \mathbf{Q}_m \Sigma_{\theta_m} \frac{1}{N_a} \mathbf{X}_m^\top \mathbf{y}_m - 2\theta_m^{\star\top} \mathbf{Q}_m \Sigma_{\theta_m} \frac{1}{N_a} \mathbf{X}_m^\top \mathbf{y}_m + \theta_m^{\star\top} \mathbf{Q}_m \theta_m^\star \Big] + 1$$

where $(a)$ follows from the definition of $\Sigma_{\theta_m} = (\hat{\mathbf{Q}}_m^{\mathrm{al}} + \gamma\mathbf{I})^{-1}$, and $\mathbf{W}_{m,N_a}^{\mathrm{im}} = \gamma^2\Sigma_{\theta_m}\mathbf{Q}_m\Sigma_{\theta_m}$.

Applying the fact that $\mathbf{y}_m = \mathbf{X}_m\theta_m^\star + \mathbf{e}_m$ and $\mathbb{E}_{\mathbf{e}_m}[\mathbf{e}_m] = \mathbf{0}$, one can further derive from (31) that

$$\mathcal{R}_{N_a}^{\mathrm{im}}(\theta_0) = \mathbb{E}\Big[\theta_0^\top\mathbf{W}_{m,N_a}^{\mathrm{im}}\theta_0 + 2\gamma(\theta_m^{\star\top}\hat{\mathbf{Q}}_m\Sigma_{\theta_m} - \theta_m^{\star\top})\mathbf{Q}_m\Sigma_{\theta_m}\theta_0 + \theta_m^{\star\top}\hat{\mathbf{Q}}_m\Sigma_{\theta_m}\mathbf{Q}_m\Sigma_{\theta_m}\hat{\mathbf{Q}}_m\theta_m^\star$$
$$- 2\theta_m^{\star\top}\mathbf{Q}_m\Sigma_{\theta_m}\hat{\mathbf{Q}}_m\theta_m^\star + \theta_m^{\star\top}\mathbf{Q}_m\theta_m^\star + \frac{1}{N_a^2}\mathbf{e}_m^\top\mathbf{X}_m\Sigma_{\theta_m}\mathbf{Q}_m\Sigma_{\theta_m}\mathbf{X}_m^\top\mathbf{e}_m\Big] + 1. \quad (32)$$

Based on the linearity of trace and expectation, and the cyclic property of trace, the last term inside the expectation in the above equation can be computed as

$$\mathbb{E}_{\mathbf{e}_m}[\mathbf{e}_m^\top\mathbf{X}_m\Sigma_{\theta_m}\mathbf{Q}_m\Sigma_{\theta_m}\mathbf{X}_m^\top\mathbf{e}_m^{\mathcal{A}}] = \mathrm{Tr}(\mathbf{X}_m\Sigma_{\theta_m}\mathbf{Q}_m\Sigma_{\theta_m}\mathbf{X}_m^\top\mathbb{E}_{\mathbf{e}_m}[\mathbf{e}_m^{\mathcal{A}}\mathbf{e}_m^\top])$$
$$= \mathrm{Tr}(\mathbf{X}_m\Sigma_{\theta_m}\mathbf{Q}_m\Sigma_{\theta_m}\mathbf{X}_m^\top) = N_a\mathrm{Tr}(\Sigma_{\theta_m}\mathbf{Q}_m\Sigma_{\theta_m}\hat{\mathbf{Q}}_m) = N_a\mathrm{Tr}(\mathbf{W}_{m,N_a}^{\mathrm{im}}\hat{\mathbf{Q}}_m).$$

To derive all the terms related to $\theta_m^\star$, based on the Woodbury matrix identity, $\mathbf{I} - \hat{\mathbf{Q}}_m\Sigma_{\theta_m} = \mathbf{I} - \Sigma_{\theta_m}\hat{\mathbf{Q}}_m = \gamma\Sigma_{\theta_m}$, we have

$$(\theta_m^{\star\top}\hat{\mathbf{Q}}_m\Sigma_{\theta_m} - \theta_m^{\star\top}) = \theta_m^{\star\top}(\hat{\mathbf{Q}}_m\Sigma_{\theta_m} - \mathbf{I}) = -\gamma\theta_m^{\star\top}\Sigma_{\theta_m} \quad (33)$$

and then the terms related to $\theta_m^\star$ in (32) can be computed by

$$\theta_m^{\star\top}\hat{\mathbf{Q}}_m\Sigma_{\theta_m}\mathbf{Q}_m\Sigma_{\theta_m}\hat{\mathbf{Q}}_m\theta_m^\star - 2\theta_m^{\star\top}\mathbf{Q}_m\Sigma_{\theta_m}\hat{\mathbf{Q}}_m\theta_m^\star + \theta_m^{\star\top}\mathbf{Q}_m\theta_m^\star$$
$$= \theta_m^{\star\top}\big((\hat{\mathbf{Q}}_m\Sigma_{\theta_m} - \mathbf{I})\mathbf{Q}_m\Sigma_{\theta_m}\hat{\mathbf{Q}}_m + \mathbf{Q}_m(\mathbf{I} - \Sigma_{\theta_m}\hat{\mathbf{Q}}_m)\big)\theta_m^\star$$
$$\overset{(a)}{=} \theta_m^{\star\top}\big(-\gamma\Sigma_{\theta_m}\mathbf{Q}_m\Sigma_{\theta_m}\hat{\mathbf{Q}}_m + \mathbf{Q}_m\gamma\Sigma_{\theta_m}\big)\theta_m^\star$$
$$\overset{(b)}{=} \gamma^{-1}\theta_m^{\star\top}\big(-\mathbf{W}_{m,N_a}^{\mathrm{im}}\hat{\mathbf{Q}}_m + (\hat{\mathbf{Q}}_m + \gamma\mathbf{I})\mathbf{W}_{m,N_a}^{\mathrm{im}}\big)\theta_m^\star \quad (34)$$

where $(a)$ follows from (33), and $(b)$ follows from the definition of $\mathbf{W}_{m,N_a}^{\mathrm{im}}$.

Combining (32) and (34) and rearranging the equations, we obtain

$$\mathcal{R}_{N_a}^{\mathrm{im}}(\theta_0) = \mathbb{E}\Big[\theta_0^\top\mathbf{W}_{m,N_a}^{\mathrm{im}}\theta_0 - 2\theta_m^{\star\top}\mathbf{W}_{m,N_a}^{\mathrm{im}}\theta_0 +$$
$$\gamma^{-1}\theta_m^{\star\top}\big(-\mathbf{W}_{m,N_a}^{\mathrm{im}}\hat{\mathbf{Q}}_m + (\hat{\mathbf{Q}}_m + \gamma\mathbf{I})\mathbf{W}_{m,N_a}^{\mathrm{im}}\big)\theta_m^\star + \frac{1}{N_a\gamma^2}\mathrm{Tr}(\mathbf{W}_{m,N_a}^{\mathrm{im}}\hat{\mathbf{Q}}_m)\Big] + 1$$
$$\overset{(c)}{=} \mathbb{E}\Big[\|\theta_0 - \theta_m^\star\|_{\mathbf{W}_{m,N_a}^{\mathrm{im}}}^2 + \gamma^{-1}\theta_m^{\star\top}\big(-\mathbf{W}_{m,N_a}^{\mathrm{im}}\hat{\mathbf{Q}}_m + \hat{\mathbf{Q}}_m\mathbf{W}_{m,N_a}^{\mathrm{im}}\big)\theta_m^\star + \frac{1}{N_a\gamma^2}\mathrm{Tr}(\mathbf{W}_{m,N_a}^{\mathrm{im}}\hat{\mathbf{Q}}_m)\Big] + 1$$
$$\overset{(d)}{=} \mathbb{E}\Big[\|\theta_0 - \theta_m^\star\|_{\mathbf{W}_{m,N_a}^{\mathrm{im}}}^2 + \frac{1}{N_a\gamma^2}\mathrm{Tr}(\mathbf{W}_{m,N_a}^{\mathrm{im}}\hat{\mathbf{Q}}_m)\Big] + 1 \quad (35)$$

where $(c)$ follows from rearranging the equations; $(d)$ follows from the fact that

$$\theta_m^{\star\top}(\mathbf{W}_{m,N_a}^{\mathrm{im}}\hat{\mathbf{Q}}_m)\theta_m^\star = \big(\theta_m^{\star\top}(\mathbf{W}_{m,N_a}^{\mathrm{im}}\hat{\mathbf{Q}}_m)\theta_m^\star\big)^\top = \theta_m^{\star\top}(\hat{\mathbf{Q}}_m\mathbf{W}_{m,N_a}^{\mathrm{im}})\theta_m^\star. \quad (36)$$

Since $\lim_{N_a\to\infty}\frac{1}{N_a}\mathbb{E}[\gamma^{-2}\mathrm{Tr}(\mathbf{W}_{m,N_a}^{\mathrm{im}}\hat{\mathbf{Q}}_{m,N_a})] = 0$, from the definition of the population risk in (2), the population risk of iMAML is given by

$$\mathcal{R}^{\mathrm{im}}(\theta_0) := \lim_{N_a\to\infty}\mathcal{R}_{N_a}^{\mathrm{im}}(\theta_0) = \mathbb{E}_m\big[\|\theta_0 - \theta_m^\star\|_{\mathbf{W}_m^{\mathrm{im}}}^2\big] + 1 \quad (37a)$$
$$\text{with}\quad \mathbf{W}_m^{\mathrm{im}} = (\gamma^{-1}\mathbf{Q}_m + \mathbf{I})^{-1}\mathbf{Q}_m(\gamma^{-1}\mathbf{Q}_m + \mathbf{I})^{-1} \quad (37b)$$

whose minimizer is given by

$$\theta_0^{\mathrm{im}} = \arg\min_{\theta_0}\mathcal{R}^{\mathrm{im}}(\theta_0) = \mathbb{E}_m\big[\mathbf{W}_m^{\mathrm{im}}\big]^{-1}\mathbb{E}_m\big[\mathbf{W}_m^{\mathrm{im}}\theta_m^\star\big]. \quad (38)$$

The above discussion provides proof for Proposition 1.

## C  Proof of Theorem 1

Section B gives solutions to the empirical and population risks. In this section, we provide proof to the main theorem, starting with the decomposition of the excess risk in Proposition 2. Note that our proof of the bound on the variance follows the idea of [5] by separately bounding the terms related to the first $k$ largest eigenvalues and the rest eigenvalues of the per-task weight matrices.

## C.1 Proof of Proposition 2

Next we analyze the excess risk defined in (9) based on the solutions of MAML and iMAML. First we restate the complete version of Proposition 2 in Lemma 4.

**Lemma 4** (Restatement of Proposition 2). *With probability at least $1 - \delta$, the excess risk of the MAML with the minimum-norm solution is bounded by*

$$\mathcal{E}^{\mathcal{A}}(\hat{\theta}_0) \lesssim \underbrace{\left\|(\sum_{m=1}^{M} \hat{\mathbf{W}}_m^{\mathcal{A}})^{\dagger}(\sum_{m=1}^{M} \hat{\mathbf{W}}_m^{\mathcal{A}}(\theta_m^{\star} - \theta_0))\right\|_{\mathbf{W}^{\mathcal{A}}}^2}_{\mathcal{E}_{\theta_m^*}} + \underbrace{\theta_0^{\top} \mathbf{B}^{\mathcal{A}} \theta_0}_{\mathcal{E}_b} + \underbrace{c_1 \sigma^2 \log \frac{1}{\delta} \mathrm{Tr}(\mathbf{C}^{\mathcal{A}})}_{\mathcal{E}_{\epsilon_m}} \quad (39)$$

*where the weight matrix and the constants are defined as*

$$\mathbf{W}^{\mathcal{A}} := \mathbb{E}_m[\mathbf{W}_m^{\mathcal{A}}], \quad \tilde{\mathbf{X}}^{\mathrm{ma}} := [\mathbf{X}_m^{\mathrm{va}}(\mathbf{I} - \alpha \hat{\mathbf{Q}}_m^{\mathrm{tr}})], \quad \tilde{\mathbf{X}}^{\mathrm{im}} := [\mathbf{X}_m^{\mathrm{va}}(\mathbf{I} + \gamma^{-1} \hat{\mathbf{Q}}_m^{\mathrm{tr}})^{-1}]$$

$$\mathbf{B}^{\mathcal{A}} := \left(\tilde{\mathbf{X}}^{\mathcal{A}\top}(\tilde{\mathbf{X}}^{\mathcal{A}}\tilde{\mathbf{X}}^{\mathcal{A}\top})^{-1}\tilde{\mathbf{X}}^{\mathcal{A}} - \mathbf{I}\right)\mathbf{W}^{\mathcal{A}}\left(\tilde{\mathbf{X}}^{\mathcal{A}\top}(\tilde{\mathbf{X}}^{\mathcal{A}}\tilde{\mathbf{X}}^{\mathcal{A}\top})^{-1}\tilde{\mathbf{X}}^{\mathcal{A}} - \mathbf{I}\right),$$

$$\mathbf{C}^{\mathcal{A}} = \mathbf{C}_1^{\mathcal{A}} + \mathbf{C}_2^{\mathcal{A}}, \quad \mathbf{C}_1^{\mathcal{A}} := (\tilde{\mathbf{X}}^{\mathcal{A}}\tilde{\mathbf{X}}^{\mathcal{A}\top})^{-1}\tilde{\mathbf{X}}^{\mathcal{A}}\mathbf{W}^{\mathcal{A}}\tilde{\mathbf{X}}^{\mathcal{A}\top}(\tilde{\mathbf{X}}^{\mathcal{A}}\tilde{\mathbf{X}}^{\mathcal{A}\top})^{-1},$$

$$\mathbf{C}_2^{\mathrm{ma}} := \frac{\alpha^2}{N_{\mathrm{tr}}}\mathbf{C}_1^{\mathrm{ma}}\mathrm{diag}[\mathbf{X}_m^{\mathrm{va}}\hat{\mathbf{Q}}_m^{\mathrm{tr}}\mathbf{X}_m^{\mathrm{va}\top}]$$

$$\mathbf{C}_2^{\mathrm{im}} := \frac{1}{N_{\mathrm{tr}}}\mathbf{C}_1^{\mathrm{im}}\mathrm{diag}[\mathbf{X}_m^{\mathrm{va}}(\mathbf{I} + \gamma^{-1}\hat{\mathbf{Q}}_m^{\mathrm{tr}})^{-1}\hat{\mathbf{Q}}_m^{\mathrm{tr}}(\mathbf{I} + \gamma^{-1}\hat{\mathbf{Q}}_m^{\mathrm{tr}})^{-1}\mathbf{X}_m^{\mathrm{va}\top}].$$

*Note that $\mathbf{C}_2^{\mathcal{A}}$ can be either $\mathbf{C}_2^{\mathrm{ma}}$ for MAML or $\mathbf{C}_2^{\mathrm{im}}$ for iMAML.*

*Proof.* The excess risk $\mathcal{E}^{\mathcal{A}}$ can be derived as

$$\mathcal{E}^{\mathcal{A}}(\hat{\theta}_0) := \mathcal{R}(\hat{\theta}_0) - \mathcal{R}(\theta_0) = \mathbb{E}_m\left[\|\hat{\theta}_0 - \theta_m^{\star}\|_{\mathbf{W}_m^{\mathcal{A}}}^2\right] - \mathbb{E}_m\left[\|\theta_0 - \theta_m^{\star}\|_{\mathbf{W}_m^{\mathcal{A}}}^2\right]$$

$$= \hat{\theta}_0^{\top}\mathbf{W}\hat{\theta}_0 - \theta_0^{\top}\mathbf{W}\theta_0 - 2(\hat{\theta}_0 - \theta_0)^{\top}\mathbb{E}_m[\mathbf{W}_m\theta_m^{\star}] = \hat{\theta}_0^{\top}\mathbf{W}\hat{\theta}_0 - \theta_0^{\top}\mathbf{W}\theta_0 - 2(\hat{\theta}_0 - \theta_0)^{\top}\mathbf{W}\theta_0$$

$$= \hat{\theta}_0^{\top}\mathbf{W}\hat{\theta}_0 - 2\hat{\theta}_0^{\top}\mathbf{W}\theta_0 + \theta_0^{\top}\mathbf{W}\theta_0 = \|\hat{\theta}_0 - \theta_0\|_{\mathbf{W}^{\mathcal{A}}}^2$$

$$= \left\|\left(\sum_{m=1}^{M}\hat{\mathbf{W}}_m\right)^{\dagger}\left(\sum_{m=1}^{M}\hat{\mathbf{W}}_m\theta_m^{\star}\right) + \Delta_M - \theta_0\right\|_{\mathbf{W}^{\mathcal{A}}}^2$$

$$\leq 2\underbrace{\left\|\left(\sum_m\hat{\mathbf{W}}_m\right)^{\dagger}\left(\sum_m\hat{\mathbf{W}}_m\theta_m^{\star}\right) - \theta_0\right\|_{\mathbf{W}^{\mathcal{A}}}^2}_{I_1} + 2\underbrace{\|\Delta_M\|_{\mathbf{W}^{\mathcal{A}}}^2}_{I_2}. \quad (40)$$

In (40), $I_1$ can be bounded by

$$I_1 = \left\|\left(\sum_m\hat{\mathbf{W}}_m\right)^{\dagger}\left(\sum_m\hat{\mathbf{W}}_m\theta_m^{\star}\right) - \theta_0\right\|_{\mathbf{W}^{\mathcal{A}}}^2$$

$$= \left\|\left(\sum_m\hat{\mathbf{W}}_m\right)^{\dagger}\left(\sum_m\hat{\mathbf{W}}_m(\theta_m^{\star} - \theta_0)\right) + \left(\left(\sum_m\hat{\mathbf{W}}_m\right)^{\dagger}\left(\sum_m\hat{\mathbf{W}}_m\right) - \mathbf{I}\right)\theta_0\right\|_{\mathbf{W}^{\mathcal{A}}}^2$$

$$\leq 2\left\|\left(\sum_m\hat{\mathbf{W}}_m\right)^{\dagger}\left(\sum_m\hat{\mathbf{W}}_m(\theta_m^{\star} - \theta_0)\right)\right\|_{\mathbf{W}^{\mathcal{A}}}^2 + 2\left\|\left(\left(\sum_m\hat{\mathbf{W}}_m\right)^{\dagger}\left(\sum_m\hat{\mathbf{W}}_m\right) - \mathbf{I}\right)\theta_0\right\|_{\mathbf{W}^{\mathcal{A}}}^2$$

$$= 2\left\|\left(\sum_m\hat{\mathbf{W}}_m\right)^{\dagger}\left(\sum_m\hat{\mathbf{W}}_m(\theta_m^{\star} - \theta_0)\right)\right\|_{\mathbf{W}^{\mathcal{A}}}^2 + 2\theta_0^{\top}\mathbf{B}\theta_0 \quad (41)$$

with the matrix $\mathbf{B}$ defined as

$$\mathbf{B} = \left(\left(\sum_m\hat{\mathbf{W}}_m\right)^{\dagger}\left(\sum_m\hat{\mathbf{W}}_m\right) - \mathbf{I}\right)\mathbf{W}^{\mathcal{A}}\left(\left(\sum_m\hat{\mathbf{W}}_m\right)^{\dagger}\left(\sum_m\hat{\mathbf{W}}_m\right) - \mathbf{I}\right)$$

$$\overset{(a)}{=} \left((\tilde{\mathbf{X}}^{\top}\tilde{\mathbf{X}})^{\dagger}\tilde{\mathbf{X}}^{\top}\tilde{\mathbf{X}} - \mathbf{I}\right)\mathbf{W}^{\mathcal{A}}\left((\tilde{\mathbf{X}}^{\top}\tilde{\mathbf{X}})^{\dagger}\tilde{\mathbf{X}}^{\top}\tilde{\mathbf{X}} - \mathbf{I}\right)$$

$$= \left(\tilde{\mathbf{X}}^{\top}(\tilde{\mathbf{X}}\tilde{\mathbf{X}}^{\top})^{-1}\tilde{\mathbf{X}} - \mathbf{I}\right)\mathbf{W}^{\mathcal{A}}\left(\tilde{\mathbf{X}}^{\top}(\tilde{\mathbf{X}}\tilde{\mathbf{X}}^{\top})^{-1}\tilde{\mathbf{X}} - \mathbf{I}\right). \quad (42)$$

And (a) is from the relationship of $\hat{\mathbf{W}}$ and $\tilde{\mathbf{X}}$, recall we use $[\cdot]$ to represent row concatenation of matrices or vectors.

In (40), $I_2$ can be bounded by

$$I_2 = \left\| \Big( \sum_{m=1}^{M} \hat{\mathbf{W}}_m^{\mathcal{A}} \Big)^{\dagger} \Big( \sum_{m=1}^{M} (\mathbf{I} - \alpha \hat{\mathbf{Q}}_m^{\mathrm{tr}}) \frac{1}{N_2} \mathbf{X}_m^{\mathrm{va}\top} \mathbf{e}_m^{\mathrm{va}} - (\mathbf{I} - \alpha \hat{\mathbf{Q}}_m^{\mathrm{tr}}) \hat{\mathbf{Q}}_m^{\mathrm{va}} \frac{\alpha}{N_{\mathrm{tr}}} \mathbf{X}_m^{\mathrm{tr}\top} \mathbf{e}_m^{\mathrm{tr}} \Big) \right\|_{\mathbf{W}^{\mathcal{A}}}^2$$

$$\overset{(b)}{=} [\mathbf{e}_m^{\mathrm{va}}]^{\top} \mathbf{C}_1^{\mathcal{A}} [\mathbf{e}_m^{\mathrm{va}}] + [\mathbf{e}_m^{\mathrm{tr}}]^{\top} \mathbf{C}_2^{\mathcal{A}} [\mathbf{e}_m^{\mathrm{tr}}] - 2[\mathbf{e}_m^{\mathrm{va}}]^{\top} C_3^{\mathcal{A}} [\mathbf{e}_m^{\mathrm{tr}}]$$

$$\leq 2[\mathbf{e}_m^{\mathrm{va}}]^{\top} \mathbf{C}_1^{\mathcal{A}} [\mathbf{e}_m^{\mathrm{va}}] + 2[\mathbf{e}_m^{\mathrm{tr}}]^{\top} \mathbf{C}_2^{\mathcal{A}} [\mathbf{e}_m^{\mathrm{tr}}]$$

$$= 2\mathrm{Tr}(\mathbf{C}_1^{\mathcal{A}} [\mathbf{e}_m^{\mathrm{va}}][\mathbf{e}_m^{\mathrm{va}}]^{\top} + \mathbf{C}_2^{\mathcal{A}} [\mathbf{e}_m^{\mathrm{tr}}][\mathbf{e}_m^{\mathrm{tr}}]^{\top})$$

$$= 2\mathrm{Tr}(\mathbf{C}_1^{\mathcal{A}} + \mathbf{C}_2^{\mathcal{A}}) + 2\mathrm{Tr}\big(\mathbf{C}_1^{\mathcal{A}}([\mathbf{e}_m^{\mathrm{va}}][\mathbf{e}_m^{\mathrm{va}}]^{\top} - \mathbf{I}) + \mathbf{C}_2^{\mathcal{A}}([\mathbf{e}_m^{\mathrm{tr}}][\mathbf{e}_m^{\mathrm{tr}}]^{\top} - \mathbf{I})\big)$$

where $(b)$ follows from expanding the quadratic terms, and

$$\mathbf{C}_1^{\mathcal{A}} = \frac{1}{N^2} \tilde{\mathbf{X}} \Big( \sum_{m=1}^{M} \hat{\mathbf{W}}_m^{\mathcal{A}} \Big)^{\dagger} \mathbf{W}^{\mathcal{A}} \Big( \sum_{m=1}^{M} \hat{\mathbf{W}}_m^{\mathcal{A}} \Big)^{\dagger} \tilde{\mathbf{X}}^{\top}$$

$$= \tilde{\mathbf{X}} (\tilde{\mathbf{X}}^{\top} \tilde{\mathbf{X}})^{\dagger} \mathbf{W}^{\mathcal{A}} (\tilde{\mathbf{X}}^{\top} \tilde{\mathbf{X}})^{\dagger} \tilde{\mathbf{X}}^{\top} = (\tilde{\mathbf{X}} \tilde{\mathbf{X}}^{\top})^{-1} \tilde{\mathbf{X}} \mathbf{W}^{\mathcal{A}} \tilde{\mathbf{X}}^{\top} (\tilde{\mathbf{X}} \tilde{\mathbf{X}}^{\top})^{-1}, \tag{43}$$

$$\mathbf{C}_2^{\mathrm{ma}} = \frac{\alpha^2}{N_{\mathrm{tr}}^2} [\mathbf{X}_m^{\mathrm{tr}} \mathbf{X}_m^{\mathrm{va}\top} \tilde{\mathbf{X}}_m] \Big( \sum_{m=1}^{M} \hat{\mathbf{W}}_m^{\mathcal{A}} \Big)^{\dagger} \mathbf{W}^{\mathcal{A}} \Big( \sum_{m=1}^{M} \hat{\mathbf{W}}_m^{\mathcal{A}} \Big)^{\dagger} [\mathbf{X}_m^{\mathrm{tr}} \mathbf{X}_m^{\mathrm{va}\top} \tilde{\mathbf{X}}_m]^{\top}$$

$$= \frac{\alpha^2}{N_{\mathrm{tr}}^2} [\mathbf{X}_m^{\mathrm{tr}} \mathbf{X}_m^{\mathrm{va}\top} \tilde{\mathbf{X}}_m] \tilde{\mathbf{X}}^{\top} (\tilde{\mathbf{X}} \tilde{\mathbf{X}}^{\top})^{-2} \tilde{\mathbf{X}} \mathbf{W}^{\mathcal{A}} \tilde{\mathbf{X}}^{\top} (\tilde{\mathbf{X}} \tilde{\mathbf{X}}^{\top})^{-2} \tilde{\mathbf{X}} [\mathbf{X}_m^{\mathrm{tr}} \mathbf{X}_m^{\mathrm{va}\top} \tilde{\mathbf{X}}_m]^{\top}. \tag{44}$$

By taking the expectation w.r.t. $\mathbf{e}_m$, we need to bound $\mathrm{Tr}(\mathbf{C}_1)$, $\mathrm{Tr}(\mathbf{C}_2)$. Based on the cyclic property of trace, $\mathrm{Tr}(\mathbf{C}_2^{\mathrm{ma}})$ can be further derived as

$$\mathrm{Tr}(\mathbf{C}_2^{\mathrm{ma}}) = \frac{\alpha^2}{N_{\mathrm{tr}}^2} \mathrm{Tr}\Big( [\mathbf{X}_m^{\mathrm{tr}} \mathbf{X}_m^{\mathrm{va}\top} \tilde{\mathbf{X}}_m] \tilde{\mathbf{X}}^{\top} (\tilde{\mathbf{X}} \tilde{\mathbf{X}}^{\top})^{-2} \tilde{\mathbf{X}} \mathbf{W}^{\mathcal{A}} \tilde{\mathbf{X}}^{\top} (\tilde{\mathbf{X}} \tilde{\mathbf{X}}^{\top})^{-2} \tilde{\mathbf{X}} [\mathbf{X}_m^{\mathrm{tr}} \mathbf{X}_m^{\mathrm{va}\top} \tilde{\mathbf{X}}_m]^{\top} \Big)$$

$$= \frac{\alpha^2}{N_{\mathrm{tr}}^2} \mathrm{Tr}\Big( \tilde{\mathbf{X}}^{\top} (\tilde{\mathbf{X}} \tilde{\mathbf{X}}^{\top})^{-2} \tilde{\mathbf{X}} \mathbf{W}^{\mathcal{A}} \tilde{\mathbf{X}}^{\top} (\tilde{\mathbf{X}} \tilde{\mathbf{X}}^{\top})^{-2} \tilde{\mathbf{X}} \sum_{m=1}^{M} \tilde{\mathbf{X}}_m^{\top} \mathbf{X}_m^{\mathrm{va}} \mathbf{X}_m^{t\top} \mathbf{X}_m^{\mathrm{tr}} \mathbf{X}_m^{\mathrm{va}\top} \tilde{\mathbf{X}}_m \Big)$$

$$= \frac{\alpha^2}{N_{\mathrm{tr}}^2} \mathrm{Tr}\Big( \tilde{\mathbf{X}}^{\top} (\tilde{\mathbf{X}} \tilde{\mathbf{X}}^{\top})^{-2} \tilde{\mathbf{X}} \mathbf{W}^{\mathcal{A}} \tilde{\mathbf{X}}^{\top} (\tilde{\mathbf{X}} \tilde{\mathbf{X}}^{\top})^{-2} \tilde{\mathbf{X}} \tilde{\mathbf{X}}^{\top} [\mathbf{X}_m^{\mathrm{va}} \mathbf{X}_m^{t\top} \mathbf{X}_m^{\mathrm{tr}} \mathbf{X}_m^{\mathrm{va}\top} \tilde{\mathbf{X}}_m] \Big).$$

Then $\mathrm{Tr}(\mathbf{C}_2^{\mathrm{ma}})$ can be further written as

$$\mathrm{Tr}(\mathbf{C}_2^{\mathrm{ma}}) = \frac{\alpha^2}{N_{\mathrm{tr}}^2} \mathrm{Tr}\Big( \tilde{\mathbf{X}}^{\top} (\tilde{\mathbf{X}} \tilde{\mathbf{X}}^{\top})^{-2} \tilde{\mathbf{X}} \mathbf{W}^{\mathcal{A}} \tilde{\mathbf{X}}^{\top} (\tilde{\mathbf{X}} \tilde{\mathbf{X}}^{\top})^{-1} \mathrm{diag}[\mathbf{X}_m^{\mathrm{va}} \mathbf{X}_m^{t\top} \mathbf{X}_m^{\mathrm{tr}} \mathbf{X}_m^{\mathrm{va}\top}] \tilde{\mathbf{X}} \Big)$$

$$= \frac{\alpha^2}{N_{\mathrm{tr}}^2} \mathrm{Tr}\Big( (\tilde{\mathbf{X}} \tilde{\mathbf{X}}^{\top})^{-1} \tilde{\mathbf{X}} \mathbf{W}^{\mathcal{A}} \tilde{\mathbf{X}}^{\top} (\tilde{\mathbf{X}} \tilde{\mathbf{X}}^{\top})^{-1} \mathrm{diag}[\mathbf{X}_m^{\mathrm{va}} \mathbf{X}_m^{\mathrm{tr}\top} \mathbf{X}_m^{\mathrm{tr}} \mathbf{X}_m^{\mathrm{va}\top}] \Big)$$

$$= \frac{\alpha^2}{N_{\mathrm{tr}}} \mathrm{Tr}\Big( (\tilde{\mathbf{X}} \tilde{\mathbf{X}}^{\top})^{-1} \tilde{\mathbf{X}} \mathbf{W}^{\mathcal{A}} \tilde{\mathbf{X}}^{\top} (\tilde{\mathbf{X}} \tilde{\mathbf{X}}^{\top})^{-1} \mathrm{diag}[\mathbf{X}_m^{\mathrm{va}} \hat{\mathbf{Q}}_m^{\mathrm{tr}} \mathbf{X}_m^{\mathrm{va}\top}] \Big). \tag{45}$$

Since we have

$$\mathbb{E}_{\epsilon}[I_2] = \mathbb{E}_{\epsilon}\big[ [\mathbf{e}_m^{\mathrm{va}}]^{\top} \mathbf{C}_1 [\mathbf{e}_m^{\mathrm{va}}] \big] + \mathbb{E}_{\epsilon}\big[ [\mathbf{e}_m^{\mathrm{tr}}]^{\top} \mathbf{C}_2 [\mathbf{e}_m^{\mathrm{tr}}] \big]$$

$$= \mathrm{Tr}(\mathbf{C}_1 \, \mathrm{Cov}[[\mathbf{e}_m^{\mathrm{va}}]]) + \mathrm{Tr}(\mathbf{C}_2 \, \mathrm{Cov}[[\mathbf{e}_m^{\mathrm{tr}}]]) = \sigma^2 \mathrm{Tr}(\mathbf{C}_1 + \mathbf{C}_2)$$

by the subGaussian concentration inequality [47], it holds with probability at least $1 - \delta$ over $\epsilon$ that

$$2I_2 \leq c_1 \sigma^2 \log \frac{1}{\delta} \mathrm{Tr}(\mathbf{C}_1 + \mathbf{C}_2). \tag{46}$$

Combining the bounds for $I_1$ and $I_2$ in (41) and (46) completes the proof. $\qquad \square$

## C.2 Proof of Lemma 1

Define

$$\Delta_{\theta_{\mathcal{A}}} := \left[ (\theta_1^\star - \theta_0^{\mathcal{A}})^\top, \ldots, (\theta_M^\star - \theta_0^{\mathcal{A}})^\top \right]^\top \in \mathbb{R}^{dM},$$

$$\mathbf{U}_{\mathcal{A}} := \left[ \hat{\mathbf{W}}_1^{\mathcal{A}} \Big( \sum_{m=1}^M \hat{\mathbf{W}}_m^{\mathcal{A}} \Big)^\dagger, \ldots, \hat{\mathbf{W}}_M^{\mathcal{A}} \Big( \sum_{m=1}^M \hat{\mathbf{W}}_m^{\mathcal{A}} \Big)^\dagger \right]^\top \in \mathbb{R}^{dM \times d}.$$

Then we can derive that

$$\left\| \Big( \sum_{m=1}^M \hat{\mathbf{W}}_m^{\mathcal{A}} \Big)^\dagger \Big( \sum_{m=1}^M \hat{\mathbf{W}}_m^{\mathcal{A}} (\theta_m^\star - \theta_0) \Big) \right\|_{\mathbf{W}^{\mathcal{A}}}^2 = \| \mathbf{U}_{\mathcal{A}}^\top \Delta_{\theta_{\mathcal{A}}} \|_{\mathbf{W}^{\mathcal{A}}}^2.$$

By the Hanson-Wright inequality, with probability at least $1 - \delta$ over $\theta_m^\star$, we have

$$\left| \left\| \mathbf{U}_{\mathcal{A}}^\top \Delta_{\theta_{\mathcal{A}}} \right\|_{\mathbf{W}^{\mathcal{A}}}^2 - \mathbb{E}_{\theta_m^\star | \hat{\mathbf{W}}_m^{\mathcal{A}}} \left[ \left\| \mathbf{U}_{\mathcal{A}}^\top \Delta_{\theta_{\mathcal{A}}} \right\|_{\mathbf{W}^{\mathcal{A}}}^2 \right] \right| = \widetilde{\mathcal{O}} \Big( \frac{R^2}{M \sqrt{d}} \Big). \tag{47}$$

To compute $\mathbb{E}_{\theta_m^\star | \hat{\mathbf{W}}_m^{\mathcal{A}}} \left[ \left\| \mathbf{U}_{\mathcal{A}}^\top \Delta_{\theta_{\mathcal{A}}} \right\|_{\mathbf{W}^{\mathcal{A}}}^2 \right]$, first recall $\mathrm{Cov}[\theta_m^\star] = \frac{R^2}{d} \mathbf{I}$, then we have

$$\mathbb{E}_{\theta_m^\star | \hat{\mathbf{W}}_m^{\mathcal{A}}} [\Delta_{\theta_{\mathcal{A}}}^\top \mathbf{U}_{\mathcal{A}} \mathbf{W}^{\mathcal{A}} \mathbf{U}_{\mathcal{A}}^\top \Delta_{\theta_{\mathcal{A}}}] = \frac{R^2}{d} \Big\langle \Big( \sum_{m=1}^M \hat{\mathbf{W}}_m^{\mathcal{A}} \Big)^\dagger \mathbf{W}^{\mathcal{A}} \Big( \sum_{m=1}^M \hat{\mathbf{W}}_m^{\mathcal{A}} \Big)^\dagger, \sum_{m=1}^M (\hat{\mathbf{W}}_m^{\mathcal{A}})^2 \Big\rangle$$

$$= \frac{R^2}{d} \mathrm{Tr} \Big( \tilde{\mathbf{X}} \Big( \sum_{m=1}^M \hat{\mathbf{W}}_m^{\mathcal{A}} \Big)^\dagger \mathbf{W}^{\mathcal{A}} \Big( \sum_{m=1}^M \hat{\mathbf{W}}_m^{\mathcal{A}} \Big)^\dagger \tilde{\mathbf{X}}^\top \mathrm{diag}[\tilde{\mathbf{X}}_m \tilde{\mathbf{X}}_m^\top] \Big)$$

$$= \frac{R^2}{d} \mathrm{Tr} \Big( \tilde{\mathbf{X}} (\tilde{\mathbf{X}}^\top \tilde{\mathbf{X}})^\dagger \mathbf{W}^{\mathcal{A}} (\tilde{\mathbf{X}}^\top \tilde{\mathbf{X}})^\dagger \tilde{\mathbf{X}}^\top \mathrm{diag}[\tilde{\mathbf{X}}_m \tilde{\mathbf{X}}_m^\top] \Big)$$

$$= \frac{R^2}{d} \mathrm{Tr}(\mathbf{C}_1^{\mathcal{A}} \mathrm{diag}[\tilde{\mathbf{X}}_m \tilde{\mathbf{X}}_m^\top]) \leq \frac{R^2}{d} \mathrm{Tr}(\mathbf{C}_1^{\mathcal{A}}) \| \mathrm{diag}[\mathbf{X}_m^{\mathrm{va}} \mathbf{X}_m^{\mathrm{va}\top}] \|$$

$$\leq \frac{R^2}{d} \mathrm{Tr}(\mathbf{C}_1^{\mathcal{A}}) \max_{m \in [M]} \| \mathbf{X}_m^{\mathrm{va}} \mathbf{X}_m^{\mathrm{va}\top} \| \leq \frac{R^2}{d} \mathrm{Tr}(\mathbf{C}_1^{\mathcal{A}}) \max_{m \in [M]} \| \mathbf{X}_m^{\mathrm{va}} \mathbf{X}_m^{\mathrm{va}\top} \| \tag{48}$$

where from Lemma 19, with high probability $\| \mathbf{X}_m^{\mathrm{va}} \mathbf{X}_m^{\mathrm{va}\top} \|$ can be bounded by

$$\| \mathbf{X}_m^{\mathrm{va}} \mathbf{X}_m^{\mathrm{va}\top} \| = \| \mathbf{X}_m^{\mathrm{va}\top} \mathbf{X}_m^{\mathrm{va}} \| \lesssim \Big( \sum_{i=1}^d \lambda_{mi}^2 + \lambda_{m1}^2 N_2 \Big) \leq \mathcal{O}(N_{\mathrm{va}}). \tag{49}$$

Combining (47), (48) and (49) leads to the following with high probability

$$\mathbb{E}_{\theta_m^\star | \hat{\mathbf{W}}_m^{\mathcal{A}}} [\Delta_{\theta_{\mathcal{A}}}^\top \mathbf{U}_{\mathcal{A}} \mathbf{W}^{\mathcal{A}} \mathbf{U}_{\mathcal{A}}^\top \Delta_{\theta_{\mathcal{A}}}] \leq \frac{R^2}{d} \mathrm{Tr}(\mathbf{C}_1^{\mathcal{A}}) \max_{m \in [M]} \| \mathbf{X}_m^{\mathrm{va}} \mathbf{X}_m^{\mathrm{va}\top} \| \lesssim \frac{R^2 N_{\mathrm{va}}}{d} \mathrm{Tr}(\mathbf{C}_1^{\mathcal{A}})$$

which proves that this term $\mathbb{E}_{\theta_m^\star | \hat{\mathbf{W}}_m^{\mathcal{A}}} [\Delta_{\theta_{\mathcal{A}}}^\top \mathbf{U}_{\mathcal{A}} \mathbf{W}^{\mathcal{A}} \mathbf{U}_{\mathcal{A}}^\top \Delta_{\theta_{\mathcal{A}}}]$ is non-dominant compared to $\mathrm{Tr}(\mathbf{C}_1^{\mathcal{A}})$.

## C.3 Proof of Lemma 2

*Proof.* Recall $\mathbf{B} := \left( \tilde{\mathbf{X}}^\top (\tilde{\mathbf{X}} \tilde{\mathbf{X}}^\top)^{-1} \tilde{\mathbf{X}} - \mathbf{I} \right) \mathbf{W} \left( \tilde{\mathbf{X}}^\top (\tilde{\mathbf{X}} \tilde{\mathbf{X}}^\top)^{-1} \tilde{\mathbf{X}} - \mathbf{I} \right)$. First note that

$$\left( \tilde{\mathbf{X}}^\top (\tilde{\mathbf{X}} \tilde{\mathbf{X}}^\top)^{-1} \tilde{\mathbf{X}} - \mathbf{I} \right) \tilde{\mathbf{X}}^\top = \tilde{\mathbf{X}}^\top - \tilde{\mathbf{X}}^\top = \mathbf{0}. \tag{50}$$

Thus, for any $\mathbf{u}$ in the column space of $\tilde{\mathbf{X}}^\top$, $\mathbf{u}$ can be represented as $\mathbf{u} = \tilde{\mathbf{X}}^\top \bar{\mathbf{u}}, \bar{\mathbf{u}} \neq \mathbf{0}$, then we have

$$\left( \tilde{\mathbf{X}}^\top (\tilde{\mathbf{X}} \tilde{\mathbf{X}}^\top)^{-1} \tilde{\mathbf{X}} - \mathbf{I} \right) \mathbf{u} = \mathbf{0}. \tag{51}$$

And for any $\mathbf{u}$ orthogonal to the colomn space of $\tilde{\mathbf{X}}^\top$, $\tilde{\mathbf{X}} \mathbf{u} = \mathbf{0}$, therefore

$$\left( \tilde{\mathbf{X}}^\top (\tilde{\mathbf{X}} \tilde{\mathbf{X}}^\top)^{-1} \tilde{\mathbf{X}} - \mathbf{I} \right) \mathbf{u} = -\mathbf{u}. \tag{52}$$

Since any $\mathbf{u} \in \mathbb{R}^d$ can be represented as a combination of a vector in the colomn space of $\tilde{\mathbf{X}}^\top$ and a vector orthogonal to the colomn space of $\tilde{\mathbf{X}}^\top$, $(\tilde{\mathbf{X}}^\top(\tilde{\mathbf{X}}\tilde{\mathbf{X}}^\top)^{-1}\tilde{\mathbf{X}} - \mathbf{I})$ has eigenvalues whose absolute values are smaller than 1, i.e.

$$\left\|\tilde{\mathbf{X}}^\top(\tilde{\mathbf{X}}\tilde{\mathbf{X}}^\top)^{-1}\tilde{\mathbf{X}} - \mathbf{I}\right\| \leq 1. \tag{53}$$

Then let $\mathbf{M} = \left(\tilde{\mathbf{X}}^\top(\tilde{\mathbf{X}}\tilde{\mathbf{X}}^\top)^{-1}\tilde{\mathbf{X}} - \mathbf{I}\right)$, expanding $\theta_0^\top \mathbf{B}\theta_0$, we have

$$
\begin{aligned}
\theta_0^\top \mathbf{B}\theta_0 &= \theta_0^\top \left(\tilde{\mathbf{X}}^\top(\tilde{\mathbf{X}}\tilde{\mathbf{X}}^\top)^{-1}\tilde{\mathbf{X}} - \mathbf{I}\right)\mathbf{W}\left(\tilde{\mathbf{X}}^\top(\tilde{\mathbf{X}}\tilde{\mathbf{X}}^\top)^{-1}\tilde{\mathbf{X}} - \mathbf{I}\right)\theta_0 \\
&\overset{(a)}{=} \theta_0^\top \mathbf{M}\left(\mathbf{W} - \frac{1}{MN_{\mathrm{va}}}\tilde{\mathbf{X}}^\top\tilde{\mathbf{X}}\right)\mathbf{M}\theta_0 \\
&= \theta_0^\top \mathbf{M}\left(\mathbf{W} - \frac{1}{MN_{\mathrm{va}}}\bar{\mathbf{X}}^\top\bar{\mathbf{X}} + \frac{1}{MN_{\mathrm{va}}}\bar{\mathbf{X}}^\top\bar{\mathbf{X}} - \frac{1}{MN_{\mathrm{va}}}\tilde{\mathbf{X}}^\top\tilde{\mathbf{X}}\right)\mathbf{M}\theta_0 \\
&\overset{(b)}{\leq} \left\|\mathbf{W} - \frac{1}{MN_{\mathrm{va}}}\bar{\mathbf{X}}^\top\bar{\mathbf{X}}\right\|\|\theta_0\|^2 + \frac{1}{MN_{\mathrm{va}}}\left\|\bar{\mathbf{X}}^\top\bar{\mathbf{X}} - \tilde{\mathbf{X}}^\top\tilde{\mathbf{X}}\right\|\|\theta_0\|^2
\end{aligned}
\tag{54}
$$

where $(a)$ follows from (50), and $(b)$ follows from (53).

Thus, due to Lemma 16, there is an absolute constant $c$ such that for any $1 \leq t \leq MN_{\mathrm{va}}$ with probability at least $1 - e^{-t}$ over $\mathbf{Z}^{\mathrm{va}}$, it holds that

$$\left\|\mathbf{W} - \frac{1}{MN_{\mathrm{va}}}\bar{\mathbf{X}}^\top\bar{\mathbf{X}}\right\|\|\theta_0\|^2 \leq c\|\theta_0\|^2\|\mathbf{W}\|\max\left\{\sqrt{\frac{r(\mathbf{W})}{MN_{\mathrm{va}}}}, \frac{r(\mathbf{W})}{MN_{\mathrm{va}}}, \sqrt{\frac{t}{MN_{\mathrm{va}}}}\right\} \tag{55}$$

where $r(\mathbf{W})$ is defined as

$$r(\mathbf{W}) := \frac{(\mathbb{E}\|\bar{\mathbf{x}}\|)^2}{\|\mathbf{W}\|} \leq \frac{\mathbb{E}\left(\|\bar{\mathbf{x}}\|^2\right)}{\|\mathbf{W}\|} = \frac{\mathrm{Tr}(\mathbf{W})}{\|\mathbf{W}\|} = r_0(\mathbf{W}). \tag{56}$$

The bound on $\left\|\bar{\mathbf{X}}^\top\bar{\mathbf{X}} - \tilde{\mathbf{X}}^\top\tilde{\mathbf{X}}\right\|$ can be found in Lemma 17, which shows when $|\alpha| < \min_m \min\{1/\lambda_{m1}, 1/\mu_1(\mathbf{\Lambda}_m^{\frac{1}{2}}\hat{\mathbf{D}}_m^{\mathrm{tr}}\mathbf{\Lambda}_m^{\frac{1}{2}})\}$, with probability at least $1 - 2Me^{-t}$ over $\mathbf{Z}^{\mathrm{tr}}$ and $\mathbf{Z}^{\mathrm{va}}$ for any $1 \leq t \leq N_{\mathrm{va}}$, it holds that

$$\frac{1}{MN_{\mathrm{va}}}\left\|\bar{\mathbf{X}}^\top\bar{\mathbf{X}} - \tilde{\mathbf{X}}^\top\tilde{\mathbf{X}}\right\| \leq \frac{c|\alpha|}{M}\sum_{m=1}^M \lambda_{m1}^2\max\left\{\sqrt{\frac{r(\mathbf{W}_m)}{N_{\mathrm{tr}}}}, \frac{r(\mathbf{W}_m)}{N_{\mathrm{tr}}}, \sqrt{\frac{t}{N_{\mathrm{tr}}}}, \frac{t}{N_{\mathrm{tr}}}\right\}. \tag{57}$$

Applying the union bound we have for MAML with $|\alpha| < \min_m \min\{1/\lambda_{m1}, 1/\mu_1(\mathbf{\Lambda}_m^{\frac{1}{2}}\hat{\mathbf{D}}_m^{\mathrm{tr}}\mathbf{\Lambda}_m^{\frac{1}{2}})\}$ and for iMAML with $\gamma > 0$, for any $1 \leq t \leq N_{\mathrm{va}}$, with probability at least $1 - (2M+1)e^{-t}$ over $\mathbf{Z}^{\mathrm{tr}}$ and $\mathbf{Z}^{\mathrm{va}}$, there exists $c > 1$ that

$$\theta_0^\top \mathbf{B}\theta_0 \lesssim \|\theta_0\|^2\|\mathbf{W}\|\max\left\{\sqrt{\frac{r(\mathbf{W})}{MN_{\mathrm{va}}}}, \frac{r(\mathbf{W})}{MN_{\mathrm{va}}}, \sqrt{\frac{t}{MN_{\mathrm{va}}}}\right\}. \tag{58}$$

The proof is complete. $\qquad\square$

## C.4 Proof of Lemma 3

To prove Lemma 3, we need to bound $\mathrm{Tr}(\mathbf{C}) = \mathrm{Tr}(\mathbf{C}_1) + \mathrm{Tr}(\mathbf{C}_2)$. We first show in Lemma 5 that $\mathrm{Tr}(\mathbf{C}_2)$ can be bounded as $\Theta(\mathrm{Tr}(\mathbf{C}_1))$. Then the key step is to bound $\mathrm{Tr}(\mathbf{C}_1)$. To bound $\mathrm{Tr}(\mathbf{C}_1)$, first we show in Lemma 7 that $\mathrm{Tr}(\mathbf{C}_1)$ can be decomposed into terms that are related to the first $k$ largest eigenvalues of $\mathbf{W}$ and the term that is only related to the rest eigenvalues. Next we bound the term related to the $d - k$ smallest eigenvalues of, as a function of $\mu_n(\mathbf{A})$, given in Lemma 8. And then we bound the term related to the $k$ largest eigenvalues, given in Lemma 9. Finally, we bound the eigenvalues of $\mu_n(\mathbf{A})$ in Lemma 10.

**Lemma 5** (Bound on $\mathrm{Tr}(\mathbf{C}_2^{\mathcal{A}})$ in terms of $\mathrm{Tr}(\mathbf{C}_1^{\mathcal{A}})$)**.** *Recall $\alpha$ is the step size for MAML, $\gamma$ is the regularization parameter for iMAML, and*

$$\mathrm{Tr}(\mathbf{C}_2^{\mathrm{ma}}) = \frac{\alpha^2}{N_{\mathrm{tr}}}\mathrm{Tr}\Big(\mathbf{C}_1^{\mathrm{ma}}\mathrm{diag}[\mathbf{X}_m^{\mathrm{va}}\hat{\mathbf{Q}}_m^{\mathrm{tr}}\mathbf{X}_m^{\mathrm{va}\top}]\Big), \tag{59}$$

$$\mathrm{Tr}(\mathbf{C}_2^{\mathrm{im}}) = \frac{1}{N_{\mathrm{tr}}}\mathrm{Tr}\Big(\mathbf{C}_1^{\mathrm{im}}\mathrm{diag}[\mathbf{X}_m^{\mathrm{va}}(\mathbf{I}+\gamma^{-1}\hat{\mathbf{Q}}_m^{\mathrm{tr}})^{-1}\hat{\mathbf{Q}}_m^{\mathrm{tr}}(\mathbf{I}+\gamma^{-1}\hat{\mathbf{Q}}_m^{\mathrm{tr}})^{-1}\mathbf{X}_m^{\mathrm{va}\top}]\Big). \tag{60}$$

*Let $c > c_\lambda + \max_m \lambda_{m1}(1 + c_{\sigma_x}t + \sqrt{c_\lambda/\lambda_{m1}})$, it holds with probability at least $1 - 2Me^{-t}$ that*

$$\mathrm{Tr}(\mathbf{C}_2^{\mathrm{ma}}) \leq \mathrm{Tr}(\mathbf{C}_1^{\mathrm{ma}})c^2\alpha^2\frac{N_{\mathrm{va}}}{N_{\mathrm{tr}}}, \quad \textit{and} \quad \mathrm{Tr}(\mathbf{C}_2^{\mathrm{im}}) \leq \mathrm{Tr}(\mathbf{C}_1^{\mathrm{im}})c^2\frac{N_{\mathrm{va}}}{N_{\mathrm{tr}}}. \tag{61}$$

*Proof.* We can derive $\mathrm{Tr}(\mathbf{C}_2^{\mathrm{ma}})$ by

$$\mathrm{Tr}(\mathbf{C}_2^{\mathrm{ma}}) = \frac{\alpha^2}{N_{\mathrm{tr}}}\mathrm{Tr}(\mathbf{C}_1\mathrm{diag}[\mathbf{X}_m^{\mathrm{va}}\hat{\mathbf{Q}}_m^{\mathrm{tr}}\mathbf{X}_m^{\mathrm{va}\top}]) \overset{(a)}{\leq} \frac{\alpha^2}{N_{\mathrm{tr}}}\mathrm{Tr}(\mathbf{C}_1^{\mathrm{ma}})\big\|\mathrm{diag}[\mathbf{X}_m^{\mathrm{va}}\hat{\mathbf{Q}}_m^{\mathrm{tr}}\mathbf{X}_m^{\mathrm{va}\top}]\big\|$$

$$\overset{(b)}{=} \frac{\alpha^2}{N_{\mathrm{tr}}}\mathrm{Tr}(\mathbf{C}_1^{\mathrm{ma}})\max_m\big\|\mathbf{X}_m^{\mathrm{va}}\hat{\mathbf{Q}}_m^{\mathrm{tr}}\mathbf{X}_m^{\mathrm{va}\top}\big\| \overset{(c)}{\leq} \frac{\alpha^2}{N_{\mathrm{tr}}}\mathrm{Tr}(\mathbf{C}_1^{\mathrm{ma}})\max_m\big\|\hat{\mathbf{Q}}_m^{\mathrm{tr}}\big\|\big\|\mathbf{X}_m^{\mathrm{va}}\mathbf{X}_m^{\mathrm{va}\top}\big\| \tag{62}$$

where $(a)$ follows from Lemma 13, $(b)$ follows because the largest eigenvalue of a symmetric block diagonal matrix is the maximum largest eigenvalue of the block matrices, $(c)$ follows because for any unit vector $\mathbf{u}$, $\mathbf{u}^\top\mathbf{X}_m^{\mathrm{va}}\hat{\mathbf{Q}}_m^{\mathrm{tr}}\mathbf{X}_m^{\mathrm{va}\top}\mathbf{u} \leq \big\|\hat{\mathbf{Q}}_m^{\mathrm{tr}}\big\|\mathbf{u}^\top\mathbf{X}_m^{\mathrm{va}}\mathbf{X}_m^{\mathrm{va}\top}\mathbf{u} \leq \big\|\hat{\mathbf{Q}}_m^{\mathrm{tr}}\big\|\big\|\mathbf{X}_m^{\mathrm{va}}\mathbf{X}_m^{\mathrm{va}\top}\big\|$.

Then because $\big\|\mathbf{X}_m^{\mathrm{va}}\mathbf{X}_m^{\mathrm{va}\top}\big\| = \big\|\mathbf{X}_m^{\mathrm{va}\top}\mathbf{X}_m^{\mathrm{va}}\big\| = N_{\mathrm{va}}\big\|\hat{\mathbf{Q}}_m^{\mathrm{va}}\big\|$. The bound on $\big\|\hat{\mathbf{Q}}_m^{\mathrm{tr}}\big\|$ and $\big\|\hat{\mathbf{Q}}_m^{\mathrm{va}}\big\|$ can be obtained by Lemma 19. Applying the union bound over $\mathbf{Z}^{\mathrm{tr}}$ and $\mathbf{Z}^{\mathrm{va}}$, we have that there exists a constant $c > 0$ that depends on $\sigma_x$ such that, for all $t \geq 1$, with probability at least $1 - 2e^{-t}$

$$\big\|\hat{\mathbf{Q}}_m^{\mathrm{tr}}\big\| \leq \lambda_{m1} + c\lambda_{m1}\max\left\{\sqrt{\frac{r(\mathbf{Q}_m)}{N_{\mathrm{tr}}}}, \frac{r(\mathbf{Q}_m)}{N_{\mathrm{tr}}}, \sqrt{\frac{t}{N_{\mathrm{tr}}}}, \frac{t}{N_{\mathrm{tr}}}\right\},$$

$$\text{and } \big\|\hat{\mathbf{Q}}_m^{\mathrm{va}}\big\| \leq \lambda_{m1} + c\lambda_{m1}\max\left\{\sqrt{\frac{r(\mathbf{Q}_m)}{N_{\mathrm{va}}}}, \frac{r(\mathbf{Q}_m)}{N_{\mathrm{va}}}, \sqrt{\frac{t}{N_{\mathrm{va}}}}, \frac{t}{N_{\mathrm{va}}}\right\}.$$

Then applying the union bound over $M$ tasks, we have that there exists a constant $c_{\sigma_x} > 0$ that depends on $\sigma_x$, and $c > c_\lambda + \max_m \lambda_{m1}(1 + c_{\sigma_x}t + \sqrt{c_\lambda/\lambda_{m1}})$ such that, for all $t \geq 1$, with probability at least $1 - 2Me^{-t}$

$$\max_m\big\|\hat{\mathbf{Q}}_m^{\mathrm{tr}}\big\|\big\|\mathbf{X}_m^{\mathrm{va}}\mathbf{X}_m^{\mathrm{va}\top}\big\| \leq c^2N_{\mathrm{va}}. \tag{63}$$

Combining the above results with (62) completes the proof for MAML.

Similarly, for iMAML, we have

$$\mathrm{Tr}(\mathbf{C}_2^{\mathrm{im}}) = \frac{1}{N_{\mathrm{tr}}}\mathrm{Tr}\Big(\mathbf{C}_1^{\mathrm{im}}\mathrm{diag}[\mathbf{X}_m^{\mathrm{va}}(\mathbf{I}+\gamma^{-1}\hat{\mathbf{Q}}_m^{\mathrm{tr}})^{-1}\hat{\mathbf{Q}}_m^{\mathrm{tr}}(\mathbf{I}+\gamma^{-1}\hat{\mathbf{Q}}_m^{\mathrm{tr}})^{-1}\mathbf{X}_m^{\mathrm{va}\top}]\Big)$$

$$\leq \frac{1}{N_{\mathrm{tr}}}\mathrm{Tr}(\mathbf{C}_1^{\mathrm{im}})\big\|\mathrm{diag}[\mathbf{X}_m^{\mathrm{va}}(\mathbf{I}+\gamma^{-1}\hat{\mathbf{Q}}_m^{\mathrm{tr}})^{-1}\hat{\mathbf{Q}}_m^{\mathrm{tr}}(\mathbf{I}+\gamma^{-1}\hat{\mathbf{Q}}_m^{\mathrm{tr}})^{-1}\mathbf{X}_m^{\mathrm{va}\top}]\big\|$$

$$= \frac{1}{N_{\mathrm{tr}}}\mathrm{Tr}(\mathbf{C}_1^{\mathrm{im}})\max_m\big\|\mathbf{X}_m^{\mathrm{va}}(\mathbf{I}+\gamma^{-1}\hat{\mathbf{Q}}_m^{\mathrm{tr}})^{-1}\hat{\mathbf{Q}}_m^{\mathrm{tr}}(\mathbf{I}+\gamma^{-1}\hat{\mathbf{Q}}_m^{\mathrm{tr}})^{-1}\mathbf{X}_m^{\mathrm{va}\top}\big\|$$

$$\leq \frac{1}{N_{\mathrm{tr}}}\mathrm{Tr}(\mathbf{C}_1^{\mathrm{im}})\max_m\big\|(\mathbf{I}+\gamma^{-1}\hat{\mathbf{Q}}_m^{\mathrm{tr}})^{-1}\hat{\mathbf{Q}}_m^{\mathrm{tr}}(\mathbf{I}+\gamma^{-1}\hat{\mathbf{Q}}_m^{\mathrm{tr}})^{-1}\big\|\big\|\mathbf{X}_m^{\mathrm{va}}\mathbf{X}_m^{\mathrm{va}\top}\big\|$$

$$\leq \frac{1}{N_{\mathrm{tr}}}\mathrm{Tr}(\mathbf{C}_1^{\mathrm{im}})\max_m\big\|\hat{\mathbf{Q}}_m^{\mathrm{tr}}\big\|\big\|\mathbf{X}_m^{\mathrm{va}}\mathbf{X}_m^{\mathrm{va}\top}\big\|. \tag{64}$$

Combining the above results with (63) on the same high probability event for $\mathbf{Z}$ completes the proof for iMAML. $\qquad\square$

Lemma 5 shows that $\mathrm{Tr}(\mathbf{C}_2)$ can be bounded as $\boldsymbol{\Theta}(\mathrm{Tr}(\mathbf{C}_1))$. Then we proceed to bound $\mathrm{Tr}(\mathbf{C}_1)$. In Lemma 7, we decompose $\mathrm{Tr}(\mathbf{C}_1)$ into terms that are related to the first $k$ largest eigenvalues of $\mathbf{W}$ and the term that is only related to the rest eigenvalues of $\mathbf{W}$.

**Lemma 6** (Bound of $\mathrm{Tr}(\mathbf{C}_1)$ in terms of $\bar{\mathbf{X}}$). *Recall* $\mathrm{Tr}(\mathbf{C}_1)$ *and* $\bar{\mathbf{X}}$ *is computed by*

$$\mathrm{Tr}(\mathbf{C}_1) = \mathrm{Tr}(\tilde{\mathbf{X}}\mathbf{W}\tilde{\mathbf{X}}^\top \mathbf{A}^{-2}), \ \ and \ \ \bar{\mathbf{X}} = [\mathbf{Z}_m^{\mathrm{va}}\bar{\boldsymbol{\Lambda}}_m \mathbf{V}_m^\top]_m$$

*Then we have with high probability*

$$\mathrm{Tr}(\mathbf{C}_1) \leq c\mathrm{Tr}(\bar{\mathbf{X}}\mathbf{W}\bar{\mathbf{X}}^\top \mathbf{A}^{-2}).$$

*Proof.* By Lemma 13 and the properties of trace, we have

$$\begin{aligned}
\mathrm{Tr}(\mathbf{C}_1) =& \mathrm{Tr}(\bar{\mathbf{X}}\mathbf{W}\bar{\mathbf{X}}^\top \mathbf{A}^{-2}) + \mathrm{Tr}(\mathbf{W}(\tilde{\mathbf{X}} - \bar{\mathbf{X}})^\top \mathbf{A}^{-2}(\tilde{\mathbf{X}} + \bar{\mathbf{X}})) \\
\leq& \mathrm{Tr}(\mathbf{A}^{-2}\bar{\mathbf{X}}\mathbf{W}\bar{\mathbf{X}}^\top) + \mathrm{Tr}(\mathbf{W})\|(\tilde{\mathbf{X}} - \bar{\mathbf{X}})^\top \mathbf{A}^{-2}(\tilde{\mathbf{X}} + \bar{\mathbf{X}})\| \\
\leq& \mathrm{Tr}(\mathbf{A}^{-2}\bar{\mathbf{X}}\mathbf{W}\bar{\mathbf{X}}^\top) + \mathrm{Tr}(\mathbf{W})\mu_n^{-2}(\mathbf{A})\|\tilde{\mathbf{X}} - \bar{\mathbf{X}}\|\|\tilde{\mathbf{X}} + \bar{\mathbf{X}}\| \\
\leq& \mathrm{Tr}(\mathbf{A}^{-2}\bar{\mathbf{X}}\mathbf{W}\bar{\mathbf{X}}^\top) + \mathrm{Tr}(\mathbf{W})\mu_n^{-2}(\mathbf{A})\|\tilde{\mathbf{X}} - \bar{\mathbf{X}}\|(2\|\bar{\mathbf{X}}\| + \|\tilde{\mathbf{X}} - \bar{\mathbf{X}}\|).
\end{aligned}$$

where $\|\tilde{\mathbf{X}} - \bar{\mathbf{X}}\|$ is bounded by Lemma 18 and $\|\bar{\mathbf{X}}\|$ is bounded by Lemma 19, which can be controlled by choosing proper hyperparameters $\gamma$ and $\alpha$ to make the first term dominate. $\qquad\square$

**Lemma 7** (Decomposition of $\mathrm{Tr}(\bar{\mathbf{X}}\mathbf{W}\bar{\mathbf{X}}^\top \mathbf{A}^{-2})$ in $\mathrm{Tr}(\mathbf{C}_1)$). *Recall* $\tilde{\mathbf{X}} = [\mathbf{Z}_m^{\mathrm{va}}\tilde{\boldsymbol{\Lambda}}_m \mathbf{P}_m]$, $\bar{\mathbf{X}} = [\mathbf{Z}_m^{\mathrm{va}}\bar{\boldsymbol{\Lambda}}_m \mathbf{V}_m]$, $\bar{\mathbf{X}}_{\mathrm{P}} = [\mathbf{Z}_m^{\mathrm{va}}\bar{\boldsymbol{\Lambda}}_m \mathbf{P}_m]$. *Define* $\mathbf{A} = \tilde{\mathbf{X}}\tilde{\mathbf{X}}^\top$, *and* $\mathbf{X}_{\mathrm{P}} = [\mathbf{Z}_m^{\mathrm{va}}\bar{\boldsymbol{\Lambda}}_m \mathbf{P}_m]$. *For both MAML and iMAML,* $\mathrm{Tr}(\bar{\mathbf{X}}\mathbf{W}\bar{\mathbf{X}}^\top \mathbf{A}^{-2})$ *in* $\mathrm{Tr}(\mathbf{C}_1)$ *can be bounded by*

$$\mathrm{Tr}(\bar{\mathbf{X}}\mathbf{W}\bar{\mathbf{X}}^\top \mathbf{A}^{-2}) \leq c\mathrm{Tr}\Big((\bar{\mathbf{X}}_{\mathrm{P}}\boldsymbol{\Lambda}_{W,0:k}\bar{\mathbf{X}}_{\mathrm{P}}^\top + \bar{\mathbf{X}}\mathbf{V}_{W,k:d}\boldsymbol{\Lambda}_{W,k:d}\mathbf{V}_{W,k:d}^\top \bar{\mathbf{X}}^\top)\mathbf{A}^{-2}\Big).$$

*Proof.* Recall the singular value decomposition of $\mathbf{W}$ as $\mathbf{W} = \mathbf{V}_W \boldsymbol{\Lambda}_W \mathbf{V}_W^\top$, then for any $0 \leq k \leq d$, $\mathbf{W}$ can be computed by

$$\mathbf{W} = \mathbf{V}_{W,0:k}\boldsymbol{\Lambda}_{W,0:k}\mathbf{V}_{W,0:k}^\top + \mathbf{V}_{W,k:d}\boldsymbol{\Lambda}_{W,k:d}\mathbf{V}_{W,k:d}^\top. \tag{65}$$

Therefore we have

$$\mathrm{Tr}(\bar{\mathbf{X}}\mathbf{W}\bar{\mathbf{X}}^\top \mathbf{A}^{-2}) = \mathrm{Tr}\Big((\bar{\mathbf{X}}\mathbf{V}_{W,0:k}\boldsymbol{\Lambda}_{W,0:k}\mathbf{V}_{W,0:k}^\top \bar{\mathbf{X}}^\top + \bar{\mathbf{X}}\mathbf{V}_{W,k:d}\boldsymbol{\Lambda}_{W,k:d}\mathbf{V}_{W,k:d}^\top \bar{\mathbf{X}}^\top)\mathbf{A}^{-2}\Big)$$

where $\bar{\mathbf{X}}\mathbf{V}_{W,0:k}\boldsymbol{\Lambda}_{W,0:k}\mathbf{V}_{W,0:k}^\top \bar{\mathbf{X}}^\top$ can be further decomposed by

$$\begin{aligned}
\bar{\mathbf{X}}\mathbf{V}_{W,0:k}\boldsymbol{\Lambda}_{W,0:k}\mathbf{V}_{W,0:k}^\top &= \bar{\mathbf{X}}_{\mathrm{P}}\boldsymbol{\Lambda}_{W,0:k}\bar{\mathbf{X}}_{\mathrm{P}}^\top \\
&+ [\mathbf{Z}_m^{\mathrm{va}}\bar{\boldsymbol{\Lambda}}_m(\mathbf{V}_m\mathbf{V}_{W,0:k} - \mathbf{P}_{m,0:k})]\boldsymbol{\Lambda}_{W,0:k}[\mathbf{Z}_m^{\mathrm{va}}\bar{\boldsymbol{\Lambda}}_m(\mathbf{V}_m\mathbf{V}_{W,0:k} - \mathbf{P}_{m,0:k})]^\top.
\end{aligned}$$

By Lemma 13, we have the last term can be bounded by

$$\begin{aligned}
&\mathrm{Tr}(\mathbf{A}^{-2}[\mathbf{Z}_m^{\mathrm{va}}\bar{\boldsymbol{\Lambda}}_m(\mathbf{V}_m^\top \mathbf{V}_{W,0:k} + \mathbf{P}_{m,0:k})]\boldsymbol{\Lambda}_{W,0:k}[\mathbf{Z}_m^{\mathrm{va}}\bar{\boldsymbol{\Lambda}}_m(\mathbf{V}_m^\top \mathbf{V}_{W,0:k} - \mathbf{P}_{m,0:k})]^\top) \\
\leq& \mathrm{Tr}(\boldsymbol{\Lambda}_{W,0:k})\mu_n(\mathbf{A})^{-2}\|[\mathbf{Z}_m^{\mathrm{va}}\bar{\boldsymbol{\Lambda}}_m(\mathbf{V}_m^\top \mathbf{V}_{W,0:k} + \mathbf{P}_{m,0:k})]\|\|[\mathbf{Z}_m^{\mathrm{va}}\bar{\boldsymbol{\Lambda}}_m(\mathbf{V}_m^\top \mathbf{V}_{W,0:k} - \mathbf{P}_{m,0:k})]^\top\|
\end{aligned}$$

where $\|[\mathbf{Z}_m^{\mathrm{va}}\bar{\boldsymbol{\Lambda}}_m(\mathbf{V}_m^\top \mathbf{V}_{W,0:k} + \mathbf{P}_{m,0:k})]\|$ can be further bounded with high probability by

$$\begin{aligned}
&\|[\mathbf{Z}_m^{\mathrm{va}}\bar{\boldsymbol{\Lambda}}_m(\mathbf{V}_m^\top \mathbf{V}_{W,0:k} + \mathbf{P}_{m,0:k})]\| \\
=& \|[\mathbf{Z}_m^{\mathrm{va}}\bar{\boldsymbol{\Lambda}}_m(\mathbf{V}_m^\top \mathbf{V}_{W,0:k} + \mathbf{P}_{m,0:k})]^\top [\mathbf{Z}_m^{\mathrm{va}}\bar{\boldsymbol{\Lambda}}_m(\mathbf{V}_m^\top \mathbf{V}_{W,0:k} + \mathbf{P}_{m,0:k})]\|^{\frac{1}{2}} \\
=& \Big\|\sum_{m=1}^M (\mathbf{V}_m^\top \mathbf{V}_{W,0:k} + \mathbf{P}_{m,0:k})^\top \bar{\boldsymbol{\Lambda}}_m^\top \mathbf{Z}_m^{\mathrm{va}\top}\mathbf{Z}_m^{\mathrm{va}}\bar{\boldsymbol{\Lambda}}_m(\mathbf{V}_m^\top \mathbf{V}_{W,0:k} + \mathbf{P}_{m,0:k})\Big\|^{\frac{1}{2}} \\
\lesssim& \sqrt{N_{\mathrm{va}}}\Big(\sum_{m=1}^M \mathrm{Tr}(\mathbf{W}_m)\Big)^{\frac{1}{2}}
\end{aligned}$$

where the last inequality follows from Lemma 16.

Similarly, $\|[\mathbf{Z}_m^{\mathrm{va}}\bar{\boldsymbol{\Lambda}}_m(\mathbf{V}_m^\top\mathbf{V}_{W,0:k} - \mathbf{P}_{m,0:k})]\|$ can be further bounded with high probability by

$$
\begin{aligned}
&\|[\mathbf{Z}_m^{\mathrm{va}}\bar{\boldsymbol{\Lambda}}_m(\mathbf{V}_m^\top\mathbf{V}_{W,0:k} - \mathbf{P}_{m,0:k})]\| \\
=&\left\|[\mathbf{Z}_m^{\mathrm{va}}\bar{\boldsymbol{\Lambda}}_m(\mathbf{V}_m^\top\mathbf{V}_{W,0:k} - \mathbf{P}_{m,0:k})]^\top[\mathbf{Z}_m^{\mathrm{va}}\bar{\boldsymbol{\Lambda}}_m(\mathbf{V}_m^\top\mathbf{V}_{W,0:k} - \mathbf{P}_{m,0:k})]\right\|^{\frac{1}{2}} \\
=&\left\|\sum_{m=1}^M (\mathbf{V}_m^\top\mathbf{V}_{W,0:k} - \mathbf{P}_{m,0:k})^\top\bar{\boldsymbol{\Lambda}}_m^\top\mathbf{Z}_m^{\mathrm{va}\top}\mathbf{Z}_m^{\mathrm{va}}\bar{\boldsymbol{\Lambda}}_m(\mathbf{V}_m^\top\mathbf{V}_{W,0:k} - \mathbf{P}_{m,0:k})\right\|^{\frac{1}{2}} \\
\lesssim&\sqrt{MN_{\mathrm{va}}}\max_m \mathrm{Tr}^{\frac{1}{2}}(\mathbf{W}_m)\|\mathbf{V}_m^\top\mathbf{V}_{W,0:k} - \mathbf{P}_{m,0:k}\|.
\end{aligned}
$$

Based on the assumption the last term is smaller compared to the rest terms. $\qquad\square$

Then we bound the term related to the $d - k$ smallest eigenvalues of $\mathbf{W}$ as a function of $\mu_n(\mathbf{A})$, given in Lemma 8.

**Lemma 8** (Bound on $\mathrm{Tr}(\bar{\mathbf{X}}\mathbf{V}_{W,k:d}\boldsymbol{\Lambda}_{W,k:d}\mathbf{V}_{W,k:d}^\top\bar{\mathbf{X}}^\top\mathbf{A}^{-2})$ in $\mathrm{Tr}(\mathbf{C}_1)$). *With probability at least $1 - e^{-t}$ over $\mathbf{Z}$, and for $c \geq t$, it holds that*

$$
\mathrm{Tr}(\bar{\mathbf{X}}\mathbf{V}_{W,k:d}\boldsymbol{\Lambda}_{W,k:d}\mathbf{V}_{W,k:d}^\top\bar{\mathbf{X}}^\top\mathbf{A}^{-2}) \leq cMN_{\mathrm{va}}\mu_n^{-2}(\mathbf{A})\sum_{i>k}\mu_i^2(\mathbf{W})
$$

*where $\mu_n$ is the smallest eigenvalue of a matrix.*

*Proof.* By Von Neumann's trace inequality in Lemma 13, $\mathrm{Tr}(\bar{\mathbf{X}}\mathbf{V}_{W,k:d}\boldsymbol{\Lambda}_{W,k:d}\mathbf{V}_{W,k:d}^\top\bar{\mathbf{X}}^\top\mathbf{A}^{-2})$ is bounded by

$$
\mathrm{Tr}(\bar{\mathbf{X}}\mathbf{V}_{W,k:d}\boldsymbol{\Lambda}_{W,k:d}\mathbf{V}_{W,k:d}^\top\bar{\mathbf{X}}^\top\mathbf{A}^{-2}) \leq \mathrm{Tr}(\mathbf{V}_{W,k:d}\boldsymbol{\Lambda}_{W,k:d}\mathbf{V}_{W,k:d}^\top\bar{\mathbf{X}}^\top\bar{\mathbf{X}})\mu_n^{-2}(\mathbf{A}).
$$

To bound $\mathrm{Tr}(\mathbf{V}_{W,k:d}\boldsymbol{\Lambda}_{W,k:d}\mathbf{V}_{W,k:d}^\top\bar{\mathbf{X}}^\top\bar{\mathbf{X}})$, we first rewrite it as

$$
\begin{aligned}
&\mathrm{Tr}(\mathbf{V}_{W,k:d}\boldsymbol{\Lambda}_{W,k:d}\mathbf{V}_{W,k:d}^\top\bar{\mathbf{X}}^\top\bar{\mathbf{X}}) = MN_{\mathrm{va}}\mathrm{Tr}\left((\mathbf{V}_{W,k:d}\boldsymbol{\Lambda}_{W,k:d}\mathbf{V}_{W,k:d}^\top)^2\right) \\
&+ \mathrm{Tr}\left(\mathbf{V}_{W,k:d}\boldsymbol{\Lambda}_{W,k:d}\mathbf{V}_{W,k:d}^\top(\bar{\mathbf{X}}^\top\bar{\mathbf{X}} - MN_{\mathrm{va}}\mathbf{V}_{W,k:d}\boldsymbol{\Lambda}_{W,k:d}\mathbf{V}_{W,k:d}^\top)\right) \\
=&MN_{\mathrm{va}}\mathrm{Tr}\left(\boldsymbol{\Lambda}_{W,k:d}^2\right) + \left\|\bar{\mathbf{X}}\mathbf{V}_{W,k:d}\boldsymbol{\Lambda}_{W,k:d}^{\frac{1}{2}}\right\|_{\mathrm{F}}^2 - MN_{\mathrm{va}}\left\|\mathbf{V}_{W,k:d}\boldsymbol{\Lambda}_{W,k:d}\mathbf{V}_{W,k:d}^\top\right\|_{\mathrm{F}}^2 \\
=&MN_{\mathrm{va}}\left(\sum_{i>k}\mu_i^2(\mathbf{W})\right) + \underbrace{\left\|\bar{\mathbf{X}}\mathbf{V}_{W,k:d}\boldsymbol{\Lambda}_{W,k:d}^{\frac{1}{2}}\right\|_{\mathrm{F}}^2 - \mathbb{E}\left[\left\|\bar{\mathbf{X}}\mathbf{V}_{W,k:d}\boldsymbol{\Lambda}_{W,k:d}^{\frac{1}{2}}\right\|_{\mathrm{F}}^2\right]}_{I_1} \quad (66)
\end{aligned}
$$

where the last equation follows because

$$
\begin{aligned}
&\mathbb{E}\left[\left\|\bar{\mathbf{X}}\mathbf{V}_{W,k:d}\boldsymbol{\Lambda}_{W,k:d}^{\frac{1}{2}}\right\|_{\mathrm{F}}^2\right] = \mathbb{E}\left[\mathrm{Tr}(\mathbf{V}_{W,k:d}\boldsymbol{\Lambda}_{W,k:d}\mathbf{V}_{W,k:d}^\top\bar{\mathbf{X}}^\top\bar{\mathbf{X}})\right] \\
=&MN_{\mathrm{va}}\mathbb{E}\left[\mathrm{Tr}(\mathbf{V}_{W,k:d}\boldsymbol{\Lambda}_{W,k:d}\mathbf{V}_{W,k:d}^\top\mathbf{W})\right] = MN_{\mathrm{va}}\mathbb{E}\left[\mathrm{Tr}\left((\mathbf{V}_{W,k:d}\boldsymbol{\Lambda}_{W,k:d}\mathbf{V}_{W,k:d}^\top)^2\right)\right] \\
=&MN_{\mathrm{va}}\left\|\mathbf{V}_{W,k:d}\boldsymbol{\Lambda}_{W,k:d}\mathbf{V}_{W,k:d}^\top\right\|_{\mathrm{F}}^2.
\end{aligned}
$$

Let $\bar{\mathbf{x}}_{m,n}$ be the $n$-th row of $\bar{\mathbf{X}}_m$, $I_1$ can be further bounded with probability at least $1 - e^{-t}$ by

$$
\begin{aligned}
|I_1| =&MN_{\mathrm{va}}\left|\frac{1}{MN_{\mathrm{va}}}\sum_{m=1}^M\sum_{n=1}^{N_{\mathrm{va}}}\left\|\bar{\mathbf{x}}_{m,n}\mathbf{V}_{W,k:d}\boldsymbol{\Lambda}_{W,k:d}^{\frac{1}{2}}\right\|_{\mathrm{F}}^2 - \mathbb{E}\left[\left\|\bar{\mathbf{x}}_{m,n}\mathbf{V}_{W,k:d}\boldsymbol{\Lambda}_{W,k:d}^{\frac{1}{2}}\right\|_{\mathrm{F}}^2\right]\right| \\
\leq&MN_{\mathrm{va}}\left\|\mathbf{V}_{W,k:d}\boldsymbol{\Lambda}_{W,k:d}\mathbf{V}_{W,k:d}^\top\right\|_{\mathrm{F}}^2\max\left\{\sqrt{\frac{t}{MN_{\mathrm{va}}}}, \frac{t}{MN_{\mathrm{va}}}\right\}
\end{aligned}
$$

where the last inequality follows because $\|\bar{\mathbf{x}}_{m,n}\mathbf{V}_{W,k:d}\boldsymbol{\Lambda}_{W,k:d}^{\frac{1}{2}}\|_{\mathrm{F}}^2$ are sub-exponential for $m \in [M], n \in [N_{\mathrm{va}}]$.

Also because $\|\mathbf{V}_{W,k:d}\boldsymbol{\Lambda}_{W,k:d}\mathbf{V}_{W,k:d}^\top\|_{\mathrm{F}}^2 = \mathrm{Tr}(\boldsymbol{\Lambda}_{W,k:d}^2) = \sum_{i>k}\mu_i^2(\mathbf{W})$, we have with probability at least $1 - e^{-t}$

$$\mathrm{Tr}(\bar{\mathbf{X}}\mathbf{V}_{W,k:d}\boldsymbol{\Lambda}_{W,k:d}\mathbf{V}_{W,k:d}^\top\bar{\mathbf{X}}^\top\mathbf{A}^{-2}) \le cMN_{\mathrm{va}}\mu_n^{-2}(\mathbf{A})\sum_{i>k}\mu_i^2(\mathbf{W}).$$

This completes the proof. $\qquad\square$

Next we bound the term related to the $k$ largest eigenvalues of $\mathbf{W}$, given in Lemma 9.

**Lemma 9** (Bound on terms in $\mathrm{Tr}(\mathbf{C}_1)$ related to the first $k$ eigenvalues). *Recall*

$$\bar{\mathbf{X}}_{\mathrm{P}} = [\mathbf{Z}_m^{\mathrm{va}}\bar{\boldsymbol{\Lambda}}_m\mathbf{P}_m]_m, \quad \bar{\mathbf{X}}_{\mathrm{P},0:k} := [\mathbf{Z}_m^{\mathrm{va}}\bar{\boldsymbol{\Lambda}}_m\mathbf{P}_{m,0:k}]_m$$

*There exists $c$ with $0 \le k \le c$ such that with probability at least $1 - 2e^{MN_{\mathrm{va}}/c}$, the following holds*

$$\mathrm{Tr}\big(\bar{\mathbf{X}}_{\mathrm{P},0:k}\boldsymbol{\Lambda}_{W,0:k}\bar{\mathbf{X}}_{\mathrm{P},0:k}^\top\mathbf{A}^{-2}\big) \le \frac{ck}{MN_{\mathrm{va}}}.$$

*Proof.* Recall $\bar{\boldsymbol{\Lambda}}_{\mathrm{P},m} = \mathbf{P}_m^\top\bar{\boldsymbol{\Lambda}}_m\mathbf{P}_m$, $\bar{\mathbf{X}}_{\mathrm{P}}$ and $\bar{\mathbf{X}}_{\mathrm{P},0:k}$ can be written as

$$\bar{\mathbf{X}}_{\mathrm{P}} = [\mathbf{Z}_m^{\mathrm{va}}\mathbf{P}_m\mathbf{P}_m^\top\bar{\boldsymbol{\Lambda}}_m\mathbf{P}_m]_m = [\mathbf{Z}_{\mathrm{P},m}^{\mathrm{va}}\bar{\boldsymbol{\Lambda}}_{\mathrm{P},m}]_m, \qquad \bar{\mathbf{X}}_{\mathrm{P},0:k} = [\mathbf{Z}_{\mathrm{P},m,0:k}^{\mathrm{va}}\bar{\boldsymbol{\Lambda}}_{\mathrm{P},m,0:k}]_m. \qquad (67)$$

Derive $\mathrm{Tr}\big(\bar{\mathbf{X}}_{\mathrm{P},0:k}\boldsymbol{\Lambda}_{W,0:k}\bar{\mathbf{X}}_{\mathrm{P},0:k}^\top\mathbf{A}^{-2}\big)$ as follows

$$\mathrm{Tr}\big(\bar{\mathbf{X}}_{\mathrm{P},0:k}\boldsymbol{\Lambda}_{W,0:k}\bar{\mathbf{X}}_{\mathrm{P},0:k}^\top\mathbf{A}^{-2}\big) = \mathrm{Tr}\big([\mathbf{Z}_{\mathrm{P},m,0:k}^{\mathrm{va}}\bar{\boldsymbol{\Lambda}}_{\mathrm{P},m,0:k}]_m\boldsymbol{\Lambda}_{W,0:k}[\mathbf{Z}_{\mathrm{P},m,0:k}^{\mathrm{va}}\bar{\boldsymbol{\Lambda}}_{\mathrm{P},m,0:k}]_m^\top\mathbf{A}^{-2}\big)$$

$$= \sum_{i=1}^{k}\lambda_{W,i}[\mathbf{z}_{\mathrm{P},m,i}^{\mathrm{va}}\bar{\lambda}_{\mathrm{P},m,i}]_m^\top\mathbf{A}^{-2}[\mathbf{z}_{\mathrm{P},m,i}^{\mathrm{va}}\bar{\lambda}_{\mathrm{P},m,i}]_m = \sum_{i=1}^{k}\lambda_{W,i}\bar{\mathbf{x}}_{\mathrm{P},i}^\top\mathbf{A}^{-2}\bar{\mathbf{x}}_{\mathrm{P},i}$$

Based on Lemma 11, let $\mathbf{A}_{-j} = \mathbf{A} - \bar{\mathbf{x}}_{\mathrm{P},j}\bar{\mathbf{x}}_{\mathrm{P},j}^\top \succ 0$, we have

$$\bar{\mathbf{x}}_{\mathrm{P},j}^\top\mathbf{A}^{-2}\bar{\mathbf{x}}_{\mathrm{P},j} = \bar{\mathbf{x}}_{\mathrm{P},j}^\top(\bar{\mathbf{x}}_{\mathrm{P},j}\bar{\mathbf{x}}_{\mathrm{P},j}^\top + \mathbf{A}_{-j})^{-2}\bar{\mathbf{x}}_{\mathrm{P},j} = \frac{\bar{\mathbf{x}}_{\mathrm{P},j}^\top\mathbf{A}_{-j}^{-2}\bar{\mathbf{x}}_{\mathrm{P},j}}{(1 + \bar{\mathbf{x}}_{\mathrm{P},j}^\top\mathbf{A}_{-j}^{-1}\bar{\mathbf{x}}_{\mathrm{P},j})^2}$$

$$\le \frac{\bar{\mathbf{x}}_{\mathrm{P},j}^\top\mathbf{A}_{-j}^{-2}\bar{\mathbf{x}}_{\mathrm{P},j}}{(\bar{\mathbf{x}}_{\mathrm{P},j}^\top\mathbf{A}_{-j}^{-1}\bar{\mathbf{x}}_{\mathrm{P},j})^2} \le \frac{\mu_n^{-2}(\mathbf{A}_{-j})\|\bar{\mathbf{x}}_{\mathrm{P},j}\|^2}{\mu_{k+1}^{-2}(\mathbf{A}_{-j})\|\Pi_{\mathscr{L}_j}\bar{\mathbf{x}}_{\mathrm{P},j}\|^4}$$

where by Lemma 15, there exists $c_{z1}$ that, with probability at least $1 - 3e^{-t}$, it holds that

$$\|\bar{\mathbf{x}}_{\mathrm{P},j}\|^2 = \sum_{m=1}^{M}\bar{\lambda}_{\mathrm{P},m,i}^2\|\mathbf{z}_{\mathrm{P},m,i}^{\mathrm{va}}\|^2 \le \sum_{m=1}^{M}\bar{\lambda}_{\mathrm{P},m,i}^2\big(N_{\mathrm{va}} + a\sigma_x^2(t + \sqrt{N_{\mathrm{va}}t})\big) \le c_{z1}N_{\mathrm{va}}\sum_{m=1}^{M}\bar{\lambda}_{\mathrm{P},m,i}^2.$$
$$(68)$$

And $\mathscr{L}_j$ is the span of the $MN_{\mathrm{va}} - k$ eigenvectors with the smallest eigenvalues of $\mathbf{A}_{-j}$, and $\Pi_{\mathscr{L}_j}$ represents the projection to $\mathscr{L}_j$. Let $M = \Pi_{\mathscr{L}_j^\perp}^\top\Pi_{\mathscr{L}_j^\perp}$. By Lemma 15, with probability at least $1 - 3e^{-t}$, it holds that

$$\|\Pi_{\mathscr{L}_j^\perp}\bar{\mathbf{x}}_{\mathrm{P},j}\|^2 = \bar{\mathbf{x}}_{\mathrm{P},j}^\top M\bar{\mathbf{x}}_{\mathrm{P},j} \le c_{z1}(2k + 4t)c_P\frac{1}{M}\sum_{m=1}^{M}\bar{\lambda}_{\mathrm{P},m,i}^2.$$

Therefore

$$\|\Pi_{\mathscr{L}_j}\bar{\mathbf{x}}_{\mathrm{P},j}\|^2 = \|\bar{\mathbf{x}}_{\mathrm{P},j}\|^2 - \|\Pi_{\mathscr{L}_j\perp}\bar{\mathbf{x}}_{\mathrm{P},j}\|^2$$

$$\ge c_{z1}(MN_{\mathrm{va}} - (2k + 4t)c_P)\frac{1}{M}\sum_{m=1}^{M}\bar{\lambda}_{\mathrm{P},m,i}^2 \ge (MN_{\mathrm{va}}/c_{z2})\frac{1}{M}\sum_{m=1}^{M}\bar{\lambda}_{\mathrm{P},m,i}^2$$

Since $\mathbf{A}_{-j} = \mathbf{A} - \bar{\mathbf{x}}_{\mathrm{P},j}\bar{\mathbf{x}}_{\mathrm{P},j}^\top \preceq \mathbf{A}$, which, combined with Lemma 12, leads to $\mu_{k+1}(\mathbf{A}_{-j}) < \mu_{k+1}(\mathbf{A}) = \mu_1(\mathbf{A}_k)$.

Since $\mu_n(\mathbf{A}_{-j}) \geq \mu_n(\mathbf{A}_k)$ , we have

$$\bar{\mathbf{x}}_{\mathrm{P},j}^\top \mathbf{A}^{-2} \bar{\mathbf{x}}_{\mathrm{P},j} \leq \frac{\mu_n^{-2}(\mathbf{A}_{-j})\|\bar{\mathbf{x}}_{\mathrm{P},j}\|^2}{\mu_{k+1}^{-2}(\mathbf{A}_{-j})\|\Pi_{\mathscr{L}_j}\bar{\mathbf{x}}_{\mathrm{P},j}\|^4} \overset{(a)}{\leq} c_1 \frac{\mu_n(\mathbf{A}_k)}{\mu_1(\mathbf{A}_k)MN_{\mathrm{va}}} \overset{(b)}{\leq} c_2 \frac{1}{MN_{\mathrm{va}}}$$

where $(a)$ is because $\mu_n(\mathbf{A}_{-j}) \geq \mu_n(\mathbf{A}_k)$ and $\mu_{k+1}(\mathbf{A}_{-j}) < \mu_1(\mathbf{A}_k)$. And $(b)$ is from Lemma 10.

□

Finally in Lemma 10, we bound the eigenvalues of $\mathbf{A}$ to complete the bound on the term related to the $d-k$ smallest eigenvalues of $\mathbf{W}$.

**Lemma 10** (Bound on eigenvalues of $\mathbf{A}$). *Recall that*

$$\mathbf{A} = \tilde{\mathbf{X}}\tilde{\mathbf{X}}^\top = [\mathbf{Z}_{m_1}^{\mathrm{va}}\tilde{\mathbf{\Lambda}}_{m_1}\mathbf{V}_{m_1}^\top\mathbf{V}_{m_2}\tilde{\mathbf{\Lambda}}_{m_2}^\top\mathbf{Z}_{m_2}^{\mathrm{va}\top}]_{m_1 m_2}$$
$$\bar{\mathbf{A}} = \bar{\mathbf{X}}\mathbf{V}_W\mathbf{V}_W^\top\bar{\mathbf{X}}^\top = [\mathbf{Z}_{m_1}^{\mathrm{va}}\bar{\mathbf{\Lambda}}_{m_1}\mathbf{V}_{m_1}^\top\mathbf{V}_W\mathbf{V}_W^\top\mathbf{V}_{m_2}\bar{\mathbf{\Lambda}}_{m_2}\mathbf{Z}_{m_2}^{\mathrm{va}\top}]_{m_1 m_2}$$
$$\bar{\mathbf{A}}_{\mathrm{P}} = \bar{\mathbf{X}}_{\mathrm{P}}\bar{\mathbf{X}}_{\mathrm{P}}^\top = [\mathbf{Z}_{m_1}^{\mathrm{va}}\bar{\mathbf{\Lambda}}_{m_1}\mathbf{P}_{m_1}\mathbf{P}_{m_2}^\top\bar{\mathbf{\Lambda}}_{m_2}\mathbf{Z}_{m_2}^{\mathrm{va}\top}]_{m_1 m_2}.$$

*Let $\mu_i(\cdot)$ denote the $i$-th largest eigenvalue of a matrix, and let $n = MN_{\mathrm{va}}$. Define $\overline{\mathbf{W}}_{\mathrm{P},M} := \frac{1}{M}\sum_{m=1}^M \mathbf{P}_m^\top\bar{\mathbf{\Lambda}}_m^2\mathbf{P}_m$, $\bar{\mathbf{A}}_{\mathrm{P},k} := \bar{\mathbf{X}}_{\mathrm{P},k:d}\bar{\mathbf{X}}_{\mathrm{P},k:d}^\top$, $\overline{\mathbf{W}}_{\mathrm{P},M,k} := \frac{1}{M}\sum_{m=1}^M \mathbf{P}_{m,k:d}^\top\bar{\mathbf{\Lambda}}_m^2\mathbf{P}_{m,k:d}$. Then there exists constants $b,c \geq 1, c_0 \geq 0$ that if $r_0(\overline{\mathbf{W}}_{M,k}) \geq bMN_{\mathrm{va}}$, with probability at least $1 - 2e^{-MN_{\mathrm{va}}/c}$*

$$\mu_n(\mathbf{A}) \geq \mu_n(\bar{\mathbf{A}}) - c_0 \geq \mu_n(\bar{\mathbf{A}}_k) - c_0 \geq \frac{1}{c}\mu_1(\mathbf{W}_k)r_0(\mathbf{W}_k) \tag{69}$$

$$\mu_1(\bar{\mathbf{A}}_k) \leq c\mu_1(\mathbf{W}_k)r_0(\mathbf{W}_k) \tag{70}$$

$$\mu_n(\mathbf{A}) \geq \mu_n(\bar{\mathbf{A}}_{\mathrm{P}}) - 2c_0 \geq \mu_n(\bar{\mathbf{A}}_{\mathrm{P},k}) - 2c_0 \geq \frac{1}{c}\mu_1(\mathbf{W}_k)r_0(\mathbf{W}_k) \tag{71}$$

$$\mu_1(\bar{\mathbf{A}}_{\mathrm{P},k}) \leq \mu_1(\bar{\mathbf{A}}_k) + c_0 \leq c\mu_1(\mathbf{W}_k)r_0(\mathbf{W}_k). \tag{72}$$

*Proof.* First $\mathbf{A}$ can be written as

$$\mathbf{A} = \bar{\mathbf{A}} + \tilde{\mathbf{X}}\tilde{\mathbf{X}}^\top - \bar{\mathbf{X}}\bar{\mathbf{X}}^\top = \bar{\mathbf{A}}_{\mathrm{P}} + \bar{\mathbf{X}}\bar{\mathbf{X}}^\top - \bar{\mathbf{X}}_{\mathrm{P}}\bar{\mathbf{X}}_{\mathrm{P}}^\top + \tilde{\mathbf{X}}\tilde{\mathbf{X}}^\top - \bar{\mathbf{X}}\bar{\mathbf{X}}^\top.$$

Therefore

$$\bar{\mathbf{A}}_{\mathrm{P}} - 2c_0\mathbf{I} \preceq \bar{\mathbf{A}} - c_0\mathbf{I} \preceq \mathbf{A} \preceq \bar{\mathbf{A}} + c_0\mathbf{I} \preceq \bar{\mathbf{A}}_{\mathrm{P}} + 2c_0\mathbf{I}$$

and $c_0 = \max\{\|\tilde{\mathbf{X}}\tilde{\mathbf{X}}^\top - \bar{\mathbf{X}}\bar{\mathbf{X}}^\top\|, \|\bar{\mathbf{X}}\bar{\mathbf{X}}^\top - \bar{\mathbf{X}}_{\mathrm{P}}\bar{\mathbf{X}}_{\mathrm{P}}^\top\|\}$, where $\|\tilde{\mathbf{X}}\tilde{\mathbf{X}}^\top - \bar{\mathbf{X}}\bar{\mathbf{X}}^\top\|$ can be bounded by

$$\|\tilde{\mathbf{X}}\tilde{\mathbf{X}}^\top - \bar{\mathbf{X}}\bar{\mathbf{X}}^\top\| \leq \|(\tilde{\mathbf{X}} + \bar{\mathbf{X}})(\tilde{\mathbf{X}} - \bar{\mathbf{X}})^\top\| \leq \|\tilde{\mathbf{X}} - \bar{\mathbf{X}}\|\big(2\|\bar{\mathbf{X}}\| + \|\tilde{\mathbf{X}} - \bar{\mathbf{X}}\|\big).$$

where $\|\tilde{\mathbf{X}} - \bar{\mathbf{X}}\|$ is bounded by Lemma 18 and $\|\bar{\mathbf{X}}\|$ is bounded by Lemma 19.

For sufficiently small $|\alpha|$ and $\gamma^{-1}$, and $c_1 > 1$, we can control

$$\|\bar{\mathbf{X}}^\top\bar{\mathbf{X}} - \tilde{\mathbf{X}}^\top\tilde{\mathbf{X}}\| \leq \frac{1}{c_1}\mu_1(\mathbf{W}_k)r_0(\mathbf{W}_k) \tag{73}$$

Furthermore, by the bounded task heterogeneity assumption, $\|\bar{\mathbf{X}}\bar{\mathbf{X}}^\top - \bar{\mathbf{X}}_{\mathrm{P}}\bar{\mathbf{X}}_{\mathrm{P}}^\top\|$ can be bounded as

$$\begin{aligned}
\|\bar{\mathbf{X}}\bar{\mathbf{X}}^\top - \bar{\mathbf{X}}_{\mathrm{P}}\bar{\mathbf{X}}_{\mathrm{P}}^\top\| &= \|(\bar{\mathbf{X}} + \bar{\mathbf{X}}_{\mathrm{P}})^\top(\bar{\mathbf{X}} - \bar{\mathbf{X}}_{\mathrm{P}})\| \\
&\leq \sum_{m=1}^M \|\mathbf{I} + \mathbf{P}_m^\top\mathbf{V}_m^\top\mathbf{V}_W\|\|\bar{\mathbf{X}}_m^\top\bar{\mathbf{X}}_m\|\|\mathbf{I} - \mathbf{V}_W^\top\mathbf{V}_m\mathbf{P}_m\| \\
&\leq 2M\max_m \|\mathbf{P}_m^\top - \mathbf{V}_W^\top\mathbf{V}_m\|\|\bar{\mathbf{X}}_m^\top\bar{\mathbf{X}}_m\| \leq \frac{1}{c_1}\mu_1(\mathbf{W}_k)r_0(\mathbf{W}_k).
\end{aligned}$$

Then we have there exists $c_1 > 1$ that

$$c_0 \leq \frac{1}{c_1}\mu_1(\mathbf{W}_k)r_0(\mathbf{W}_k). \tag{74}$$

Next we bound $\|\bar{\mathbf{A}}\|$ and $\|\bar{\mathbf{A}}_{\mathrm{P}}\|$. For $\|\bar{\mathbf{A}}\|$ we have

$$\|\bar{\mathbf{A}}\| = \|\bar{\mathbf{X}}\mathbf{V}_W \mathbf{V}_W^\top \bar{\mathbf{X}}^\top\| = \|\mathbf{V}_W^\top \bar{\mathbf{X}}^\top \bar{\mathbf{X}} \mathbf{V}_W\|$$

$$\leq MN_{\mathrm{va}}\left\|\mathbf{\Lambda}_W\right\| + \left\|MN_{\mathrm{va}}\mathbf{\Lambda}_W - \mathbf{V}_W^\top \bar{\mathbf{X}}^\top \bar{\mathbf{X}} \mathbf{V}_W\right\|$$

$$\leq MN_{\mathrm{va}}\mu_1(\mathbf{\Lambda}_W) + \left\|MN_{\mathrm{va}}\mathbf{\Lambda}_W - \mathbf{V}_W^\top \bar{\mathbf{X}}^\top \bar{\mathbf{X}} \mathbf{V}_W\right\|.$$

From Lemma 19, we have there exists a constant $c$ that with probability at least $1 - e^{-t}$

$$\|\bar{\mathbf{A}}\| \leq cMN_{\mathrm{va}}\mu_1(\mathbf{\Lambda}_W) + cMN_{\mathrm{va}}\mu_1(\mathbf{\Lambda}_W)c_{r_0}(r_0(\mathbf{\Lambda}_W), N, t),$$

$$\|\bar{\mathbf{A}}\| \geq cMN_{\mathrm{va}}\mu_1(\mathbf{\Lambda}_W) - cMN_{\mathrm{va}}\mu_1(\mathbf{\Lambda}_W)c_{r_0}(r_0(\mathbf{\Lambda}_W), N, t).$$

Similarly, because

$$\mathbf{W}_k = \mathbf{V}_{W,k:d}^\top \mathbf{W} \mathbf{V}_{W,k:d}, \ \ \mathrm{Tr}(\mathbf{W}_k) = \mathrm{Tr}(\mathbf{\Lambda}_{W,k:d}), \ \ \mu_1(\mathbf{W}_k) = \mu_1(\mathbf{\Lambda}_{W,k:d}) = \mu_{k+1}(\mathbf{W}).$$

If $r_k(\mathbf{W}) \geq bMN_{\mathrm{va}}$, then there exists a constant $c$ that depends on $\sigma_x$ such that with probability at least $1 - 2e^{-MN_{\mathrm{va}}/c}$

$$\|\bar{\mathbf{A}}_k\| \geq \frac{1}{c}\mu_{k+1}(\mathbf{W})r_k(\mathbf{W}), \ \ \|\bar{\mathbf{A}}_k\| \leq c\mu_{k+1}(\mathbf{W})r_k(\mathbf{W}).$$

$\square$

## D  Auxiliary Lemmas

### D.1  Algebraic properties

**Lemma 11.** *(Lemma 20 in [5]) Suppose $k < n$, $\mathbf{A} \in \mathbb{R}^{n \times n}$ is an invertible matrix, and $\mathbf{Z} \in \mathbb{R}^{n \times k}$ is such that $\mathbf{Z}\mathbf{Z}^\top + \mathbf{A}$ is invertible. Then*

$$\mathbf{Z}^\top(\mathbf{Z}\mathbf{Z}^\top + \mathbf{A})^{-2}\mathbf{Z} = (\mathbf{I} + \mathbf{Z}^\top \mathbf{A}^{-1}\mathbf{Z})^{-1}\mathbf{Z}^\top \mathbf{A}^{-2}\mathbf{Z}(\mathbf{I} + \mathbf{Z}^\top \mathbf{A}^{-1}\mathbf{Z})^{-1}. \tag{75}$$

**Lemma 12** (Weyl's inequality [51])**.** *Let $\mathbf{B} = \mathbf{A} + \mathbf{E}$, $\mathbf{A}, \mathbf{E}$ be $n \times n$ Hermitian matrices. Let $\mu_i(\cdot)$ denote the $i$-th largest eigenvalues of a matrix. Then, we have*

$$\mu_i(\mathbf{A}) + \mu_n(\mathbf{E}) \leq \mu_i(\mathbf{B}) \leq \mu_i(\mathbf{A}) + \mu_1(\mathbf{E}), \quad \forall i \in [n].$$

**Lemma 13** (Von Neumann's trace inequality [34])**.** *If $\mathbf{A}, \mathbf{B} \in \mathbb{R}^{n \times n}$. Let $\sigma_i(\cdot)$ denote the $i$-th largest singular values of a matrix. $\sigma_1(\mathbf{A}) \geq \cdots \geq \sigma_n(\mathbf{A})$, $\sigma_1(\mathbf{B}) \geq \cdots \geq \sigma_n(\mathbf{B})$ respectively, then*

$$|\mathrm{Tr}(\mathbf{A}\mathbf{B})| \leq \sum_{i=1}^n \sigma_i(\mathbf{A})\sigma_i(\mathbf{B}) \leq \sigma_1(\mathbf{B})\sum_{i=1}^n \sigma_i(\mathbf{A}). \tag{76}$$

### D.2  Concentration inequalities

**Lemma 14.** *(Corollary 23 in [5]) There is a universal constant $c$ such that for any non-increasing sequence $\{\lambda_i\}_{i=1}^\infty$ of non-negative numbers such that $\sum_{i=1}^\infty \lambda_i < \infty$, and any independent, centered, $\sigma$-subexponential random variables $\{\xi_i\}_{i=1}^\infty$, and any $t > 0$, with probability at least $1 - 2e^{-t}$*

$$\left|\sum_{i=1}^\infty \lambda_i \xi_i\right| \leq c\sigma \max\left\{t\lambda_1, \sqrt{t\sum_{i=1}^\infty \lambda_i^2}\right\}.$$

**Lemma 15.** *(Corollary 24 in [5]) Suppose $\mathbf{z} \in \mathbb{R}^n$ is a centered random vector with independent $\sigma^2$-subGaussian entries with unit variances, $\mathscr{L}$ is a random subspace of $\mathbb{R}^n$ of codimension $k$, and $\mathscr{L}$ is independent of $\mathbf{z}$. Then for some constant $a$ and any $t > 0$, with probability at least $1 - 3e^{-t}$,*

$$\|\mathbf{z}\|^2 \leq n + a\sigma^2(t + \sqrt{nt}), \qquad \|\Pi_{\mathscr{L}}\mathbf{z}\|^2 \geq n - a\sigma^2(k + t + \sqrt{nt})$$

*where $\Pi_{\mathscr{L}}$ is the orthogonal projection on $\mathscr{L}$.*

**Lemma 16** (Theorem 9 in [31])**.** *Let $\mathbf{x}, \mathbf{x}_1, \ldots, \mathbf{x}_n$ be i.i.d. weakly square integrable centered random vectors in a separable Banach space with covariance $\mathbf{\Sigma}$ and sample covariance $\hat{\mathbf{\Sigma}}$. If $\mathbf{x}$ is subgaussian and pregaussian, define $r(\mathbf{\Sigma}) := (\mathbb{E}[\|\mathbf{x}\|])^2/\|\mathbf{\Sigma}\|$, then there exists a constant $c > 0$ such that, for all $t \geq 1$, with probability at least $1 - e^{-t}$*

$$\|\hat{\mathbf{\Sigma}} - \mathbf{\Sigma}\| \leq c\|\mathbf{\Sigma}\| \max\left\{\sqrt{\frac{r(\mathbf{\Sigma})}{n}}, \frac{r(\mathbf{\Sigma})}{n}, \sqrt{\frac{t}{n}}, \frac{t}{n}\right\}. \tag{77}$$

### D.3 Other supporting lemmas

**Lemma 17** (Bound of $\|\tilde{\mathbf{X}}^\top\tilde{\mathbf{X}} - \bar{\mathbf{X}}^\top\bar{\mathbf{X}}\|$). *Recall that $\bar{\mathbf{X}}^\top\bar{\mathbf{X}}$ and $\tilde{\mathbf{X}}^\top\tilde{\mathbf{X}}$ are computed by*

$$\bar{\mathbf{X}}^\top\bar{\mathbf{X}} = N_{\mathrm{va}} \sum_{m=1}^{M} \mathbf{V}_m \bar{\mathbf{\Lambda}}_m \hat{\mathbf{D}}_m^{\mathrm{va}} \bar{\mathbf{\Lambda}}_m \mathbf{V}_m^\top, \text{ and } \quad \tilde{\mathbf{X}}^\top\tilde{\mathbf{X}} = N_{\mathrm{va}} \sum_{m=1}^{M} \mathbf{V}_m \tilde{\mathbf{\Lambda}}_m^\top \hat{\mathbf{D}}_m^{\mathrm{va}} \tilde{\mathbf{\Lambda}}_m \mathbf{V}_m^\top.$$

*For MAML, for $|\alpha| < \min_m \min\{1/\lambda_{m1}, 1/\mu_1(\hat{\mathbf{Q}}_m^{\mathrm{tr}})\}$, and for $1 \le t \le N_{\mathrm{va}}$, there exists $c > 1$ such that with probability at least $1 - 2Me^{-t}$*

$$\left\|\bar{\mathbf{X}}^{\mathrm{ma}\top}\bar{\mathbf{X}}^{\mathrm{ma}} - \tilde{\mathbf{X}}^{\mathrm{ma}\top}\tilde{\mathbf{X}}^{\mathrm{ma}}\right\| \le c|\alpha| N_{\mathrm{va}} \sum_{m=1}^{M} \lambda_{m1}^2 c_{r_0}(r_0(\mathbf{\Lambda}_m), N_{\mathrm{tr}}, t).$$

*For iMAML, for $\gamma > 0$, and for $1 \le t \le N_{\mathrm{va}}$, there exists $c > 1$ such that with probability at least $1 - 2Me^{-t}$*

$$\left\|\bar{\mathbf{X}}^{\mathrm{im}\top}\bar{\mathbf{X}}^{\mathrm{im}} - \tilde{\mathbf{X}}^{\mathrm{im}\top}\tilde{\mathbf{X}}^{\mathrm{im}}\right\| \le c\gamma^{-1} N_{\mathrm{va}} \sum_{m=1}^{M} \lambda_{m1}^2 c_{r_0}(r_0(\mathbf{\Lambda}_m), N_{\mathrm{tr}}, t).$$

*Proof.* First we have the following relationship

$$\bar{\mathbf{X}}^\top\bar{\mathbf{X}} - \tilde{\mathbf{X}}^\top\tilde{\mathbf{X}} = \frac{1}{2}\Big((\bar{\mathbf{X}} + \tilde{\mathbf{X}})^\top(\bar{\mathbf{X}} - \tilde{\mathbf{X}}) + (\bar{\mathbf{X}} - \tilde{\mathbf{X}})^\top(\bar{\mathbf{X}} + \tilde{\mathbf{X}})\Big).$$

Therefore we have

$$\left\|\bar{\mathbf{X}}^\top\bar{\mathbf{X}} - \tilde{\mathbf{X}}^\top\tilde{\mathbf{X}}\right\| \le \left\|(\bar{\mathbf{X}} + \tilde{\mathbf{X}})^\top(\bar{\mathbf{X}} - \tilde{\mathbf{X}})\right\| = \left\|\sum_{m=1}^{M} \mathbf{V}_m(\bar{\mathbf{\Lambda}}_m + \tilde{\mathbf{\Lambda}}_m)^\top \mathbf{Z}_m^\top \mathbf{Z}_m (\bar{\mathbf{\Lambda}}_m - \tilde{\mathbf{\Lambda}}_m)\mathbf{V}_m^\top\right\|$$

For MAML, we have

$$\left\|\bar{\mathbf{X}}^{\mathrm{ma}\top}\bar{\mathbf{X}}^{\mathrm{ma}} - \tilde{\mathbf{X}}^{\mathrm{ma}\top}\tilde{\mathbf{X}}^{\mathrm{ma}}\right\|$$

$$\le N_{\mathrm{va}}\left\|\sum_{m=1}^{M} \mathbf{V}_m\Big((\mathbf{I} - \alpha\mathbf{\Lambda}_m) + (\mathbf{I} - \alpha\mathbf{\Lambda}_m^{\frac{1}{2}}\hat{\mathbf{D}}_m^{\mathrm{tr}}\mathbf{\Lambda}_m^{\frac{1}{2}})\Big)\mathbf{\Lambda}_m^{\frac{1}{2}}\hat{\mathbf{D}}_m^{\mathrm{va}}\mathbf{\Lambda}_m^{\frac{1}{2}}\big(\alpha\mathbf{\Lambda}_m^{\frac{1}{2}}(\hat{\mathbf{D}}_m^{\mathrm{tr}} - \mathbf{I})\mathbf{\Lambda}_m^{\frac{1}{2}}\big)\mathbf{V}_m^\top\right\|$$

$$\le N_{\mathrm{va}} \sum_{m=1}^{M} \underbrace{\left\|(\mathbf{I} - \alpha\mathbf{\Lambda}_m) + (\mathbf{I} - \alpha\mathbf{\Lambda}_m^{\frac{1}{2}}\hat{\mathbf{D}}_m^{\mathrm{tr}}\mathbf{\Lambda}_m^{\frac{1}{2}})\right\|}_{I_1} \underbrace{\left\|\mathbf{\Lambda}_m^{\frac{1}{2}}\hat{\mathbf{D}}_m^{\mathrm{va}}\mathbf{\Lambda}_m^{\frac{1}{2}}\right\|}_{I_2} \underbrace{\left\|\alpha\mathbf{\Lambda}_m^{\frac{1}{2}}(\hat{\mathbf{D}}_m^{\mathrm{tr}} - \mathbf{I})\mathbf{\Lambda}_m^{\frac{1}{2}}\right\|}_{I_3}$$

where we choose $\alpha$ such that $\|\alpha\mathbf{\Lambda}_m^{\frac{1}{2}}\hat{\mathbf{D}}_m^{\mathrm{tr}}\mathbf{\Lambda}_m^{\frac{1}{2}}\| < 1$ and $\|\alpha\mathbf{\Lambda}_m\| < 1$. Therefore

$$I_1 = \left\|(\mathbf{I} - \alpha\mathbf{\Lambda}_m) + (\mathbf{I} - \alpha\mathbf{\Lambda}_m^{\frac{1}{2}}\hat{\mathbf{D}}_m^{\mathrm{tr}}\mathbf{\Lambda}_m^{\frac{1}{2}})\right\| \le 4. \tag{78}$$

Also based on Lemma 16 we can bound $I_2$ and $I_3$ since $\mathbf{\Lambda}_m^{\frac{1}{2}}\hat{\mathbf{D}}_m^{\mathrm{tr}}\mathbf{\Lambda}_m^{\frac{1}{2}}$ and $\mathbf{\Lambda}_m^{\frac{1}{2}}\hat{\mathbf{D}}_m^{\mathrm{va}}\mathbf{\Lambda}_m^{\frac{1}{2}}$ are the sample covariances of $\mathbf{\Lambda}_m$.

There exists a constant $c$ that for all $t \ge 1$, with probability at least $1 - e^{-t}$ we have

$$I_2 = \left\|\mathbf{\Lambda}_m^{\frac{1}{2}}\hat{\mathbf{D}}_m^{\mathrm{va}}\mathbf{\Lambda}_m^{\frac{1}{2}}\right\| \le \left\|\mathbf{\Lambda}_m\right\| + \left\|\mathbf{\Lambda}_m - \mathbf{\Lambda}_m^{\frac{1}{2}}\hat{\mathbf{D}}_m^{\mathrm{va}}\mathbf{\Lambda}_m^{\frac{1}{2}}\right\|$$
$$\le \lambda_{m1} + c\lambda_{m1}c_{r_0}(r_0(\mathbf{\Lambda}_m), N_{\mathrm{va}}, t) \tag{79}$$

and for all $t \ge 1$, with probability at least $1 - e^{-t}$ we have

$$I_3 = \left\|\alpha\mathbf{\Lambda}_m^{\frac{1}{2}}(\hat{\mathbf{D}}_m^{\mathrm{tr}} - \mathbf{I})\mathbf{\Lambda}_m^{\frac{1}{2}}\right\| \le c|\alpha|\lambda_{m1}c_{r_0}(r_0(\mathbf{\Lambda}_m), N_{\mathrm{tr}}, t) \tag{80}$$

Combining the bounds for $I_1, I_2, I_3$ and applying union bound over training and validation data for all tasks, when $|\alpha| < \min_m \min\{1/\lambda_{m1}, 1/\mu_1(\mathbf{\Lambda}_m^{\frac{1}{2}}\hat{\mathbf{D}}_m^{\mathrm{tr}}\mathbf{\Lambda}_m^{\frac{1}{2}})\}$, and for $1 \le t \le N_{\mathrm{va}}$, there exists $c > 1$ such that the following holds with probability at least $1 - 2Me^{-t}$

$$\left\|\bar{\mathbf{X}}^{\mathrm{ma}\top}\bar{\mathbf{X}}^{\mathrm{ma}} - \tilde{\mathbf{X}}^{\mathrm{ma}\top}\tilde{\mathbf{X}}^{\mathrm{ma}}\right\| \le c|\alpha| N_{\mathrm{va}} \sum_{m=1}^{M} \lambda_{m1}^2 c_{r_0}(r_0(\mathbf{\Lambda}_m), N_{\mathrm{tr}}, t).$$

Similarly, for iMAML, we have

$$\left\|\bar{\mathbf{X}}^{\mathrm{im}\top}\bar{\mathbf{X}}^{\mathrm{im}} - \tilde{\mathbf{X}}^{\mathrm{im}\top}\tilde{\mathbf{X}}^{\mathrm{im}}\right\| \le \left\|\sum_{m=1}^{M}\mathbf{V}_m(\bar{\mathbf{\Lambda}}_m + \tilde{\mathbf{\Lambda}}_m)^\top\mathbf{Z}_m^\top\mathbf{Z}_m(\bar{\mathbf{\Lambda}}_m - \tilde{\mathbf{\Lambda}}_m)\mathbf{V}_m^\top\right\|$$

$$=N_{\mathrm{va}}\left\|\sum_{m=1}^{M}\mathbf{V}_m\left((\mathbf{I}+\gamma^{-1}\mathbf{\Lambda}_m)^{-1} + (\mathbf{I}+\gamma^{-1}\mathbf{\Lambda}_m^{\frac{1}{2}}\hat{\mathbf{D}}_m^{\mathrm{tr}}\mathbf{\Lambda}_m^{\frac{1}{2}})^{-1}\right)\mathbf{\Lambda}_m^{\frac{1}{2}}\hat{\mathbf{D}}_m^{\mathrm{va}}\mathbf{\Lambda}_m^{\frac{1}{2}}\right.$$

$$\left.\cdot\left((\mathbf{I}+\gamma^{-1}\mathbf{\Lambda}_m)^{-1}(\gamma^{-1}\mathbf{\Lambda}_m^{\frac{1}{2}}(\hat{\mathbf{D}}_m^{\mathrm{tr}} - \mathbf{I})\mathbf{\Lambda}_m^{\frac{1}{2}})(\mathbf{I}+\gamma^{-1}\mathbf{\Lambda}_m^{\frac{1}{2}}\hat{\mathbf{D}}_m^{\mathrm{tr}}\mathbf{\Lambda}_m^{\frac{1}{2}})^{-1}\right)\mathbf{V}_m^\top\right\|$$

$$\le N_{\mathrm{va}}\sum_{m=1}^{M}\underbrace{\left\|(\mathbf{I}+\gamma^{-1}\mathbf{\Lambda}_m)^{-1} + (\mathbf{I}+\gamma^{-1}\mathbf{\Lambda}_m^{\frac{1}{2}}\hat{\mathbf{D}}_m^{\mathrm{tr}}\mathbf{\Lambda}_m^{\frac{1}{2}})^{-1}\right\|}_{I_4}\underbrace{\left\|\mathbf{\Lambda}_m^{\frac{1}{2}}\hat{\mathbf{D}}_m^{\mathrm{va}}\mathbf{\Lambda}_m^{\frac{1}{2}}\right\|}_{I_2}$$

$$\cdot\underbrace{\left\|(\mathbf{I}+\gamma^{-1}\mathbf{\Lambda}_m)^{-1}(\gamma^{-1}\mathbf{\Lambda}_m^{\frac{1}{2}}(\hat{\mathbf{D}}_m^{\mathrm{tr}} - \mathbf{I})\mathbf{\Lambda}_m^{\frac{1}{2}})(\mathbf{I}+\gamma^{-1}\mathbf{\Lambda}_m^{\frac{1}{2}}\hat{\mathbf{D}}_m^{\mathrm{tr}}\mathbf{\Lambda}_m^{\frac{1}{2}})^{-1}\right\|}_{I_5}$$

where $I_4$ can be bounded by

$$I_4 = \left\|(\mathbf{I}+\gamma^{-1}\mathbf{\Lambda}_m)^{-1} + (\mathbf{I}+\gamma^{-1}\mathbf{\Lambda}_m^{\frac{1}{2}}\hat{\mathbf{D}}_m^{\mathrm{tr}}\mathbf{\Lambda}_m^{\frac{1}{2}})^{-1}\right\| \le 2. \tag{81}$$

And $I_5$ can be bounded by

$$I_5 = \left\|(\mathbf{I}+\gamma^{-1}\mathbf{\Lambda}_m)^{-1}(\gamma^{-1}\mathbf{\Lambda}_m^{\frac{1}{2}}(\hat{\mathbf{D}}_m^{\mathrm{tr}} - \mathbf{I})\mathbf{\Lambda}_m^{\frac{1}{2}})(\mathbf{I}+\gamma^{-1}\mathbf{\Lambda}_m^{\frac{1}{2}}\hat{\mathbf{D}}_m^{\mathrm{tr}}\mathbf{\Lambda}_m^{\frac{1}{2}})^{-1}\right\|$$

$$\le \left\|(\mathbf{I}+\gamma^{-1}\mathbf{\Lambda}_m)^{-1}\right\|\left\|\gamma^{-1}\mathbf{\Lambda}_m^{\frac{1}{2}}(\hat{\mathbf{D}}_m^{\mathrm{tr}} - \mathbf{I})\mathbf{\Lambda}_m^{\frac{1}{2}}\right\|\left\|(\mathbf{I}+\gamma^{-1}\mathbf{\Lambda}_m^{\frac{1}{2}}\hat{\mathbf{D}}_m^{\mathrm{tr}}\mathbf{\Lambda}_m^{\frac{1}{2}})^{-1}\right\|$$

$$\le \left\|\gamma^{-1}\mathbf{\Lambda}_m^{\frac{1}{2}}(\hat{\mathbf{D}}_m^{\mathrm{tr}} - \mathbf{I})\mathbf{\Lambda}_m^{\frac{1}{2}}\right\| \tag{82}$$

Based on Lemma 16, we can bound $I_5$ similarly as $I_3$. There exists a constant $c$ that for all $t \ge 1$, with probability at least $1 - e^{-t}$ we have

$$I_5 \le \left\|\gamma^{-1}\mathbf{\Lambda}_m^{\frac{1}{2}}(\hat{\mathbf{D}}_m^{\mathrm{tr}} - \mathbf{I})\mathbf{\Lambda}_m^{\frac{1}{2}}\right\| \le c\gamma^{-1}\lambda_{m1}c_{r_0}(r_0(\mathbf{\Lambda}_m), N_{\mathrm{tr}}, t). \tag{83}$$

Combining the bounds for $I_4$, $I_2$, $I_5$ and applying union bound over training and validation data for all tasks, for $\gamma > 0$, and for $1 \le t \le N_{\mathrm{va}}$, there exists $c > 1$ such that the following holds with probability at least $1 - 2Me^{-t}$

$$\left\|\bar{\mathbf{X}}^{\mathrm{im}\top}\bar{\mathbf{X}}^{\mathrm{im}} - \tilde{\mathbf{X}}^{\mathrm{im}\top}\tilde{\mathbf{X}}^{\mathrm{im}}\right\| \le c\gamma^{-1}N_{\mathrm{va}}\sum_{m=1}^{M}\lambda_{m1}^2 c_{r_0}(r_0(\mathbf{\Lambda}_m), N_{\mathrm{tr}}, t).$$

This completes the proof for Lemma 17. $\qquad\square$

**Lemma 18** (Bound of $\|\tilde{\mathbf{X}} - \bar{\mathbf{X}}\|$). *Recall that $\bar{\mathbf{X}}$ and $\tilde{\mathbf{X}}$ are defined as*

$$\bar{\mathbf{X}} = [\mathbf{Z}_m^{\mathrm{va}}\bar{\mathbf{\Lambda}}_m\mathbf{V}_m^\top], \text{ and } \quad \tilde{\mathbf{X}} = [\mathbf{Z}_m^{\mathrm{va}}\tilde{\mathbf{\Lambda}}_m\mathbf{V}_m^\top].$$

*Define $c_{r_0}(r(\mathbf{\Lambda}), N, t) := \max\left\{\sqrt{\frac{r(\mathbf{\Lambda})}{N}}, \frac{r(\mathbf{\Lambda})}{N}, \sqrt{\frac{t}{N}}, \frac{t}{N}\right\}$. For MAML, and for $1 \le t \le N_{\mathrm{va}}$, there exists $c > 1$ such that with probability at least $1 - 2Me^{-t}$*

$$\left\|\bar{\mathbf{X}}^{\mathrm{ma}} - \tilde{\mathbf{X}}^{\mathrm{ma}}\right\| \le c|\alpha|N_{\mathrm{va}}\left(\sum_{m=1}^{M}\lambda_{m1}^2 c_{r_0}^2(r(\mathbf{\Lambda}_m), N_{\mathrm{tr}}, t)\right)^{\frac{1}{2}}.$$

*For iMAML, and for $\gamma > 0$, and for $1 \le t \le N_{\mathrm{va}}$, there exists $c > 1$ such that with probability at least $1 - 2Me^{-t}$*

$$\left\|\bar{\mathbf{X}}^{\mathrm{im}} - \tilde{\mathbf{X}}^{\mathrm{im}}\right\| \le c\gamma^{-1}N_{\mathrm{va}}\left(\sum_{m=1}^{M}\lambda_{m1}^2 c_{r_0}^2(r(\mathbf{\Lambda}_m), N_{\mathrm{tr}}, t)\right)^{\frac{1}{2}}.$$

*Proof.* First we have the following relationship

$$\|\tilde{\mathbf{X}} - \bar{\mathbf{X}}\| = \left\|(\bar{\mathbf{X}} - \tilde{\mathbf{X}})^\top(\bar{\mathbf{X}} - \tilde{\mathbf{X}})\right\|^{\frac{1}{2}}.$$

For MAML, we have

$$\left\|(\bar{\mathbf{X}}^{\mathrm{ma}} - \tilde{\mathbf{X}}^{\mathrm{ma}})^\top(\bar{\mathbf{X}}^{\mathrm{ma}} - \tilde{\mathbf{X}}^{\mathrm{ma}})\right\|$$

$$= N_{\mathrm{va}}\left\| \sum_{m=1}^{M} \mathbf{V}_m\big(\alpha\mathbf{\Lambda}_m^{\frac{1}{2}}(\hat{\mathbf{D}}_m^{\mathrm{tr}} - \mathbf{I})\mathbf{\Lambda}_m^{\frac{1}{2}}\big)\mathbf{\Lambda}_m^{\frac{1}{2}}\hat{\mathbf{D}}_m^{\mathrm{va}}\mathbf{\Lambda}_m^{\frac{1}{2}}\big(\alpha\mathbf{\Lambda}_m^{\frac{1}{2}}(\hat{\mathbf{D}}_m^{\mathrm{tr}} - \mathbf{I})\mathbf{\Lambda}_m^{\frac{1}{2}}\big)\mathbf{V}_m^\top\right\|$$

$$\leq N_{\mathrm{va}} \sum_{m=1}^{M} \underbrace{\left\|\alpha\mathbf{\Lambda}_m^{\frac{1}{2}}(\hat{\mathbf{D}}_m^{\mathrm{tr}} - \mathbf{I})\mathbf{\Lambda}_m^{\frac{1}{2}}\right\|}_{I_1} \underbrace{\left\|\mathbf{\Lambda}_m^{\frac{1}{2}}\hat{\mathbf{D}}_m^{\mathrm{va}}\mathbf{\Lambda}_m^{\frac{1}{2}}\right\|}_{I_2} \underbrace{\left\|\alpha\mathbf{\Lambda}_m^{\frac{1}{2}}(\hat{\mathbf{D}}_m^{\mathrm{tr}} - \mathbf{I})\mathbf{\Lambda}_m^{\frac{1}{2}}\right\|}_{I_1}$$

where based on Lemma 16 we can bound $I_1$ and $I_2$ since $\mathbf{\Lambda}_m^{\frac{1}{2}}\hat{\mathbf{D}}_m^{\mathrm{tr}}\mathbf{\Lambda}_m^{\frac{1}{2}}$ and $\mathbf{\Lambda}_m^{\frac{1}{2}}\hat{\mathbf{D}}_m^{\mathrm{va}}\mathbf{\Lambda}_m^{\frac{1}{2}}$ are the sample covariances of $\mathbf{\Lambda}_m$.

There exists a constant $c$ that for all $t \geq 1$, with probability at least $1 - e^{-t}$ we have

$$\begin{aligned} I_2 &= \left\|\mathbf{\Lambda}_m^{\frac{1}{2}}\hat{\mathbf{D}}_m^{\mathrm{va}}\mathbf{\Lambda}_m^{\frac{1}{2}}\right\| \leq \left\|\mathbf{\Lambda}_m\right\| + \left\|\mathbf{\Lambda}_m - \mathbf{\Lambda}_m^{\frac{1}{2}}\hat{\mathbf{D}}_m^{\mathrm{va}}\mathbf{\Lambda}_m^{\frac{1}{2}}\right\| \\ &\leq \lambda_{m1} + c\lambda_{m1}c_{r_0}(r_0(\mathbf{\Lambda}_m), N_{\mathrm{va}}, t) \end{aligned} \tag{84}$$

and for all $t \geq 1$, with probability at least $1 - e^{-t}$ we have

$$I_1 = \left\|\alpha\mathbf{\Lambda}_m^{\frac{1}{2}}(\hat{\mathbf{D}}_m^{\mathrm{tr}} - \mathbf{I})\mathbf{\Lambda}_m^{\frac{1}{2}}\right\| \leq c|\alpha|\lambda_{m1}c_{r_0}(r_0(\mathbf{\Lambda}_m), N_{\mathrm{tr}}, t) \tag{85}$$

Combining the bounds for $I_1$, $I_2$ and applying union bound over training and validation data for all tasks, we have for $1 \leq t \leq N_{\mathrm{va}}$, there exists $c > 1$ such that the following holds with probability at least $1 - 2Me^{-t}$

$$\left\|(\bar{\mathbf{X}}^{\mathrm{ma}} - \tilde{\mathbf{X}}^{\mathrm{ma}})^\top(\bar{\mathbf{X}}^{\mathrm{ma}} - \tilde{\mathbf{X}}^{\mathrm{ma}})\right\| \leq c|\alpha|^2 N_{\mathrm{va}} \sum_{m=1}^{M} \lambda_{m1}^3 c_{\mathrm{sample}}^2(r(\mathbf{\Lambda}_m), N_{\mathrm{tr}}, t).$$

Similarly, for iMAML, we have

$$\left\|(\bar{\mathbf{X}}^{\mathrm{im}} - \tilde{\mathbf{X}}^{\mathrm{im}})^\top(\bar{\mathbf{X}}^{\mathrm{im}} - \tilde{\mathbf{X}}^{\mathrm{im}})\right\| = \left\| \sum_{m=1}^{M} \mathbf{V}_m(\bar{\mathbf{\Lambda}}_m - \tilde{\mathbf{\Lambda}}_m)^\top\mathbf{Z}_m^\top\mathbf{Z}_m(\bar{\mathbf{\Lambda}}_m - \tilde{\mathbf{\Lambda}}_m)\mathbf{V}_m^\top\right\|$$

$$= N_{\mathrm{va}}\left\| \sum_{m=1}^{M} \mathbf{V}_m\big((\mathbf{I} + \gamma^{-1}\mathbf{\Lambda}_m)^{-1}(\gamma^{-1}\mathbf{\Lambda}_m^{\frac{1}{2}}(\hat{\mathbf{D}}_m^{\mathrm{tr}} - \mathbf{I})\mathbf{\Lambda}_m^{\frac{1}{2}})(\mathbf{I} + \gamma^{-1}\mathbf{\Lambda}_m^{\frac{1}{2}}\hat{\mathbf{D}}_m^{\mathrm{tr}}\mathbf{\Lambda}_m^{\frac{1}{2}})^{-1}\big)\mathbf{\Lambda}_m^{\frac{1}{2}}\hat{\mathbf{D}}_m^{\mathrm{va}}\mathbf{\Lambda}_m^{\frac{1}{2}} \right.$$

$$\left. \cdot \big((\mathbf{I} + \gamma^{-1}\mathbf{\Lambda}_m)^{-1}(\gamma^{-1}\mathbf{\Lambda}_m^{\frac{1}{2}}(\hat{\mathbf{D}}_m^{\mathrm{tr}} - \mathbf{I})\mathbf{\Lambda}_m^{\frac{1}{2}})(\mathbf{I} + \gamma^{-1}\mathbf{\Lambda}_m^{\frac{1}{2}}\hat{\mathbf{D}}_m^{\mathrm{tr}}\mathbf{\Lambda}_m^{\frac{1}{2}})^{-1}\big)\mathbf{V}_m^\top\right\|$$

$$\leq N_{\mathrm{va}} \sum_{m=1}^{M} \underbrace{\left\|(\mathbf{I} + \gamma^{-1}\mathbf{\Lambda}_m)^{-1}(\gamma^{-1}\mathbf{\Lambda}_m^{\frac{1}{2}}(\hat{\mathbf{D}}_m^{\mathrm{tr}} - \mathbf{I})\mathbf{\Lambda}_m^{\frac{1}{2}})(\mathbf{I} + \gamma^{-1}\mathbf{\Lambda}_m^{\frac{1}{2}}\hat{\mathbf{D}}_m^{\mathrm{tr}}\mathbf{\Lambda}_m^{\frac{1}{2}})^{-1}\right\|}_{I_3} \underbrace{\left\|\mathbf{\Lambda}_m^{\frac{1}{2}}\hat{\mathbf{D}}_m^{\mathrm{va}}\mathbf{\Lambda}_m^{\frac{1}{2}}\right\|}_{I_2}$$

$$\cdot \underbrace{\left\|(\mathbf{I} + \gamma^{-1}\mathbf{\Lambda}_m)^{-1}(\gamma^{-1}\mathbf{\Lambda}_m^{\frac{1}{2}}(\hat{\mathbf{D}}_m^{\mathrm{tr}} - \mathbf{I})\mathbf{\Lambda}_m^{\frac{1}{2}})(\mathbf{I} + \gamma^{-1}\mathbf{\Lambda}_m^{\frac{1}{2}}\hat{\mathbf{D}}_m^{\mathrm{tr}}\mathbf{\Lambda}_m^{\frac{1}{2}})^{-1}\right\|}_{I_3}$$

where $I_3$ can be bounded by

$$\begin{aligned} I_3 &= \left\|(\mathbf{I} + \gamma^{-1}\mathbf{\Lambda}_m)^{-1}(\gamma^{-1}\mathbf{\Lambda}_m^{\frac{1}{2}}(\hat{\mathbf{D}}_m^{\mathrm{tr}} - \mathbf{I})\mathbf{\Lambda}_m^{\frac{1}{2}})(\mathbf{I} + \gamma^{-1}\mathbf{\Lambda}_m^{\frac{1}{2}}\hat{\mathbf{D}}_m^{\mathrm{tr}}\mathbf{\Lambda}_m^{\frac{1}{2}})^{-1}\right\| \\ &\leq \left\|(\mathbf{I} + \gamma^{-1}\mathbf{\Lambda}_m)^{-1}\right\|\left\|\gamma^{-1}\mathbf{\Lambda}_m^{\frac{1}{2}}(\hat{\mathbf{D}}_m^{\mathrm{tr}} - \mathbf{I})\mathbf{\Lambda}_m^{\frac{1}{2}}\right\|\left\|(\mathbf{I} + \gamma^{-1}\mathbf{\Lambda}_m^{\frac{1}{2}}\hat{\mathbf{D}}_m^{\mathrm{tr}}\mathbf{\Lambda}_m^{\frac{1}{2}})^{-1}\right\| \\ &\leq \left\|\gamma^{-1}\mathbf{\Lambda}_m^{\frac{1}{2}}(\hat{\mathbf{D}}_m^{\mathrm{tr}} - \mathbf{I})\mathbf{\Lambda}_m^{\frac{1}{2}}\right\| \end{aligned} \tag{86}$$

Based on Lemma 16, we can bound $I_3$ similarly as $I_1$. There exists a constant $c$ that for all $t \geq 1$, with probability at least $1 - e^{-t}$ we have

$$I_3 \leq \left\| \gamma^{-1} \mathbf{\Lambda}_m^{\frac{1}{2}} (\hat{\mathbf{D}}_m^{\mathrm{tr}} - \mathbf{I}) \mathbf{\Lambda}_m^{\frac{1}{2}} \right\| \leq c\gamma^{-1}\lambda_{m1} c_{r_0}(r(\mathbf{\Lambda}_m), N_{\mathrm{tr}}, t). \tag{87}$$

Combining the bounds for $I_2$, $I_3$ and applying union bound over training and validation data for all tasks, for $\gamma > 0$, and for $1 \leq t \leq N_{\mathrm{va}}$, there exists $c > 1$ such that the following holds with probability at least $1 - 2Me^{-t}$

$$\left\| (\bar{\mathbf{X}}^{\mathrm{im}} - \tilde{\mathbf{X}}^{\mathrm{im}})^\top (\bar{\mathbf{X}}^{\mathrm{im}} - \tilde{\mathbf{X}}^{\mathrm{im}}) \right\| \leq c\gamma^{-2} N_{\mathrm{va}} \sum_{m=1}^{M} \lambda_{m1}^3 c_{r_0}^2(r(\mathbf{\Lambda}_m), N_{\mathrm{tr}}, t).$$

This completes the proof for Lemma 18. $\qquad\square$

**Lemma 19** (Bound of $\|\mathbf{\Lambda}^{\frac{1}{2}}\mathbf{Z}^\top\mathbf{Z}\mathbf{\Lambda}^{\frac{1}{2}}\|$, $\|\mathbf{Z}\mathbf{\Lambda}\mathbf{Z}^\top\|$ and $\|\mathbf{Z}\mathbf{\Lambda}^{\frac{1}{2}}\|$). *Let $\mathbf{Z} \in \mathbb{R}^{N \times d}$, consists of centered, independent, $\sigma_x$-subGaussian entries. And $\mathbf{\Lambda} = \mathrm{diag}(\lambda_1, \ldots, \lambda_d) \in \mathbb{R}^{d \times d}$ be a positive definite diagonal matrix with $\lambda_1 \geq \lambda_2 \geq \cdots \geq \lambda_d$. Then $\|\mathbf{\Lambda}^{\frac{1}{2}}\mathbf{Z}^\top\mathbf{Z}\mathbf{\Lambda}^{\frac{1}{2}}\| = \|\mathbf{Z}\mathbf{\Lambda}\mathbf{Z}^\top\|$, and there exists a constant $c > 0$ such that, for all $t \geq 1$, with probability at least $1 - e^{-t}$*

$$\|\mathbf{Z}\mathbf{\Lambda}\mathbf{Z}^\top\| \leq N\lambda_1 + cN\lambda_1 c_{r_0}(r_0(\mathbf{\Lambda}), N, t), \quad \|\mathbf{Z}\mathbf{\Lambda}^{\frac{1}{2}}\| \leq \sqrt{N\lambda_1}\left(1 + cc_{r_0}(r_0(\mathbf{\Lambda}), N, t)\right)^{\frac{1}{2}}.$$

*Proof.*

$$\|\mathbf{\Lambda}^{\frac{1}{2}}\mathbf{Z}^\top\mathbf{Z}\mathbf{\Lambda}^{\frac{1}{2}}\| = \|\mathbf{\Lambda}^{\frac{1}{2}}\mathbf{Z}^\top\mathbf{Z}\mathbf{\Lambda}^{\frac{1}{2}} - N\mathbf{\Lambda} + N\mathbf{\Lambda}\| \leq N\left\|\frac{1}{N}\mathbf{\Lambda}^{\frac{1}{2}}\mathbf{Z}^\top\mathbf{Z}\mathbf{\Lambda}^{\frac{1}{2}} - \mathbf{\Lambda}\right\| + N\|\mathbf{\Lambda}\|.$$

By Lemma 16, we have there exists a constant $c > 0$ such that, for all $t \geq 1$, with probability at least $1 - e^{-t}$

$$\left\|\frac{1}{N}\mathbf{\Lambda}^{\frac{1}{2}}\mathbf{Z}^\top\mathbf{Z}\mathbf{\Lambda}^{\frac{1}{2}} - \mathbf{\Lambda}\right\| \leq c\|\mathbf{\Lambda}\|c_{r_0}(r_0(\mathbf{\Lambda}), N, t).$$

Therefore, there exists a constant $c > 0$ such that, for all $t \geq 1$, with probability at least $1 - e^{-t}$

$$\left\|\mathbf{\Lambda}^{\frac{1}{2}}\mathbf{Z}^\top\mathbf{Z}\mathbf{\Lambda}^{\frac{1}{2}}\right\| \leq N\lambda_1 + cN\lambda_1 c_{r_0}(r_0(\mathbf{\Lambda}), N, t).$$

Because $\|\mathbf{Z}\mathbf{\Lambda}\mathbf{Z}^\top\| = \|\mathbf{\Lambda}^{\frac{1}{2}}\mathbf{Z}^\top\mathbf{Z}\mathbf{\Lambda}^{\frac{1}{2}}\|$, and $\|\mathbf{Z}\mathbf{\Lambda}^{\frac{1}{2}}\| = \|\mathbf{\Lambda}^{\frac{1}{2}}\mathbf{Z}^\top\mathbf{Z}\mathbf{\Lambda}^{\frac{1}{2}}\|^{\frac{1}{2}}$, it leads to the conclusion. $\quad\square$