# OpenReview forum: "Understanding Benign Overfitting in Gradient-Based Meta Learning"
_NeurIPS.cc/2022/Conference — NeurIPS 2022 Accept_

### Official Review · Reviewer_FrXq · 2022-07-05

**Rating:** 7
**Confidence:** 4
**Soundness:** 3 good
**Presentation:** 4 excellent
**Contribution:** 3 good

**Summary:**

The paper provides generalization bounds for overparamterized linear model in meta learning, for three different methods: empirical risk minimization (ERM), model agnostic meta learning (MAML) and implicit model agnostic meta learning (iMAML).

Model. In a meta learning problem, there are M tasks and labeled data $\{(x_{i, m}, y_{i,m})\}_{i \in [N], m \in [M]}$ are drawn i.i.d. from each distribution. The paper considers linear model, where the goal is to come up a share $\theta^{*} \in \mathbb{R}^{d}$ that fits M tasks: For ERM, it just minimizes the square loss; while for MAML and iMAML, it reduces to minimize certain regularized square loss. The paper considers the ``overparameterized'' regime where $d\geq MN$, and the generalization bound explain the the benign overfitting phenomena in certain angle.

Results. It is hard to summarize the generalization in a few words. But it follows the seminal work of [Barlett et al. PNAS 2020], but with an extra term that characterize the cross-task data heterogeneity, which is defined as the maximum discrepancy between data covariance matrix of each task, which is unique to meta-learning or multi-task learning.

Methods. The method generally follows the work of [Barlett et al. PNAS 2020] with extra effort handling the cross-task data heterogeneity, i.e., data coming from different task could have different eigenvector. I feel it is non-trivial to deal with this factor.

**Questions:**

The paper outline some technical difficulty in the second, and resolves nicely in the proof. Personally I would recommend explicit state how they resolve these issues in their proof outline. One technical question that appears to be non-trivial to me is that how you handle different eigen-vectors from different tasks? It seems the covariance matrix could be quite different (though the eigenvalue is assumed to aligned well). Can you provide some intuition.


Another minor comments is to state that all figures are numerical calculation of the generalization bound in some place, it is bit confusing to me.




**Strengths And Weaknesses:**

Strength. The paper provides the first generalization bound for meta learning under over-parameterized linear model, the proof handles data coming from different source with different covariance matrix (eigenvector especially) is non-trivial to me. The paper is very well-written.

Weakness. There is no major weakness of the paper. Here is one minor question.

The data assumes overparameterized model for multi-task learning and the benign overfitting somewhat motivates from empirical observation of double descent. Are these valid? E.g. does double descent happen for meta learning, and does the neural network for MAML very over-parameterized when data comes from multi-task are available? It would be good to have some discussion later.

--------------------- Post rebuttal ---------------------
I have read the rebuttal and keep my evaluation.

---

> ### Author Response · Authors · 2022-08-02
> **Response to Reviewer FrXq**
>
> ## Response to Reviewer FrXq
>
> We thank the reviewer for the support and the careful review. Our point-to-point response to your remaining concerns follows next.
>
>
> **Q1. Empirical example that supports benign overfitting for meta learning and very over-parameterized neural networks for MAML.**
>
>
> We provide example in the answer to **General Response-Q1**.
>
>
> **Q2. Personally I would recommend explicitly state how they resolve these issues in their proof outline. How you handle different eigen-vectors from different tasks? Can you provide some intuition.**
>
>
> Thanks for the suggestion. Due to space limitation, we highlighted some proof rationale in the appendix. Indeed, including them in the proof outline will improve readability.
>
>
>
> In particular, the challenge of handling multi-task data matrices with different eigenvectors mainly appears in the proof of Lemma 3. We have highlighted some key steps in the appendix, but will add more and move them to the main paper. Roughly speaking, the proof needs to deal with the trace of the product of symmetric postive semidefinite (PSD) matrices with different eigenvectors. For example, we need to bound $\mathrm{tr}(AB)$ where $A,B$ are symmetric PSD matrices with eigenvalues $a_1\geq a_2\geq \dots \geq a_d > 0$ and $b_1\geq b_2\geq \dots \geq b_d > 0$, respectively. We use the Von Neumann’s trace inequality (and other basic properties of the trace) in Lemma 7. Von Neumann’s trace inequality gives that  $\mathrm{tr}(AB) \leq \sum_{i=1}^d a_i b_i$, which shows that the trace of the product of symmetric PSD matrices with different eigenvectors are bounded by the sum of the product of their ordered eigenvalues.
>
>
>
>
> When the additional page allows in the final version, we will promise to make corresponding changes by explicitly stating how we resolve these issues in the proof outline in the paper.

---

### Official Review · Reviewer_WJZn · 2022-07-09

**Rating:** 6
**Confidence:** 3
**Soundness:** 3 good
**Presentation:** 3 good
**Contribution:** 2 fair

**Summary:**

The paper aims to answer whether overparameterized nested meta learning models would lead to overfitting or not. As a first step, the paper theoretically analyzes the generalization performance of relatively tractable linear models in the meta linear regression problem with the minimum norm solution. By decomposing excess risk into three individual terms and bounding them separately, the bound for the excess risk can be obtained. Based on the bound, condition for benign overfitting in meta learning is derived. Numerical experiments validated the theoretical findings.

**Questions:**

- The MAML analyzed in the paper only uses one gradient step in adaptation, while a more realistic scenario is to use multiple gradient descents in the adaptation phase. How does this change affect your analysis?

- Example 1 and Example 2 show the settings that satisfy the benign overfitting condition. What setting can violate such condition?

- Why larger data heterogeneity makes it more difficult for the benign overfitting condition to be satisfied for both MAML and iMAML?

- What's the connection between the derived results and real meta-learning applications? How can the derived results help understand more practical meta-learning settings?

**Limitations:**

Yes.

**Strengths And Weaknesses:**

Pros:

+ The paper demonstrated how the analysis of benign overfitting in linear regression can be extended to meta linear regression problems. The analysis provides understanding on how data data heterogeneity, model adaptation affect benign overfitting in meta-learning. The derived bound of the excess risk is verified by showing the same trend as the simulated excess risk in numerical simulations.

+ The paper is well organized. It clearly shows the assumption of the derivations and highlights the key difference between different methods, e.g. Figure 1 and Table 2. The contents in the paper are easy to follow.

Cons:

- It is not clear how the benign overfitting condition in (13) can be used to determine whether certain settings are benign or not. For example, the parameters M, d in the settings of Example 1 and Example 2 are finite while (13) expresses the condition in limit form. More explanations are needed to demonstrate the significance of the derived benign overfitting condition.

- The use of "nested meta-learning" is confusing since meta-learning itself indicates a nested structure. Using "nested" before meta-learning may refer to a new method where multiple (more than two) levels of optimization are used. However, the paper actually analyzed MAML and iMAML, which are gradient-based meta-learning methods using two-level optimization.

- There is a large gap between the theoretical analysis and and the meta-learning methods used in practice in terms of the data, model architecture, and prediction task. Although the setting in this paper facilitates theoretical analysis, it would be better to show the possible connection between derived results and real meta-learning applications via discussion or experiments.


--------------------- Post rebuttal ---------------------
I have read the rebuttal and all my concerns have been addressed.

---

> ### Author Response · Authors · 2022-08-02
> **Response to Reviewer WJZn**
>
> ## Response to Reviewer WJZn
>
> We thank the reviewer for the support and the careful review. Our response to your remaining concerns follows next.
>
> **W1 & Q2. Not clear how the benign overfitting condition in (13) can be used to determine whether certain settings are benign or not. For example, $M, d$ in the settings of Example 1 and Example 2 are finite while (13) expresses the condition in the form of limit.**
>
> Indeed, the equation (13) in its current form presents the benign overfitting condition in the asymptotic case. In the finite-dimensional case, we essentially require the corresponding benign overfitting condition
>
> $$
>  r_ {0}(\overline{\mathbf{W}}_ {M}^{\mathcal{A}}) = o({MN}),
>  \quad k^* = o({MN}),
>  \quad R_ {k^*}(\overline{\mathbf{W}}_ {M}^{\mathcal{A}}) = \omega \big( MN \big).
> $$
>
> Note that in the finite dimensional case, $r_ 0(\overline{\mathbf{W}}_ {M}^{\mathcal{A}}) = \Theta(1)$, the first and second conditions are satisfied. To satisfy the third condition above, the smallest $d - k^*$ eigenvalues of $\overline{\mathbf{W}}_ {M}^{\mathcal{A}}$ need to be at least as large as a small constant or slowly decaying.
>
>
> Regarding when it violates the benign overfitting condition，we can modify the example from Theorem 6 of [5] (in the submission). Denote the model dimension as $d_{MN}$ which is changing with $MN$. If the eigenvalues of ${\mathbf{W}}_{m}^{\mathcal{A}} = {\mathbf{W}}^{\mathcal{A}}$, denoted as $\mu_i$, follow the form $\mu_i = i^{-\beta}$, then the necessary and sufficient condition for benign overfitting in this special case is either one of the two:
>
> $$
> \textrm{I}:\beta \in (0,1), d_{MN} = \omega(MN), d_{MN} = o\big((MN)^{1/(1-\beta)}\big)
> \quad
> \textrm{II}:\beta=1, d_{MN} = e^{\omega(\sqrt{MN})}, d_{MN} = e^{o(MN)}.
> $$
>
>
> If the above condition does not hold, it violates the benign overfitting condition.
>
>
>
> **W2. "Nested meta-learning" is confusing.**
>
>
> Thanks for the suggestion. We introduce this term to distinguish from "representation-based meta learning". Following your suggestion, we have changed "nested meta learning" to  "gradient-based meta learning".
>
>
> **W3&Q4. Gap between theoretical analysis and practical meta-learning methods in terms of the data, model architecture, and prediction task. Show the possible connection via discussion or experiments.**
>
>
> Thanks for the suggestion. We discuss the connection of theoretical analysis and practical meta-learning methods via both similarity and differences.
>
> 1. *Common between theory and practice.*
> One connection is to show the benign overfitting phenomenon also exists in practical meta-learning settings. For more details, please see **General Response-Q1**.
>
> 2. *Gap between theory and practice.*
> Please see **General Response-Q2**.
>
>
>
> **Q1. How to change the analysis to multi-step MAML?**
>
> We believe the analysis can be extended to multi-step MAML. For $t$-step update, the resulting mapping from ${\theta}_ {0}$ to ${\theta}_ {m}$ can be obtained in closed form by the applying the gradient update $t$ times.
>
> In this case, the closed form solution of $\hat{{\theta}}_{0}$ can still be obtained, and thus the similar analysis as the one-step case can be conducted. In addition, the larger $t$ is, the closer the solution is to the solution of iMAML. We will pursue this in our future work.
>
>
>
>
> **Q3. Why larger data heterogeneity makes it more difficult for the benign overfitting condition to be satisfied for both MAML and iMAML?**
>
> Larger data heterogeneity results in larger excess risk error bound based on our theory.
> In the examples we assume the cross-task data heterogeneity $\mathbb{V}$ is bounded. However, in practice, as number of meta-training tasks increases, $\mathbb{V}$ can grow bigger, or even in a rate faster than the decreasing rate of $k^*/({MN})$, and $MN/\big( R_{k^*}(\overline{\mathbf{W}}_{M}^{\mathcal{A}}) \big)$. In such case benign overfitting is not guaranteed. Therefore, the benign overfitting condition also depends on the growing rate of $\mathbb{V}$, the larger it is, the harder for benign overfitting condition to be satisfied.
>
> Intuitively, larger data heterogeneity among the sampled tasks means the eigenvalues of the data matrices for different tasks are more different. Meanwhile the benign overfitting condition requires the eigenvalues of the average weight matrices to be controlled. The higher data heterogeneity, the more difficult for the eigenvalues to be controlled, therefore the more difficult for benign overfitting condition to be satisfied.
>
>
> We hope the above response can resolve your concerns. Feel free to let us know if you want us to provide more clarifications.

---

### Official Review · Reviewer_sqGF · 2022-07-10

**Rating:** 6
**Confidence:** 4
**Soundness:** 3 good
**Presentation:** 3 good
**Contribution:** 3 good

**Summary:**

This paper analyzes the “benign overfitting” phenomenon of nested meta-learning, where an over-parameterized model can still work well in the meta linear regression setting. Both MAML and iMAML are covered by the analyses. Numerical simulations validate the results of the theoretical analysis.


**Questions:**

Please refer to the weakness in the previous parts.

To better show what it the “benign overfitting" in meta-learning, the authors may consider showing some empirical results of various meta-learning methods at first (even not fully consistent with the assumptions in the paper). Based on the final results of the paper, we can see the benign overfitting in the meta linear regression, but we are not sure whether the same phenomenon will occur in more meta-learning tasks.

Since the analysis is based on the case where the total number of meta-training data is smaller than the number of parameters in the model, which is not always the case in meta-learning, the authors may provide more real examples where we will meet the case.

**Limitations:**

The authors have adequately addressed the limitations and potential negative social impact of the work.


**Strengths And Weaknesses:**

The paper analyzes an interesting phenomenon in meta-learning in the meta linear regression setting. The authors need to clarify the paper only deals with the meta linear regression, which simplifies the analysis with a different thread of tools. BTW, the "nested meta-learning" is just the bi-level optimization in meta-learning. Since most meta-learning methods adopt the bi-level form, it is not necessary to introduce a new term.

In addition, the authors need to discuss the gap between the analysis and the practical MAML/iMAML, since the latter methods work on few-shot learning with novel classes. Some discussions on how the analysis can extend to practical cases or which intuitions it provides are welcome.

The difference between the tools used for the analysis in benign overfitting and [1] should be discussed.

[1] Kong et al. Meta-learning for Mixed Linear Regression. ICML 2020.

---

> ### Author Response · Authors · 2022-08-02
> **Response to Reviewer sqGF**
>
> ## Response to Reviewer sqGF
> We thank the reviewer for the support and the careful review. Our response to your remaining concerns follows below.
>
> **W1. Clarify the paper only deals with meta linear regression.**
>
> Thanks for the suggestion. Indeed, we have mentioned in the abstract, line 9 that "our analysis uses the relatively tractable linear models"; in line 150 we have mentioned that our analysis uses "linear data model"; in the conclusion section, line 325, we have mentioned that "we focus on linear models".
> Following your suggestion, we have further emphasized this point in several places including i) the contribution part in Section 1.2; ii) the subtitles and key proposition in Section 3.1; and iii) the key statement in Section 3.2 of the revised version.
>
>
>
> **W2. Not necessary to introduce "nested meta learning".**
>
> Thanks for the suggestion. We introduce this term to distinguish our work from "representation-based meta learning". We will change "nested meta learning" to  "gradient-based meta learning" which is also suggested by Reviewer WJZn. Following your suggestion, we have revised this in the revised version.
>
>
> **W3. Discuss the gap between the analysis and practice and how the analysis can be extended to practical cases or which intuitions it provide.**
>
> Thanks for the suggestion.
>
> 1. *Gap between analysis and practice.*
>
> Please see **General Response-Q2**.
>
> 2. *Practical case where the model works on few-shot learning with novel classes.*
>
> Our analysis does handle few-shot learning with novel tasks sampled from the same underlying task distribution. Few-shot learning means the number of training data per task $N$ is small. During meta-testing, we use the meta-trained model to predict on novel tasks sampled from the same underlying task distribution as the meta-training data. One limitation is that our analysis focuses on meta linear regression, but not classification. Extension to analysis on meta classification will be our future work. When additional page is allowed, we will clarify this in the next version of the paper.
>
>
>
>
> **W4. Difference between the tools used for the analysis in this paper and [1].**
>
>
> In general, both [1] and our paper use random matrix theory as the main tool to study statistical properties of the meta linear regression. However, [1] focuses on sample complexity to reach certain statistical error while we focus on how the excess risk, a measure of the generalization ability, depends on the task and data distribution.
>
> Nevertheless, the settings and the specific techniques in our paper differ from those in [1]. To be more specific, we list some of the major differences below.
>
> 1. Referece [1] studies meta learning under mixed linear regression, which differs from our data model. For example, [1] assumes the data matrix is isotropic, i.e. $E[x_m x_m^{\top}]=I_d$.
> In contrast, our paper does not assume the data matrix is isotropic. In fact, isotropic data matrix does not satisfy the asymptotic benign overfitting condition. We show that when the eigenvalues of the data matrix satisfy certain conditions, the overparameterized model can still generalize well. And how the excess risk changes with the eigenvalue distribution of the data matrix is therefore the focus of our study.
>
> 2. Referece [1] also assumes that the sufficient meta-training tasks are given, or the overparameterization happens, if any, only in the per-task level. The focus of their work is to study when can abundant tasks with small data compensate for lack of tasks with big data. And they identify sufficient conditions for this. To achieve this, the key technique they use is the concentration inequality when sufficient number of tasks or sufficient number of data per task are given.
>
> 3. Referece [1] assumes the task parameters come from a discrete distribution of a fixed support size while we do not assume a discrete distribution of the task parameters. And [1] develops an algorithm that first clusters these tasks and then estimates the task parameters. The algorithm is different from MAML, which is the focus of our study.
>
> To summarize, they mainly focus on analyzing sample complexity in isotropic data case without overparameterization in the meta level, while we focus on studying the benign overfitting condition in the overparameterized case. We have also included comparison with this work in Table 1 in Section 1.2 of the revised paper.
>
> >[1] Kong et al. Meta-learning for Mixed Linear Regression. ICML, 2020.

---

> > ### Comment · Reviewer_sqGF · 2022-08-08
> > **Thanks for the response**
> >
> > I have read the response, and the authors have addressed my concerns.

---

> ### Author Response · Authors · 2022-08-03
> **Response to Reviewer sqGF - Questions**
>
> **Q1. Add empirical results of various meta learning methods at first to show whether benign overfitting phenomenon will occur in more meta-learning tasks.**
>
> Thanks for the great suggestion! Following your suggestion, we have provided some empirical results in **General Response-Q1**. The empirical results demonstrate that the benign overfitting phenomenon will occur in more practical meta-learning tasks such as few-shot meta learning for image classification.
>
> We will include this in the revision of the main text if additional page is allowed.
>
>
> **Q2. Real examples where the number of meta training data is smaller than the number of parameters.**
>
>
>
> As suggested, we have provided real examples in **General Response-Q1**.
>
>
> We greatly appreciate your helpful comments. We hope these can resolve your concerns.

---

### Official Review · Reviewer_TRZn · 2022-07-11

**Rating:** 7
**Confidence:** 1
**Soundness:** 3 good
**Presentation:** 3 good
**Contribution:** 3 good

**Summary:**

This paper explores the generalization of overparameterized meta-learning models with a nested structure, specifically model-agnostic meta-learning (MAML) and implicit MAML (iMAML). Under some theoretical assumptions, the authors derive an upper bound for the excess risk of MAML and iMAML in the overparameterized setting. Further, they derive conditions for benign overfitting in nested meta-learning and provide hyperparameters for MAML and iMAML that preserve the benign overfitting.

**Questions:**

Is there an intuitive way to explain why the cross-task variance upper bound primarily depends on 1/M asymptotically?

**Limitations:**

The authors adequately address the limitations of their work. Specifically, they acknowledge that their current analysis is limited to the non-Bayesian setting and the linear model assumption.

**Strengths And Weaknesses:**

The paper does a good job of explaining the motivations for studying the problem. In addition, the authors present their contributions clearly and provide a proof outline of their main result, which contributes to the clarity of the paper.

The main weakness of the paper is in the discussion of the experimental results. I believe that the significance of the theoretical results could be further emphasized by explaining the experimental results in more depth (and adding some numerical results in a tabular format).

---

> ### Author Response · Authors · 2022-08-02
> **Response to Reviewer TRZn**
>
> ## Response to Reviewer TRZn
>
> We thank the reviewer for the support and the careful review. Our response to your comments follows.
>
> **W1. Explain the experimental results in more depth and add some numerical results in a tabular format.**
>
> Thanks for the suggestion. We have added more explanation in the revised version of the paper and we also included numerical results in the tabular format. See Section E. Additional experiments in the appendix.
>
>
>
>
> **Q1. Why the cross-task variance upper bound primarily depends on $1/M$ asymptotically?**
>
>
> Since we assume the tasks are sampled from an underlying task distribution, as the number of tasks $M$ grows larger, the average statistics across tasks converge to the expected statistics across tasks. Intuitively, sample average of independent but non-identical random variables has variance that decreases with the number of samples, which is proportional to $1/M$.

---

> > ### Comment · Reviewer_TRZn · 2022-08-09
> > **Thank you for the clarifications**
> >
> > I acknowledge that I have read the authors’ responses and thank them for positively addressing my comments.

---

### Author Response · Authors · 2022-08-02
**General Response**

## General Response

We thank the reviewers for their constructive comments! We will first address the common questions raised by several reviewers and then respond to each reviewer.

**Q1. Connection to practical meta-learning cases (All reviewers).**

Reviewers' commments can be summarized into the following two subquestions.

**Q1.1** Whether meta-learning model is overparameterized in practice given multiple tasks?

**Q1.2** Does benign overfitting happen in practice?



Indeed, overparameterization and benign overfitting happen in meta learning empirically. Below we list some practical data quoted from existing works on empirical study of MAML. We estimate the total number of meta-training data by the meta-training iterations times the batch size (task number for each iteration) times the number of shots.


For  **Q1.1 overparameterization**, the following table quotes results from Table 2 in [1] and Table A5 in [2], which clearly shows that ResNet-10 and ResNet-12 are overparameterized in the meta level while conv-4 is underparameterized in the meta-training level. Please note that ResNet-10 and ResNet-12 outperform conv-4, which means overparameterization in the meta level does not harm testing performance.


| Model (on mini-ImageNet)    |MAML Conv-4 [2]  |MAML ResNet-10 [2] | MAML ResNet-12 [1] |
| :---     |   :----: |:----: |---:|
| # params |    0.07M |  6M  | 8M|
| # meta-training data (1 shot)| 1M |  1M   | 0.3M|
| # meta-training data (5 shot)| 3M |  3M   | 1.5M|
| Test accuracy (1 shot) | 46.47±0.82 |  54.69±0.89  |    51.03±0.50  |
| Test accuracy (5 shot) | 62.71±0.71  |  66.62±0.83  |    68.26±0.47  |

For  **Q1.2 benign overfitting**, we copy Figure 3 in [2] below (if the link is not visible in OpenReview, it can be viewed in a browser). Using ResNet-12 or larger model, the number of model parameters are usually orders of magnitute higher than the total number of meta-training data, especially in 1-shot learning. It can be seen that for in almost all experiments, the testing performance of MAML improves as the number of model parameters increases (e.g. ResNet compared to ConvNet). Therefore, the benign overfitting happens in these practical cases.



![](https://i.imgur.com/MJe0AQe.png)



>[1] Towards Gradient-based Bilevel Optimization with Non-convex Followers and Beyond. NeurIPS, 2021.
>
>[2] A closer look at few-shot classification. ICLR, 2019.


**Q2. Gap between theory and practice (data, model, and task). How the theory can be extended to practical cases or which intuitions it provide? (Reviewers sqGF, WJZn)**

1. *Connection and gap between the analysis and practice.*
**General Response-Q1** shows that overparameterization in the meta level does happen in practice such as in *few-shot* image classification. In addition, benign overfitting happens in the meta learning settings. These observations justify that the benign overfitting of meta learning that the paper studied indeed happens in practice. The gap is that our current analysis focuses on the meta linear models in the regression settings while the practical meta learning may often use nonlinear models in the classification settings.



2. *How our analysis can be extended.*
Although our analysis focuses on the linear models, it can still provide the fundamental insights on the benign overfitting in practical meta learning cases where we reuse the feature extractor from pre-trained models and only meta-train the parameters in the last linear layer for regression problems; see [3]. Then the feature vector $x_m$ in our analysis represents the last feature layer output from the feature extractor. In addition, linear models have been frequently used in the recent generalization analysis of MAML (e.g., [3, 9, 13, 21] in the submission) since the dynamics of SGD on overparameterized neural networks can be effectively captured by that on linear models. Going beyond the considered linear models, it is also promising and possible to extend our analysis to nonlinear cases via means of random features and neural tangent kernels. In addition, while our current analysis focuses on the regression task, it is promising to build upon our work and the recent work on benign overfitting in classification (e.g., [4]) to study the benign overfitting in the meta classification task.

>[3] Few-shot Image Classification: Just Use a Library of Pre-trained Feature Extractors and a Simple Classifier. ICCV, 2021.
>
>[4] Benign overfitting in multiclass classification: All roads lead to interpolation. NeurIPS 2021

---

### Meta-Review · Area_Chair_wGiJ · 2022-08-27

**Recommendation:** Accept
**Confidence:** Certain

**Metareview:**

This paper explores the generalization of minimum norm optima for various meta-learning objectives, including basic ERM, model-agnostic meta-learning (MAML) and implicit MAML (iMAML). The generative model considered is "mixed linear regression", in which each of tasks follows a linear + Gaussian noise data model (a different direction per task). The main conceptual takeaway is to tease out how the task heterogeneity affects the generalization bounds (through a "cross variance" quantity), and to compare how much "overfitting" happens for the different objectives. The proof techniques largely follow [Bartlett et al. PNAS 2020] --- due to the linearity/Gaussianity, it boils down to a series of concentration bounds.  Since data coming from different tasks has a different SVD, this makes the proofs non-trivial to extend.

**Award:**

No

---

### Decision · Program_Chairs · 2022-09-14

Accept